# DPM-Solver-v3: Improved Diffusion ODE Solver with Empirical Model Statistics

**Kaiwen Zheng**[*†1], **Cheng Lu**[*1], **Jianfei Chen**[1], **Jun Zhu**[‡123]

[1]Dept. of Comp. Sci. & Tech., Institute for AI, BNRist Center, THBI Lab
[1]Tsinghua-Bosch Joint ML Center, Tsinghua University, Beijing, China
[2]Shengshu Technology, Beijing    [3]Pazhou Lab (Huangpu), Guangzhou, China
{zkwthu,lucheng.lc15}@gmail.com; {jianfeic, dcszj}@tsinghua.edu.cn

## Abstract

Diffusion probabilistic models (DPMs) have exhibited excellent performance for high-fidelity image generation while suffering from inefficient sampling. Recent works accelerate the sampling procedure by proposing fast ODE solvers that leverage the specific ODE form of DPMs. However, they highly rely on specific parameterization during inference (such as noise/data prediction), which might not be the optimal choice. In this work, we propose a novel formulation towards the optimal parameterization during sampling that minimizes the first-order discretization error of the ODE solution. Based on such formulation, we propose *DPM-Solver-v3*, a new fast ODE solver for DPMs by introducing several coefficients efficiently computed on the pretrained model, which we call *empirical model statistics*. We further incorporate multistep methods and a predictor-corrector framework, and propose some techniques for improving sample quality at small numbers of function evaluations (NFE) or large guidance scales. Experiments show that DPM-Solver-v3 achieves consistently better or comparable performance in both unconditional and conditional sampling with both pixel-space and latent-space DPMs, especially in 5∼10 NFEs. We achieve FIDs of 12.21 (5 NFE), 2.51 (10 NFE) on unconditional CIFAR10, and MSE of 0.55 (5 NFE, 7.5 guidance scale) on Stable Diffusion, bringing a speed-up of 15%∼30% compared to previous state-of-the-art training-free methods. Code is available at https://github.com/thu-ml/DPM-Solver-v3.

## 1 Introduction

Diffusion probabilistic models (DPMs) [47, 15, 51] are a class of state-of-the-art image generators. By training with a strong encoder, a large model, and massive data as well as leveraging techniques such as guided sampling, DPMs are capable of generating high-resolution photorealistic and artistic images on text-to-image tasks. However, to generate high-quality visual content, DPMs usually require hundreds of model evaluations to gradually remove noise using a pretrained model [15], which is much more time-consuming compared to other deep generative models such as generative adversarial networks (GANs) [13]. The sampling overhead of DPMs emerges as a crucial obstacle hindering their integration into downstream tasks.

To accelerate the sampling process of DPMs, one can employ training-based methods [37, 53, 45] or training-free samplers [48, 51, 28, 3, 52, 56]. We focus on the latter approach since it requires no

---

[†]Work done during an internship at Shengshu Technology
[*]Equal contribution
[‡]Corresponding author

37th Conference on Neural Information Processing Systems (NeurIPS 2023).

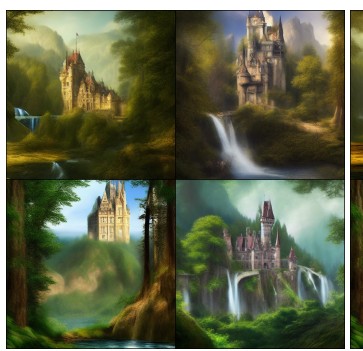 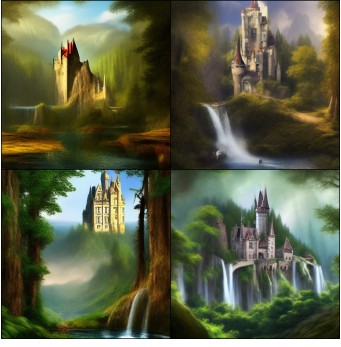 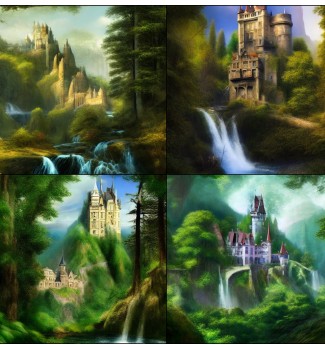

(a) DPM-Solver++ [32] (MSE 0.60)  (b) UniPC [58] (MSE 0.65)  (c) DPM-Solver-v3 (**Ours**, MSE 0.55)

Figure 1: Random samples of Stable-Diffusion [43] with a classifier-free guidance scale 7.5, using only 5 number of function evaluations (NFE) and text prompt *"A beautiful castle beside a waterfall in the woods, by Josef Thoma, matte painting, trending on artstation HQ"*.

extra training and is more flexible. Recent advanced training-free samplers [56, 31, 32, 58] mainly rely on the ODE form of DPMs, since its absence of stochasticity is essential for high-quality samples in around 20 steps. Besides, ODE solvers built on exponential integrators [18] converge faster. To change the original diffusion ODE into the form of exponential integrators, they need to cancel its linear term and obtain an ODE solution, where only the noise predictor needs to be approximated, and other terms can be exactly computed. Besides, Lu et al. [32] find that it is better to use another ODE solution where instead the data predictor needs to be approximated. How to choose such model parameterization (e.g., noise/data prediction) in the sampling of DPMs remains unrevealed.

In this work, we systematically study the problem of model parameterization and ODE formulation for fast sampling of DPMs. We first show that by introducing three types of coefficients, the original ODE solution can be reformulated to an equivalent one that contains a new model parameterization. Besides, inspired by exponential Rosenbrock-type methods [19] and first-order discretization error analysis, we also show that there exists an optimal set of coefficients efficiently computed on the pretrained model, which we call *empirical model statistics* (EMS). Building upon our novel ODE formulation, we further propose a new high-order solver for diffusion ODEs named *DPM-Solver-v3*, which includes a multistep predictor-corrector framework of any order, as well as some novel techniques such as pseudo high-order method to boost the performance at extremely few steps or large guidance scale.

We conduct extensive experiments with both pixel-space and latent-space DPMs to verify the effectiveness of DPM-Solver-v3. Compared to previous fast samplers, DPM-Solver-v3 can consistently improve the sample quality in 5∼20 steps, and make a significant advancement within 10 steps.

## 2 Background

### 2.1 Diffusion Probabilistic Models

Suppose we have a $D$-dimensional data distribution $q_0(\boldsymbol{x}_0)$. Diffusion probabilistic models (DPMs) [47, 15, 51] define a forward diffusion process $\{q_t\}_{t=0}^T$ to gradually degenerate the data $\boldsymbol{x}_0 \sim q_0(\boldsymbol{x}_0)$ with Gaussian noise, which satisfies the transition kernel $q_{0t}(\boldsymbol{x}_t|\boldsymbol{x}_0) = \mathcal{N}(\alpha_t\boldsymbol{x}_0, \sigma_t^2\boldsymbol{I})$, such that $q_T(\boldsymbol{x}_T)$ is approximately pure Gaussian. $\alpha_t, \sigma_t$ are smooth scalar functions of $t$, which are called *noise schedule*. The transition can be easily applied by $\boldsymbol{x}_t = \alpha_t\boldsymbol{x}_0 + \sigma_t\boldsymbol{\epsilon}, \boldsymbol{\epsilon} \sim \mathcal{N}(\boldsymbol{0}, \boldsymbol{I})$. To train DPMs, a neural network $\boldsymbol{\epsilon}_\theta(\boldsymbol{x}_t, t)$ is usually parameterized to predict the noise $\boldsymbol{\epsilon}$ by minimizing $\mathbb{E}_{\boldsymbol{x}_0 \sim q_0(\boldsymbol{x}_0), \boldsymbol{\epsilon} \sim \mathcal{N}(\boldsymbol{0}, \boldsymbol{I}), t \sim \mathcal{U}(0, T)}\left[w(t)\|\boldsymbol{\epsilon}_\theta(\boldsymbol{x}_t, t) - \boldsymbol{\epsilon}\|_2^2\right]$, where $w(t)$ is a weighting function. Sampling of DPMs can be performed by solving *diffusion ODE* [51] from time $T$ to time $0$:

$$\frac{\mathrm{d}\boldsymbol{x}_t}{\mathrm{d}t} = f(t)\boldsymbol{x}_t + \frac{g^2(t)}{2\sigma_t}\boldsymbol{\epsilon}_\theta(\boldsymbol{x}_t, t), \tag{1}$$

where $f(t) = \frac{\mathrm{d}\log\alpha_t}{\mathrm{d}t}$, $g^2(t) = \frac{\mathrm{d}\sigma_t^2}{\mathrm{d}t} - 2\frac{\mathrm{d}\log\alpha_t}{\mathrm{d}t}\sigma_t^2$ [23]. In addition, the conditional sampling by DPMs can be conducted by guided sampling [10, 16] with a conditional noise predictor $\boldsymbol{\epsilon}_\theta(\boldsymbol{x}_t, t, c)$, where $c$ is the condition. Specifically, classifier-free guidance [16] combines the unconditional/conditional model and obtains a new noise predictor $\boldsymbol{\epsilon}'_\theta(\boldsymbol{x}_t, t, c) := s\boldsymbol{\epsilon}_\theta(\boldsymbol{x}_t, t, c) + (1-s)\boldsymbol{\epsilon}_\theta(\boldsymbol{x}_t, t, \emptyset)$,

where $\emptyset$ is a special condition standing for the unconditional case, $s > 0$ is the guidance scale that controls the trade-off between image-condition alignment and diversity; while classifier guidance [10] uses an extra classifier $p_\phi(c|\boldsymbol{x}_t, t)$ to obtain a new noise predictor $\boldsymbol{\epsilon}'_\theta(\boldsymbol{x}_t, t, c) := \boldsymbol{\epsilon}_\theta(\boldsymbol{x}_t, t) - s\sigma_t \nabla_{\boldsymbol{x}} \log p_\phi(c|\boldsymbol{x}_t, t)$.

In addition, except for the noise prediction, DPMs can be parameterized as score predictor $\boldsymbol{s}_\theta(\boldsymbol{x}_t, t)$ to predict $\nabla_{\boldsymbol{x}} \log q_t(\boldsymbol{x}_t, t)$, or data predictor $\boldsymbol{x}_\theta(\boldsymbol{x}_t, t)$ to predict $\boldsymbol{x}_0$. Under variance-preserving (VP) noise schedule which satisfies $\alpha_t^2 + \sigma_t^2 = 1$ [51], "v" predictor $\boldsymbol{v}_\theta(\boldsymbol{x}_t, t)$ is also proposed to predict $\alpha_t \boldsymbol{\epsilon} - \sigma_t \boldsymbol{x}_0$ [45]. These different parameterizations are theoretically equivalent, but have an impact on the empirical performance when used in training [23, 59].

## 2.2 Fast Sampling of DPMs with Exponential Integrators

Among the methods for solving the diffusion ODE (1), recent works [56, 31, 32, 58] find that ODE solvers based on exponential integrators [18] are more efficient and robust at a small number of function evaluations (<50). Specifically, an insightful observation by Lu et al. [31] is that, by change-of-variable from $t$ to $\lambda_t := \log(\alpha_t/\sigma_t)$ (half of the log-SNR), the diffusion ODE is transformed to

$$\frac{\mathrm{d}\boldsymbol{x}_\lambda}{\mathrm{d}\lambda} = \frac{\dot{\alpha}_\lambda}{\alpha_\lambda}\boldsymbol{x}_\lambda - \sigma_\lambda \boldsymbol{\epsilon}_\theta(\boldsymbol{x}_\lambda, \lambda), \tag{2}$$

where $\dot{\alpha}_\lambda := \frac{\mathrm{d}\alpha_\lambda}{\mathrm{d}\lambda}$. By utilizing the semi-linear structure of the diffusion ODE and exactly computing the linear term [56, 31], we can obtain the ODE solution as Eq. (3) (left). Such exact computation of the linear part reduces the discretization errors [31]. Moreover, by leveraging the equivalence of different parameterizations, DPM-Solver++ [32] rewrites Eq. (2) by the data predictor $\boldsymbol{x}_\theta(\boldsymbol{x}_\lambda, \lambda) := (\boldsymbol{x}_\lambda - \sigma_\lambda \boldsymbol{\epsilon}_\theta(\boldsymbol{x}_\lambda, \lambda))/\alpha_\lambda$ and obtains another ODE solution as Eq. (3) (right). Such solution does not need to change the pretrained noise prediction model $\boldsymbol{\epsilon}_\theta$ during the sampling process, and empirically outperforms previous samplers based on $\boldsymbol{\epsilon}_\theta$ [31].

$$\frac{\boldsymbol{x}_t}{\alpha_t} = \frac{\boldsymbol{x}_s}{\alpha_s} - \int_{\lambda_s}^{\lambda_t} e^{-\lambda} \boldsymbol{\epsilon}_\theta(\boldsymbol{x}_\lambda, \lambda) \mathrm{d}\lambda, \quad \frac{\boldsymbol{x}_t}{\sigma_t} = \frac{\boldsymbol{x}_s}{\sigma_s} + \int_{\lambda_s}^{\lambda_t} e^{\lambda} \boldsymbol{x}_\theta(\boldsymbol{x}_\lambda, \lambda) \mathrm{d}\lambda \tag{3}$$

However, to the best of our knowledge, the parameterizations for sampling are still manually selected and limited to noise/data prediction, which are not well-studied.

## 3 Method

We now present our method. We start with a new formulation of the ODE solution with extra coefficients, followed by our high-order solver and some practical considerations. In the following discussions, we assume all the products between vectors are element-wise, and $\boldsymbol{f}^{(k)}(\boldsymbol{x}_\lambda, \lambda) = \frac{\mathrm{d}^k \boldsymbol{f}(\boldsymbol{x}_\lambda, \lambda)}{\mathrm{d}\lambda^k}$ is the $k$-th order total derivative of any function $\boldsymbol{f}$ w.r.t. $\lambda$.

### 3.1 Improved Formulation of Exact Solutions of Diffusion ODEs

As mentioned in Sec. 2.2, it is promising to explore the semi-linear structure of diffusion ODEs for fast sampling [56, 31, 32]. *Firstly*, we reveal one key insight that we can choose the linear part according to Rosenbrock-type exponential integrators [19, 18]. To this end, we consider a general form of diffusion ODEs by rewriting Eq. (2) as

$$\frac{\mathrm{d}\boldsymbol{x}_\lambda}{\mathrm{d}\lambda} = \underbrace{\left(\frac{\dot{\alpha}_\lambda}{\alpha_\lambda} - \boldsymbol{l}_\lambda\right)\boldsymbol{x}_\lambda}_{\text{linear part}} - \underbrace{(\sigma_\lambda \boldsymbol{\epsilon}_\theta(\boldsymbol{x}_\lambda, \lambda) - \boldsymbol{l}_\lambda \boldsymbol{x}_\lambda)}_{\text{non-linear part}, := \boldsymbol{N}_\theta(\boldsymbol{x}_\lambda, \lambda)}, \tag{4}$$

where $\boldsymbol{l}_\lambda$ is a $D$-dimensional undetermined coefficient depending on $\lambda$. We choose $\boldsymbol{l}_\lambda$ to restrict the Frobenius norm of the gradient of the non-linear part w.r.t. $\boldsymbol{x}$:

$$\boldsymbol{l}_\lambda^* = \underset{\boldsymbol{l}_\lambda}{\arg\min} \, \mathbb{E}_{p_\lambda^\theta(\boldsymbol{x}_\lambda)} \|\nabla_{\boldsymbol{x}} \boldsymbol{N}_\theta(\boldsymbol{x}_\lambda, \lambda)\|_F^2, \tag{5}$$

where $p_\lambda^\theta$ is the distribution of samples on the ground-truth ODE trajectories at $\lambda$ (i.e., model distribution). Intuitively, it makes $\boldsymbol{N}_\theta$ insensitive to the errors of $\boldsymbol{x}$ and cancels all the "linearty" of

$N_\theta$. With $l_\lambda = l_\lambda^*$, by the "*variation-of-constants*" formula [1], starting from $\boldsymbol{x}_{\lambda_s}$ at time $s$, the exact solution of Eq. (4) at time $t$ is

$$\boldsymbol{x}_{\lambda_t} = \alpha_{\lambda_t} e^{-\int_{\lambda_s}^{\lambda_t} \boldsymbol{l}_\lambda \mathrm{d}\lambda} \left( \frac{\boldsymbol{x}_{\lambda_s}}{\alpha_{\lambda_s}} - \int_{\lambda_s}^{\lambda_t} e^{\int_{\lambda_s}^{\lambda} \boldsymbol{l}_\tau \mathrm{d}\tau} \underbrace{\boldsymbol{f}_\theta(\boldsymbol{x}_\lambda, \lambda)}_{\text{approximated}} \mathrm{d}\lambda \right), \tag{6}$$

where $\boldsymbol{f}_\theta(\boldsymbol{x}_\lambda, \lambda) := \frac{\boldsymbol{N}_\theta(\boldsymbol{x}_\lambda, \lambda)}{\alpha_\lambda}$. To calculate the solution in Eq. (6), we need to approximate $\boldsymbol{f}_\theta$ for each $\lambda \in [\lambda_s, \lambda_t]$ by certain polynomials [31, 32].

*Secondly*, we reveal another key insight that we can choose different functions to be approximated instead of $\boldsymbol{f}_\theta$ and further reduce the discretization error, which is related to the total derivatives of the approximated function. To this end, we consider a scaled version of $\boldsymbol{f}_\theta$ i.e., $\boldsymbol{h}_\theta(\boldsymbol{x}_\lambda, \lambda) := e^{-\int_{\lambda_s}^{\lambda} \boldsymbol{s}_\tau \mathrm{d}\tau} \boldsymbol{f}_\theta(\boldsymbol{x}_\lambda, \lambda)$ where $\boldsymbol{s}_\lambda$ is a $D$-dimensional coefficient dependent on $\lambda$, and then Eq. (6) becomes

$$\boldsymbol{x}_{\lambda_t} = \alpha_{\lambda_t} e^{-\int_{\lambda_s}^{\lambda_t} \boldsymbol{l}_\lambda \mathrm{d}\lambda} \left( \frac{\boldsymbol{x}_{\lambda_s}}{\alpha_{\lambda_s}} - \int_{\lambda_s}^{\lambda_t} e^{\int_{\lambda_s}^{\lambda} (\boldsymbol{l}_\tau + \boldsymbol{s}_\tau) \mathrm{d}\tau} \underbrace{\boldsymbol{h}_\theta(\boldsymbol{x}_\lambda, \lambda)}_{\text{approximated}} \mathrm{d}\lambda \right). \tag{7}$$

Comparing with Eq. (6), we change the approximated function from $\boldsymbol{f}_\theta$ to $\boldsymbol{h}_\theta$ by using an additional scaling term related to $\boldsymbol{s}_\lambda$. As we shall see, the first-order discretization error is positively related to the norm of the first-order derivative $\boldsymbol{h}_\theta^{(1)} = e^{-\int_{\lambda_s}^{\lambda} \boldsymbol{s}_\tau \mathrm{d}\tau}(\boldsymbol{f}_\theta^{(1)} - \boldsymbol{s}_\lambda \boldsymbol{f}_\theta)$. Thus, we aim to minimize $\|\boldsymbol{f}_\theta^{(1)} - \boldsymbol{s}_\lambda \boldsymbol{f}_\theta\|_2$, in order to reduce $\|\boldsymbol{h}_\theta^{(1)}\|_2$ and the discretization error. As $\boldsymbol{f}_\theta$ is a fixed function depending on the pretrained model, this minimization problem essentially finds a linear function of $\boldsymbol{f}_\theta$ to approximate $\boldsymbol{f}_\theta^{(1)}$. To achieve better linear approximation, we further introduce a bias term $\boldsymbol{b}_\lambda \in \mathbb{R}^D$ and construct a function $\boldsymbol{g}_\theta$ satisfying $\boldsymbol{g}_\theta^{(1)} = e^{-\int_{\lambda_s}^{\lambda} \boldsymbol{s}_\tau \mathrm{d}\tau}(\boldsymbol{f}_\theta^{(1)} - \boldsymbol{s}_\lambda \boldsymbol{f}_\theta - \boldsymbol{b}_\lambda)$, which gives

$$\boldsymbol{g}_\theta(\boldsymbol{x}_\lambda, \lambda) := e^{-\int_{\lambda_s}^{\lambda} \boldsymbol{s}_\tau \mathrm{d}\tau} \boldsymbol{f}_\theta(\boldsymbol{x}_\lambda, \lambda) - \int_{\lambda_s}^{\lambda} e^{-\int_{\lambda_s}^{r} \boldsymbol{s}_\tau \mathrm{d}\tau} \boldsymbol{b}_r \mathrm{d}r. \tag{8}$$

With $\boldsymbol{g}_\theta$, Eq. (7) becomes

$$\boldsymbol{x}_{\lambda_t} = \alpha_{\lambda_t} \underbrace{e^{-\int_{\lambda_s}^{\lambda_t} \boldsymbol{l}_\lambda \mathrm{d}\lambda}}_{\text{linear coefficient}} \left( \frac{\boldsymbol{x}_{\lambda_s}}{\alpha_{\lambda_s}} - \int_{\lambda_s}^{\lambda_t} \underbrace{e^{\int_{\lambda_s}^{\lambda} (\boldsymbol{l}_\tau + \boldsymbol{s}_\tau) \mathrm{d}\tau}}_{\text{scaling coefficient}} \left( \underbrace{\boldsymbol{g}_\theta(\boldsymbol{x}_\lambda, \lambda)}_{\text{approximated}} + \underbrace{\int_{\lambda_s}^{\lambda} e^{-\int_{\lambda_s}^{r} \boldsymbol{s}_\tau \mathrm{d}\tau} \boldsymbol{b}_r \mathrm{d}r}_{\text{bias coefficient}} \right) \mathrm{d}\lambda \right). \tag{9}$$

Such formulation is equivalent to Eq. (3) but introduces three types of coefficients and a new parameterization $\boldsymbol{g}_\theta$. We show in Appendix I.1 that the generalized parameterization $\boldsymbol{g}_\theta$ in Eq. (8) can cover a wide range of parameterization families in the form of $\boldsymbol{\psi}_\theta(\boldsymbol{x}_\lambda, \lambda) = \boldsymbol{\alpha}(\lambda)\boldsymbol{\epsilon}_\theta(\boldsymbol{x}_\lambda, \lambda) + \boldsymbol{\beta}(\lambda)\boldsymbol{x}_\lambda + \boldsymbol{\gamma}(\lambda)$. We aim to reduce the discretization error by finding better coefficients than previous works [31, 32].

Now we derive the concrete formula for analyzing the first-order discretization error. By replacing $\boldsymbol{g}_\theta(\boldsymbol{x}_\lambda, \lambda)$ with $\boldsymbol{g}_\theta(\boldsymbol{x}_{\lambda_s}, \lambda_s)$ in Eq. (9), we obtain the first-order approximation $\hat{\boldsymbol{x}}_{\lambda_t} = \alpha_{\lambda_t} e^{-\int_{\lambda_s}^{\lambda_t} \boldsymbol{l}_\lambda \mathrm{d}\lambda} \left( \frac{\boldsymbol{x}_{\lambda_s}}{\alpha_{\lambda_s}} - \int_{\lambda_s}^{\lambda_t} e^{\int_{\lambda_s}^{\lambda} (\boldsymbol{l}_\tau + \boldsymbol{s}_\tau) \mathrm{d}\tau} \left( \boldsymbol{g}_\theta(\boldsymbol{x}_{\lambda_s}, \lambda_s) + \int_{\lambda_s}^{\lambda} e^{-\int_{\lambda_s}^{r} \boldsymbol{s}_\tau \mathrm{d}\tau} \boldsymbol{b}_r \mathrm{d}r \right) \mathrm{d}\lambda \right)$. As $\boldsymbol{g}_\theta(\boldsymbol{x}_{\lambda_s}, \lambda_s) = \boldsymbol{g}_\theta(\boldsymbol{x}_\lambda, \lambda) + (\lambda_s - \lambda)\boldsymbol{g}_\theta^{(1)}(\boldsymbol{x}_\lambda, \lambda) + \mathcal{O}((\lambda - \lambda_s)^2)$ by Taylor expansion, it follows that the first-order discretization error can be expressed as

$$\hat{\boldsymbol{x}}_{\lambda_t} - \boldsymbol{x}_{\lambda_t} = \alpha_{\lambda_t} e^{-\int_{\lambda_s}^{\lambda_t} \boldsymbol{l}_\lambda \mathrm{d}\lambda} \left( \int_{\lambda_s}^{\lambda_t} e^{\int_{\lambda_s}^{\lambda} \boldsymbol{l}_\tau \mathrm{d}\tau} (\lambda - \lambda_s) \Big( \boldsymbol{f}_\theta^{(1)}(\boldsymbol{x}_\lambda, \lambda) - \boldsymbol{s}_\lambda \boldsymbol{f}_\theta(\boldsymbol{x}_\lambda, \lambda) - \boldsymbol{b}_\lambda \Big) \mathrm{d}\lambda \right) + \mathcal{O}(h^3), \tag{10}$$

where $h = \lambda_t - \lambda_s$. Thus, given the optimal $\boldsymbol{l}_\lambda = \boldsymbol{l}_\lambda^*$ in Eq. (5), the discretization error $\hat{\boldsymbol{x}}_{\lambda_t} - \boldsymbol{x}_{\lambda_t}$ mainly depends on $\boldsymbol{f}_\theta^{(1)} - \boldsymbol{s}_\lambda \boldsymbol{f}_\theta - \boldsymbol{b}_\lambda$. Based on this insight, we choose the coefficients $\boldsymbol{s}_\lambda, \boldsymbol{b}_\lambda$ by solving

$$\boldsymbol{s}_\lambda^*, \boldsymbol{b}_\lambda^* = \underset{\boldsymbol{s}_\lambda, \boldsymbol{b}_\lambda}{\operatorname{argmin}} \, \mathbb{E}_{p_\lambda^\theta(\boldsymbol{x}_\lambda)} \left[ \left\| \boldsymbol{f}_\theta^{(1)}(\boldsymbol{x}_\lambda, \lambda) - \boldsymbol{s}_\lambda \boldsymbol{f}_\theta(\boldsymbol{x}_\lambda, \lambda) - \boldsymbol{b}_\lambda \right\|_2^2 \right]. \tag{11}$$

For any $\lambda$, $\boldsymbol{l}_\lambda^*, \boldsymbol{s}_\lambda^*, \boldsymbol{b}_\lambda^*$ all have analytic solutions involving the Jacobian-vector-product of the pretrained model $\boldsymbol{\epsilon}_\theta$, and they can be unbiasedly evaluated on a few datapoints $\{\boldsymbol{x}_\lambda^{(n)}\}_K \sim p_\lambda^\theta(\boldsymbol{x}_\lambda)$ via Monte-Carlo estimation (detailed in Section 3.4 and Appendix C.1.1). Therefore, we call $\boldsymbol{l}_\lambda, \boldsymbol{s}_\lambda, \boldsymbol{b}_\lambda$ *empirical model statistics* (EMS). In the following sections, we'll show that by approximating $\boldsymbol{g}_\theta$ with Taylor expansion, we can develop our high-order solver for diffusion ODEs.

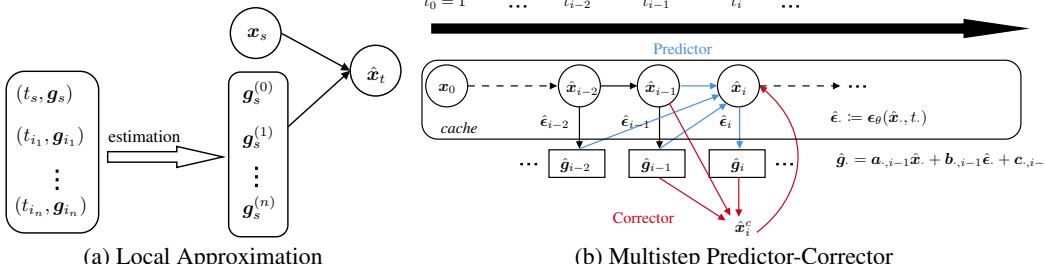

(a) Local Approximation  (b) Multistep Predictor-Corrector

Figure 2: Illustration of our high-order solver. (a) $(n+1)$-th order local approximation from time $s$ to time $t$, provided $n$ extra function values of $\boldsymbol{g}_\theta$. (b) Multistep predictor-corrector procedure as our global solver. A combination of second-order predictor and second-order corrector is shown. $a_{\cdot,i-1}, b_{\cdot,i-1}, c_{\cdot,i-1}$ are abbreviations of coefficients in Eq. (8).

## 3.2 Developing High-Order Solver

In this section, we propose our high-order solver for diffusion ODEs with local accuracy and global convergence order guarantee by leveraging our proposed solution formulation in Eq. (9). The proposed solver and analyses are highly motivated by the methods of exponential integrators [17, 18] in the ODE literature and their previous applications in the field of diffusion models [56, 31, 32, 58]. Though the EMS are designed to minimize the first-order error, they can also help our high-order solver (see Appendix I.2).

For simplicity, denote $\boldsymbol{A}(\lambda_s, \lambda_t) \coloneqq e^{-\int_{\lambda_s}^{\lambda_t} \boldsymbol{l}_\tau \mathrm{d}\tau}$, $\boldsymbol{E}_{\lambda_s}(\lambda) \coloneqq e^{\int_{\lambda_s}^{\lambda} (\boldsymbol{l}_\tau + \boldsymbol{s}_\tau) \mathrm{d}\tau}$, $\boldsymbol{B}_{\lambda_s}(\lambda) \coloneqq \int_{\lambda_s}^{\lambda} e^{-\int_{\lambda_s}^{r} \boldsymbol{s}_\tau \mathrm{d}\tau} \boldsymbol{b}_r \mathrm{d}r$. Though high-order ODE solvers essentially share the same mathematical principles, since we utilize a more complicated parameterization $\boldsymbol{g}_\theta$ and ODE formulation in Eq. (9) than previous works [56, 31, 32, 58], we divide the development of high-order solver into simplified local and global parts, which are not only easier to understand, but also neat and general for any order.

### 3.2.1 Local Approximation

Firstly, we derive formulas and properties of the local approximation, which describes how we transit locally from time $s$ to time $t$. It can be divided into two parts: discretization of the integral in Eq. (9) and estimating the high-order derivatives in the Taylor expansion.

**Discretization.** Denote $\boldsymbol{g}_s^{(k)} \coloneqq \boldsymbol{g}_\theta^{(k)}(\boldsymbol{x}_{\lambda_s}, \lambda_s)$. For $n \geq 0$, to obtain the $(n+1)$-th order discretization of Eq. (9), we take the $n$-th order Taylor expansion of $\boldsymbol{g}_\theta(\boldsymbol{x}_\lambda, \lambda)$ w.r.t. $\lambda$ at $\lambda_s$: $\boldsymbol{g}_\theta(\boldsymbol{x}_\lambda, \lambda) = \sum_{k=0}^{n} \frac{(\lambda - \lambda_s)^k}{k!} \boldsymbol{g}_s^{(k)} + \mathcal{O}((\lambda - \lambda_s)^{n+1})$. Substituting it into Eq. (9) yields

$$\frac{\boldsymbol{x}_t}{\alpha_t} = \boldsymbol{A}(\lambda_s, \lambda_t) \left( \frac{\boldsymbol{x}_s}{\alpha_s} - \int_{\lambda_s}^{\lambda_t} \boldsymbol{E}_{\lambda_s}(\lambda) \boldsymbol{B}_{\lambda_s}(\lambda) \mathrm{d}\lambda - \sum_{k=0}^{n} \boldsymbol{g}_s^{(k)} \int_{\lambda_s}^{\lambda_t} \boldsymbol{E}_{\lambda_s}(\lambda) \frac{(\lambda - \lambda_s)^k}{k!} \mathrm{d}\lambda \right) + \mathcal{O}(h^{n+2}) \tag{12}$$

Here we only need to estimate the $k$-th order total derivatives $\boldsymbol{g}_\theta^{(k)}(\boldsymbol{x}_{\lambda_s}, \lambda_s)$ for $0 \leq k \leq n$, since the other terms are determined once given $\lambda_s, \lambda_t$ and $\boldsymbol{l}_\lambda, \boldsymbol{s}_\lambda, \boldsymbol{b}_\lambda$, which we'll discuss next.

**High-order derivative estimation.** For $(n+1)$-th order approximation, we use the finite difference of $\boldsymbol{g}_\theta(\boldsymbol{x}_\lambda, \lambda)$ at previous $n+1$ steps $\lambda_{i_n}, \ldots, \lambda_{i_1}, \lambda_s$ to estimate each $\boldsymbol{g}_\theta^{(k)}(\boldsymbol{x}_{\lambda_s}, \lambda_s)$. Such derivation is to match the coefficients of Taylor expansions. Concretely, denote $\delta_k \coloneqq \lambda_{i_k} - \lambda_s$, $\boldsymbol{g}_{i_k} \coloneqq \boldsymbol{g}_\theta(\boldsymbol{x}_{\lambda_{i_k}}, \lambda_{i_k})$, and the estimated high-order derivatives $\hat{\boldsymbol{g}}_s^{(k)}$ can be solved by the following linear system:

$$\begin{pmatrix} \delta_1 & \delta_1^2 & \cdots & \delta_1^n \\ \vdots & \vdots & \ddots & \vdots \\ \delta_n & \delta_n^2 & \cdots & \delta_n^n \end{pmatrix} \begin{pmatrix} \hat{\boldsymbol{g}}_s^{(1)} \\ \vdots \\ \frac{\hat{\boldsymbol{g}}_s^{(n)}}{n!} \end{pmatrix} = \begin{pmatrix} \boldsymbol{g}_{i_1} - \boldsymbol{g}_s \\ \vdots \\ \boldsymbol{g}_{i_n} - \boldsymbol{g}_s \end{pmatrix} \tag{13}$$

Then by substituting $\hat{\boldsymbol{g}}_s^{(k)}$ into Eq. (12) and dropping the $\mathcal{O}(h^{n+2})$ error terms, we obtain the $(n+1)$-th order local approximation:

$$\frac{\hat{\boldsymbol{x}}_t}{\alpha_t} = \boldsymbol{A}(\lambda_s, \lambda_t)\left(\frac{\boldsymbol{x}_s}{\alpha_s} - \int_{\lambda_s}^{\lambda_t} \boldsymbol{E}_{\lambda_s}(\lambda)\boldsymbol{B}_{\lambda_s}(\lambda)\mathrm{d}\lambda - \sum_{k=0}^{n} \hat{\boldsymbol{g}}_s^{(k)} \int_{\lambda_s}^{\lambda_t} \boldsymbol{E}_{\lambda_s}(\lambda)\frac{(\lambda - \lambda_s)^k}{k!}\mathrm{d}\lambda\right) \quad (14)$$

where $\hat{\boldsymbol{g}}_\theta^{(0)}(\boldsymbol{x}_{\lambda_s}, \lambda_s) = \boldsymbol{g}_s$. Eq. (13) and Eq. (14) provide an update rule to transit from time $s$ to time $t$ and get an approximated solution $\hat{\boldsymbol{x}}_t$, when we already have the solution $\boldsymbol{x}_s$. For $(n + 1)$-th order approximation, we need $n$ extra solutions $\boldsymbol{x}_{\lambda_{i_k}}$ and their corresponding function values $\boldsymbol{g}_{i_k}$. We illustrate the procedure in Fig. 2(a) and summarize it in Appendix C.2. In the following theorem, we show that under some assumptions, such local approximation has a guarantee of order of accuracy.

**Theorem 3.1** (Local order of accuracy, proof in Appendix B.2.1). *Suppose $\boldsymbol{x}_{\lambda_{i_k}}$ are exact (i.e., on the ground-truth ODE trajectory passing $\boldsymbol{x}_s$) for $k = 1, \ldots, n$, then under some regularity conditions detailed in Appendix B.1, the local truncation error $\hat{\boldsymbol{x}}_t - \boldsymbol{x}_t = \mathcal{O}(h^{n+2})$, which means the local approximation has $(n + 1)$-th order of accuracy.*

Besides, we have the following theorem showing that, whatever the order is, the local approximation is unbiased given our choice of $\boldsymbol{s}_\lambda, \boldsymbol{b}_\lambda$ in Eq. (11). In practice, the phenomenon of reduced bias can be empirically observed (Section 4.3).

**Theorem 3.2** (Local unbiasedness, proof in Appendix B.4). *Given the optimal $\boldsymbol{s}_\lambda^*, \boldsymbol{b}_\lambda^*$ in Eq. (11), For the $(n + 1)$-th order approximation, suppose $\boldsymbol{x}_{\lambda_{i_1}}, \ldots, \boldsymbol{x}_{\lambda_{i_n}}$ are on the ground-truth ODE trajectory passing $\boldsymbol{x}_{\lambda_s}$, then $\mathbb{E}_{p_{\lambda_s}^\theta(\boldsymbol{x}_s)}[\hat{\boldsymbol{x}}_t - \boldsymbol{x}_t] = 0$.*

### 3.2.2 Global Solver

Given $M + 1$ time steps $\{t_i\}_{i=0}^M$, starting from some initial value, we can repeat the local approximation $M$ times to make consecutive transitions from each $t_{i-1}$ to $t_i$ until we reach an acceptable solution. At each step, we apply multistep methods [1] by caching and reusing the previous $n$ values at timesteps $t_{i-1-n}, \ldots, t_{i-2}$, which is proven to be more stable when NFE is small [32, 56]. Moreover, we also apply the predictor-corrector method [58] to refine the approximation at each step without introducing extra NFE. Specifically, the $(n + 1)$-th order predictor is the case of the local approximation when we choose $(t_{i_n}, \ldots, t_{i_1}, s, t) = (t_{i-1-n}, \ldots, t_{i-2}, t_{i-1}, t_i)$, and the $(n + 1)$-th order corrector is the case when we choose $(t_{i_n}, \ldots, t_{i_1}, s, t) = (t_{i-n}, \ldots, t_{i-2}, t_i, t_{i-1}, t_i)$. We present our $(n + 1)$-th order multistep predictor-corrector procedure in Appendix C.2. We also illustrate a second-order case in Fig. 2(b). Note that different from previous works, in the local transition from $t_{i-1}$ to $t_i$, the previous function values $\hat{\boldsymbol{g}}_{i_k}$ ($1 \le k \le n$) used for derivative estimation are dependent on $i$ and are different during the sampling process because $\boldsymbol{g}_\theta$ is dependent on the current $t_{i-1}$ (see Eq. (8)). Thus, we directly cache $\hat{\boldsymbol{x}}_i, \hat{\boldsymbol{\epsilon}}_i$ and reuse them to compute $\hat{\boldsymbol{g}}_i$ in the subsequent steps. Notably, our proposed solver also has a global convergence guarantee, as shown in the following theorem. For simplicity, we only consider the predictor case and the case with corrector can also be proved by similar derivations in [58].

**Theorem 3.3** (Global order of convergence, proof in Appendix B.2.2). *For $(n+1)$-th order predictor, if we iteratively compute a sequence $\{\hat{\boldsymbol{x}}_i\}_{i=0}^M$ to approximate the true solutions $\{\boldsymbol{x}_i\}_{i=0}^M$ at $\{t_i\}_{i=0}^M$, then under both local and global assumptions detailed in Appendix B.1, the final error $|\hat{\boldsymbol{x}}_M - \boldsymbol{x}_M| = \mathcal{O}(h^{n+1})$, where $|\cdot|$ denotes the element-wise absolute value, and $h = \max_{1 \le i \le M}(\lambda_i - \lambda_{i-1})$.*

### 3.3 Practical Techniques

In this section, we introduce some practical techniques that further improve the sample quality in the case of small NFE or large guidance scales.

**Pseudo-order solver for small NFE.** When NFE is extremely small (e.g., 5~10), the error at each timestep becomes rather large, and incorporating too many previous values by high-order solver at each step will cause instabilities. To alleviate this problem, we propose a technique called *pseudo-order* solver: when estimating the $k$-th order derivative, we only utilize the previous $k + 1$ function values of $\boldsymbol{g}_\theta$, instead of all the $n$ previous values as in Eq. (13). For each $k$, we can obtain $\hat{\boldsymbol{g}}_s^{(k)}$ by

solving a part of Eq. (13) and taking the last element:

$$
\begin{pmatrix}
\delta_1 & \delta_1^2 & \cdots & \delta_1^k \\
\vdots & \vdots & \ddots & \vdots \\
\delta_k & \delta_k^2 & \cdots & \delta_k^k
\end{pmatrix}
\begin{pmatrix}
\cdot \\
\vdots \\
\frac{\hat{\boldsymbol{g}}_s^{(k)}}{k!}
\end{pmatrix}
=
\begin{pmatrix}
\boldsymbol{g}_{i_1} - \boldsymbol{g}_s \\
\vdots \\
\boldsymbol{g}_{i_k} - \boldsymbol{g}_s
\end{pmatrix},
\quad k = 1, 2, \ldots, n
\tag{15}
$$

In practice, we do not need to solve $n$ linear systems. Instead, the solutions for $\hat{\boldsymbol{g}}_s^{(k)}, k = 1, \ldots, n$ have a simpler recurrence relation similar to Neville's method [36] in Lagrange polynomial interpolation. Denote $i_0 := s$ so that $\delta_0 = \lambda_{i_0} - \lambda_s = 0$, we have

**Theorem 3.4** (Pseudo-order solver). *For each $k$, the solution in Eq. (15) is $\hat{\boldsymbol{g}}_s^{(k)} = k! D_0^{(k)}$, where*

$$
\begin{aligned}
D_l^{(0)} &:= \boldsymbol{g}_{i_l}, \quad l = 0, 1, \ldots, n \\
D_l^{(k)} &:= \frac{D_{l+1}^{(k-1)} - D_l^{(k-1)}}{\delta_{l+k} - \delta_l}, \quad l = 0, 1, \ldots, n - k
\end{aligned}
\tag{16}
$$

Proof in Appendix B.3. Note that the pseudo-order solver of order $n > 2$ no longer has the guarantee of $n$-th order of accuracy, which is not so important when NFE is small. In our experiments, we mainly rely on two combinations: when we use $n$-th order predictor, we then combine it with $n$-th order corrector or $(n + 1)$-th pseudo-order corrector.

**Half-corrector for large guidance scales.** When the guidance scale is large in guided sampling, we find that corrector may have negative effects on the sample quality. We propose a useful technique called *half-corrector* by using the corrector only in the time region $t \leq 0.5$. Correspondingly, the strategy that we use corrector at each step is called *full-corrector*.

### 3.4 Implementation Details

In this section, we give some implementation details about how to compute and integrate the EMS in our solver and how to adapt them to guided sampling.

**Estimating EMS.** For a specific $\lambda$, the EMS $\boldsymbol{l}_\lambda^*, \boldsymbol{s}_\lambda^*, \boldsymbol{b}_\lambda^*$ can be estimated by firstly drawing $K$ (1024~4096) datapoints $\boldsymbol{x}_\lambda \sim p_\lambda^\theta(\boldsymbol{x}_\lambda)$ with 200-step DPM-Solver++ [32] and then analytically computing some terms related to $\boldsymbol{\epsilon}_\theta$ (detailed in Appendix C.1.1). In practice, we find it both convenient and effective to choose the distribution of the dataset $q_0$ to approximate $p_0^\theta$. Thus, without further specifications, we directly use samples from $q_0$.

**Estimating integrals of EMS.** We estimate EMS on $N$ (120 ∼ 1200) timesteps $\lambda_{j_0}, \lambda_{j_1}, \ldots, \lambda_{j_N}$ and use trapezoidal rule to estimate the integrals in Eq. (12) (see Appendix I.3 for the estimation error analysis). We also apply some pre-computation for the integrals to avoid extra computation costs during sampling, detailed in Appendix C.1.2.

**Adaptation to guided sampling.** Empirically, we find that within a common range of guidance scales, we can simply compute the EMS on the model without guidance, and it can work for both unconditional sampling and guided sampling cases. See Appendix J for more discussions.

### 3.5 Comparison with Existing Methods

By comparing with existing diffusion ODE solvers that are based on exponential integrators [56, 31, 32, 58], we can conclude that (1) Previous ODE formulations with noise/data prediction are special cases of ours by setting $\boldsymbol{l}_\lambda, \boldsymbol{s}_\lambda, \boldsymbol{b}_\lambda$ to specific values. (2) Our first-order discretization can be seen as improved DDIM. See more details in Appendix A.

## 4 Experiments

In this section, we show that DPM-Solver-v3 can achieve consistent and notable speed-up for both unconditional and conditional sampling with both pixel-space and latent-space DPMs. We conduct extensive experiments on diverse image datasets, where the resolution ranges from 32 to 256. First, we present the main results of sample quality comparison with previous state-of-the-art training-free

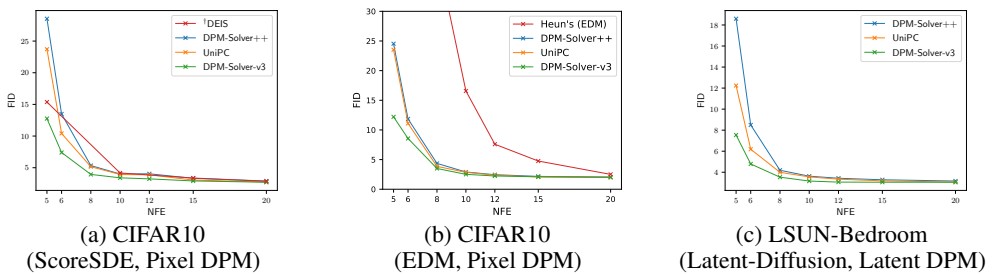

(a) CIFAR10
(ScoreSDE, Pixel DPM)

(b) CIFAR10
(EDM, Pixel DPM)

(c) LSUN-Bedroom
(Latent-Diffusion, Latent DPM)

Figure 3: Unconditional sampling results. We report the FID↓ of the methods with different numbers of function evaluations (NFE), evaluated on 50k samples. †We borrow the results reported in their original paper directly.

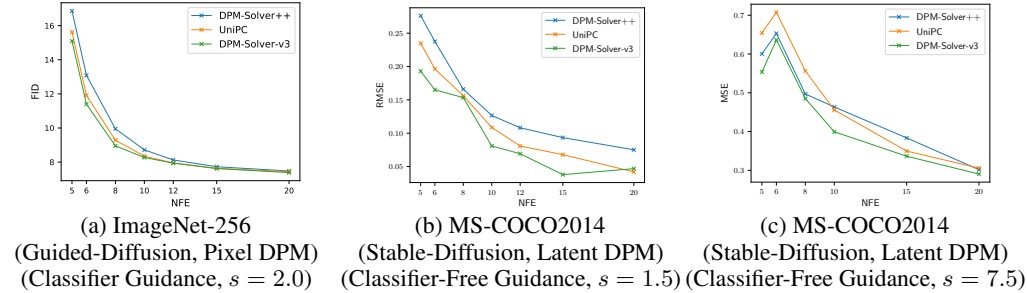

(a) ImageNet-256
(Guided-Diffusion, Pixel DPM)
(Classifier Guidance, $s = 2.0$)

(b) MS-COCO2014
(Stable-Diffusion, Latent DPM)
(Classifier-Free Guidance, $s = 1.5$)

(c) MS-COCO2014
(Stable-Diffusion, Latent DPM)
(Classifier-Free Guidance, $s = 7.5$)

Figure 4: Conditional sampling results. We report the FID↓ or MSE↓ of the methods with different numbers of function evaluations (NFE), evaluated on 10k samples.

methods. Then we illustrate the effectiveness of our method by visualizing the EMS and samples. Additional ablation studies are provided in Appendix G. On each dataset, we choose a sufficient number of timesteps $N$ and datapoints $K$ for computing the EMS to reduce the estimation error, while the EMS can still be computed within hours. After we obtain the EMS and precompute the integrals involving them, there is **negligible extra overhead** in the sampling process. We provide the runtime comparison in Appendix E. We refer to Appendix D for more detailed experiment settings.

## 4.1 Main Results

We present the results in $5 \sim 20$ number of function evaluations (NFE), covering both few-step cases and the almost converged cases, as shown in Fig. 3 and Fig. 4. For the sake of clarity, we mainly compare DPM-Solver-v3 to DPM-Solver++ [32] and UniPC [58], which are the most state-of-the-art diffusion ODE solvers. We also include the results for DEIS [56] and Heun's 2nd order method in EDM [21], but only for the datasets on which they originally reported. We don't show the results for other methods such as DDIM [48], PNDM [28], since they have already been compared in previous works and have inferior performance. The quantitative results on CIFAR10 [24] are listed in Table 1, and more detailed quantitative results are presented in Appendix F.

**Unconditional sampling** We first evaluate the unconditional sampling performance of different methods on CIFAR10 [24] and LSUN-Bedroom [55]. For CIFAR10 we use two pixel-space DPMs,

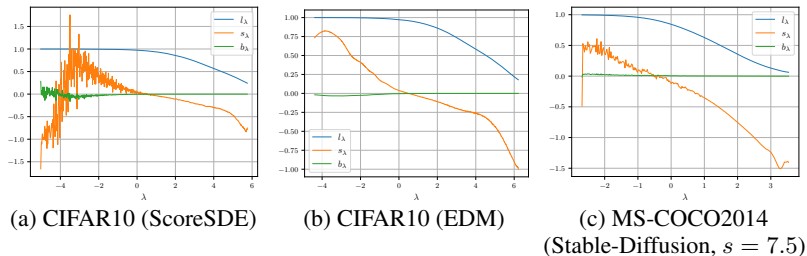

(a) CIFAR10 (ScoreSDE)

(b) CIFAR10 (EDM)

(c) MS-COCO2014
(Stable-Diffusion, $s = 7.5$)

Figure 5: Visualization of the EMS $\boldsymbol{l}_\lambda, \boldsymbol{s}_\lambda, \boldsymbol{b}_\lambda$ w.r.t. $\lambda$ estimated on different models.

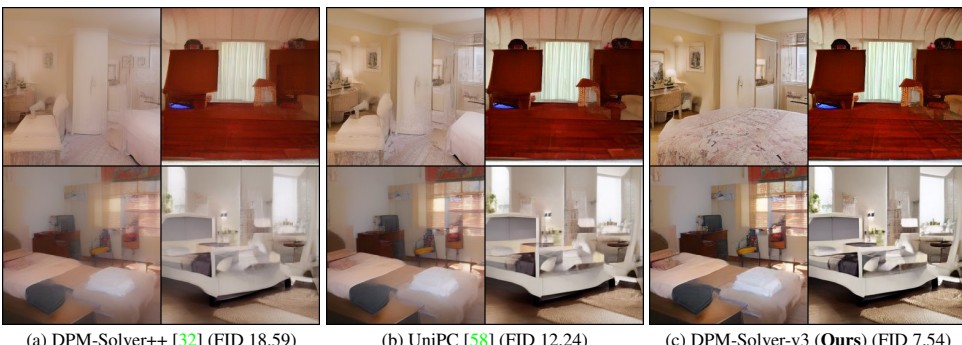

(a) DPM-Solver++ [32] (FID 18.59)    (b) UniC [58] (FID 12.24)    (c) DPM-Solver-v3 (**Ours**) (FID 7.54)

Figure 6: Random samples of Latent-Diffusion [43] on LSUN-Bedroom [55] with only NFE = 5.

Table 1: Quantitative results on CIFAR10 [24]. We report the FID↓ of the methods with different numbers of function evaluations (NFE), evaluated on 50k samples. †We borrow the results reported in their original paper directly.

| Method | Model | NFE | | | | | | | |
|---|---|---|---|---|---|---|---|---|---|
| | | 5 | 6 | 8 | 10 | 12 | 15 | 20 | 25 |
| †DEIS [56] | ScoreSDE [51] | 15.37 | \ | \ | 4.17 | \ | 3.37 | 2.86 | \ |
| DPM-Solver++ [32] | | 28.53 | 13.48 | 5.34 | 4.01 | 4.04 | 3.32 | 2.90 | 2.76 |
| UniPC [58] | | 23.71 | 10.41 | 5.16 | 3.93 | 3.88 | 3.05 | 2.73 | 2.65 |
| DPM-Solver-v3 | | **12.76** | **7.40** | **3.94** | **3.40** | **3.24** | **2.91** | **2.71** | **2.64** |
| Heun's 2nd [21] | EDM [21] | 320.80 | 103.86 | 39.66 | 16.57 | 7.59 | 4.76 | 2.51 | 2.12 |
| DPM-Solver++ [32] | | 24.54 | 11.85 | 4.36 | 2.91 | 2.45 | 2.17 | 2.05 | 2.02 |
| UniPC [58] | | 23.52 | 11.10 | 3.86 | 2.85 | 2.38 | **2.08** | **2.01** | **2.00** |
| DPM-Solver-v3 | | **12.21** | **8.56** | **3.50** | **2.51** | **2.24** | 2.10 | 2.02 | **2.00** |

one is based on ScoreSDE [51] which is a widely adopted model by previous samplers, and another is based on EDM [21] which achieves the best sample quality. For LSUN-Bedroom, we use the latent-space Latent-Diffusion model [43]. We apply the multistep 3rd-order version for DPM-Solver++, UniPC and DPM-Solver-v3 by default, which performs best in the unconditional setting. For UniPC, we report the better result of their two choices $B_1(h) = h$ and $B_2(h) = e^h - 1$ at each NFE. For our DPM-Solver-v3, we tune the strategies of whether to use the pseudo-order predictor/corrector at each NFE on CIFAR10, and use the pseudo-order corrector on LSUN-Bedroom. As shown in Fig. 3, we find that DPM-Solver-v3 can achieve consistently better FID, which is especially notable when NFE is 5∼10. For example, we improve the FID on CIFAR10 with 5 NFE from 23 to 12 with ScoreSDE, and achieve an FID of 2.51 with only 10 NFE with the advanced DPM provided by EDM. On LSUN-Bedroom, with around 12 minutes computing of the EMS, DPM-Solver-v3 converges to the FID of 3.06 with 12 NFE, which is approximately **60% sampling cost** of the previous best training-free method (20 NFE by UniPC).

**Conditional sampling.** We then evaluate the conditional sampling performance, which is more widely used since it allows for controllable generation with user-customized conditions. We choose two conditional settings, one is classifier guidance on pixel-space Guided-Diffusion [10] model trained on ImageNet-256 dataset [9] with 1000 class labels as conditions; the other is classifier-free guidance on latent-space Stable-Diffusion model [43] trained on LAION-5B dataset [46] with CLIP [41] embedded text prompts as conditions. We evaluate the former at the guidance scale of 2.0, following the best choice in [10]; and the latter at the guidance scale of 1.5 (following the original paper) or 7.5 (following the official code) with prompts random selected from MS-COCO2014 validation set [26]. Note that the FID of Stable-Diffusion samples saturates to 15.0∼16.0 even within 10 steps when the latent codes are far from convergence, possibly due to the powerful image decoder (see Appendix H). Thus, following [32], we measure the mean square error (MSE) between the generated latent code $\hat{x}$ and the ground-truth solution $x^*$ (i.e., $\|\hat{x} - x^*\|_2^2/D$) to evaluate convergence, starting from the same Gaussian noise. We obtain $x^*$ by 200-step DPM-Solver++, which is enough to ensure the convergence.

We apply the multistep 2nd-order version for DPM-Solver++, UniPC and DPM-Solver-v3, which performs best in conditional setting. For UniPC, we only apply the choice $B_2(h) = e^h - 1$, which performs better than $B_1(h)$. For our DPM-Solver-v3, we use the pseudo-order corrector by default, and report the best results between using half-corrector/full-corrector on Stable-Diffusion ($s = 7.5$). As shown in Fig. 4, DPM-Solver-v3 can achieve better sample quality or convergence at most NFEs, which indicates the effectiveness of our method and techniques under the conditional setting. It's worth noting that UniPC, which adopts an extra corrector, performs even worse than DPM-Solver++ when NFE<10 on Stable-Diffusion ($s = 7.5$). With the combined effect of the EMS and the half-corrector technique, we successfully outperform DPM-Solver++ in such a case. Detailed comparisons can be found in the ablations in Appendix G.

### 4.2 Visualizations of Estimated EMS

We further visualize the estimated EMS in Fig. 5. Since $l_\lambda, s_\lambda, b_\lambda$ are $D$-dimensional vectors, we average them over all dimensions to obtain a scalar. From Fig. 5, we find that $l_\lambda$ gradually changes from 1 to near 0 as the sampling proceeds, which suggests we should gradually slide from data prediction to noise prediction. As for $s_\lambda, b_\lambda$, they are more model-specific and display many fluctuations, especially for ScoreSDE model [51] on CIFAR10. Apart from the estimation error of the EMS, we suspect that it comes from the fluctuations of $\epsilon_\theta^{(1)}$, which is caused by the periodicity of trigonometric functions in the positional embeddings of the network input. It's worth noting that the fluctuation of $s_\lambda, b_\lambda$ will not cause instability in our sampler (see Appendix J).

### 4.3 Visual Quality

We present some qualitative comparisons in Fig. 6 and Fig. 1. We can find that previous methods tend to have a small degree of color contrast at small NFE, while our method is less biased and produces more visual details. In Fig. 1(b), we can observe that previous methods with corrector may cause distortion at large guidance scales (in the left-top image, a part of the castle becomes a hill; in the left-bottom image, the hill is translucent and the castle is hanging in the air), while ours won't. Additional samples are provided in Appendix K.

## 5 Conclusion

We study the ODE parameterization problem for fast sampling of DPMs. Through theoretical analysis, we find a novel ODE formulation with empirical model statistics, which is towards the optimal one to minimize the first-order discretization error. Based on such improved ODE formulation, we propose a new fast solver named DPM-Solver-v3, which involves a multistep predictor-corrector framework of any order and several techniques for improved sampling with few steps or large guidance scale. Experiments demonstrate the effectiveness of DPM-Solver-v3 in both unconditional conditional sampling with both pixel-space latent-space pre-trained DPMs, and the significant advancement of sample quality in 5~10 steps.

**Limitations and broader impact** Despite the great speedup in small numbers of steps, DPM-Solver-v3 still lags behind training-based methods and is not fast enough for real-time applications. Besides, we conducted theoretical analyses of the local error, but didn't explore the global design spaces, such as the design of timestep schedules during sampling. And commonly, there are potential undesirable effects that DPM-Solver-v3 may be abused to accelerate the generation of fake and malicious content.

## Acknowledgements

This work was supported by the National Key Research and Development Program of China (No. 2021ZD0110502), NSFC Projects (Nos. 62061136001, 62106123, 62076147, U19A2081, 61972224, 62106120), Tsinghua Institute for Guo Qiang, and the High Performance Computing Center, Tsinghua University. J.Z is also supported by the XPlorer Prize.

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

# A  Related Work

Diffusion probabilistic models (DPMs) [47, 15, 51], also known as score-based generative models (SGMs), have achieved remarkable generation ability on image domain [10, 21], yielding extensive applications such as speech, singing and video synthesis [6, 27, 14], controllable image generation, translation and editing [38, 42, 43, 35, 57, 8], likelihood estimation [50, 23, 30, 59], data compression [23] and inverse problem solving [7, 22].

## A.1  Fast Sampling Methods for DPMs

Fast sampling methods based on extra training or optimization include learning variances in the reverse process [37, 2], learning sampling schedule [53], learning high-order model derivatives [11], model refinement [29] and model distillation [45, 33, 49]. Though distillation-based methods can generate high-quality samples in less than 5 steps, they additionally bring onerous training costs. Moreover, the distillation process will inevitably lose part of the information of the original model, and is hard to be adapted to pre-trained large DPMs [43, 44, 42] and conditional sampling [34]. Some of distillation-based methods also lack the ability to make flexible trade-offs between sample speed and sample quality.

In contrast, training-free samplers are more lightweight and flexible. Among them, samplers based on diffusion ODEs generally require fewer steps than those based on diffusion SDEs [51, 40, 3, 48, 32], since SDEs introduce more randomness and make the denoising harder in the sampling process. Previous samplers handle the diffusion ODEs with different methods, such as Heun's methods [21], splitting numerical methods [54], pseudo numerical methods [28], Adams methods [25] or exponential integrators [56, 31, 32, 58].

## A.2  Comparison with Existing Solvers Based on Exponential Integrators

Table 2: Comparison between DPM-Solver-v3 and other high-order diffusion ODE solvers based on exponential integrators.

|  | DEIS [56] | DPM-Solver [31] | DPM-Solver++ [32] | UniPC [58] | DPM-Solver-v3 (**Ours**) |
|---|---|---|---|---|---|
| First-Order | DDIM | DDIM | DDIM | DDIM | Improved DDIM |
| Taylor Expanded Predictor | $\epsilon_\theta$ for $t$ | $\epsilon_\theta$ for $\lambda$ | $x_\theta$ for $\lambda$ | $x_\theta$ for $\lambda$ | $g_\theta$ for $\lambda$ |
| Solver Type (High-Order) | Multistep | Singlestep | Multistep | Multistep | Multistep |
| Applicable for Guided Sampling | ✓ | ✗ | ✓ | ✓ | ✓ |
| Corrector Supported | ✗ | ✗ | ✗ | ✓ | ✓ |
| Model-Specific | ✗ | ✗ | ✗ | ✗ | ✓ |

In this section, we make some theoretical comparisons between DPM-Solver-v3 and existing diffusion ODE solvers that are based on exponential integrators [56, 31, 32, 58]. We summarize the results in Table 2, and provide some analysis below.

**Previous ODE formulation as special cases of ours.**  First, we compare the formulation of the ODE solution. Our reformulated ODE solution in Eq. (9) involves extra coefficients $l_\lambda, s_\lambda, b_\lambda$, and it corresponds to a new predictor $g_\theta$ to be approximated by Taylor expansion. By comparing ours with previous ODE formulations in Eq. (3) and their corresponding noise/data prediction, we can easily figure out that they are special cases of ours by setting $l_\lambda, s_\lambda, b_\lambda$ to specific values:

- Noise prediction: $l_\lambda = 0, s_\lambda = -1, b_\lambda = 0$

- Data prediction: $l_\lambda = 1, s_\lambda = 0, b_\lambda = 0$

It's worth noting that, though our ODE formulation can degenerate to previous ones, it may still result in different solvers, since previous works conduct some equivalent substitution of the same order in the local approximation (for example, the choice of $B_1(h) = h$ or $B_2(h) = e^h - 1$ in UniPC [58]). We never conduct such substitution, thus saving the efforts to tune it.

Moreover, under our framework, we find that **DPM-Solver++ is a model-agnostic approximation of DPM-Solver-v3, under the Gaussian assumption**. Specifically, according to Eq. (5), we have

$$\boldsymbol{l}_\lambda^* = \underset{\boldsymbol{l}_\lambda}{\arg\min}\, \mathbb{E}_{p_\lambda^\theta(\boldsymbol{x}_\lambda)} \|\sigma_\lambda \nabla_{\boldsymbol{x}} \boldsymbol{\epsilon}_\theta(\boldsymbol{x}_\lambda, \lambda) - \boldsymbol{l}_\lambda\|_F^2, \tag{17}$$

If we assume $q_\lambda(\boldsymbol{x}_\lambda) \approx \mathcal{N}(\boldsymbol{x}_\lambda | \alpha_\lambda \boldsymbol{x}_0, \sigma_\lambda^2 \boldsymbol{I})$ for some fixed $\boldsymbol{x}_0$, then the optimal noise predictor is

$$\boldsymbol{\epsilon}_\theta(\boldsymbol{x}_\lambda, \lambda) \approx -\sigma_\lambda \nabla_{\boldsymbol{x}} \log q_\lambda(\boldsymbol{x}_\lambda) = \frac{\boldsymbol{x}_\lambda - \alpha_t \boldsymbol{x}_0}{\sigma_\lambda}. \tag{18}$$

It follows that $\sigma_\lambda \nabla_{\boldsymbol{x}} \boldsymbol{\epsilon}_\theta(\boldsymbol{x}_\lambda, \lambda) \approx \boldsymbol{I}$, thus $\boldsymbol{l}_\lambda^* \approx \boldsymbol{1}$ by Eq. (5), which corresponds the data prediction model used in DPM-Solver++. Moreover, for small enough $\lambda$ (i.e., $t$ near to $T$), the Gaussian assumption is almost true (see Section 4.2), thus the data-prediction DPM-Solver++ approximately computes all the linear terms at the initial stage. To the best of our knowledge, this is the first explanation for the reason why the data-prediction DPM-Solver++ outperforms the noise-prediction DPM-Solver.

**First-order discretization as improved DDIM**  Previous methods merely use noise/data parameterization, whether or not they change the time domain from $t$ to $\lambda$. While they differ in high-order cases, they are proven to coincide in the first-order case, which is DDIM [48] (deterministic case, $\eta = 0$):

$$\hat{\boldsymbol{x}}_t = \frac{\alpha_t}{\alpha_s} \hat{\boldsymbol{x}}_s - \alpha_t \left( \frac{\sigma_s}{\alpha_s} - \frac{\sigma_t}{\alpha_t} \right) \boldsymbol{\epsilon}_\theta(\hat{\boldsymbol{x}}_s, \lambda_s) \tag{19}$$

However, the first-order case of our method is

$$\begin{aligned}
\hat{\boldsymbol{x}}_t = \frac{\alpha_t}{\alpha_s} \boldsymbol{A}(\lambda_s, \lambda_t) \left( \left( 1 + \boldsymbol{l}_{\lambda_s} \int_{\lambda_s}^{\lambda_t} \boldsymbol{E}_{\lambda_s}(\lambda) \mathrm{d}\lambda \right) \hat{\boldsymbol{x}}_s - \left( \sigma_s \int_{\lambda_s}^{\lambda_t} \boldsymbol{E}_{\lambda_s}(\lambda) \mathrm{d}\lambda \right) \boldsymbol{\epsilon}_\theta(\hat{\boldsymbol{x}}_s, \lambda_s) \right) \\
- \alpha_t \boldsymbol{A}(\lambda_s, \lambda_t) \int_{\lambda_s}^{\lambda_t} \boldsymbol{E}_{\lambda_s}(\lambda) \boldsymbol{B}_{\lambda_s}(\lambda) \mathrm{d}\lambda
\end{aligned} \tag{20}$$

which is not DDIM since we choose a better parameterization by the estimated EMS. Empirically, our first-order solver performs better than DDIM, as detailed in Appendix G.

# B  Proofs

## B.1  Assumptions

In this section, we will give some mild conditions under which the local order of accuracy of Algorithm 1 and the global order of convergence of Algorithm 2 (predictor) are guaranteed.

### B.1.1  Local

First, we will give the assumptions to bound the local truncation error.

**Assumption B.1.** The total derivatives of the noise prediction model $\frac{\mathrm{d}^k \boldsymbol{\epsilon}_\theta(\boldsymbol{x}_\lambda, \lambda)}{\mathrm{d}\lambda^k}, k = 1, \dots, n$ exist and are continuous.

**Assumption B.2.** The coefficients $\boldsymbol{l}_\lambda, \boldsymbol{s}_\lambda, \boldsymbol{b}_\lambda$ are continuous and bounded. $\frac{\mathrm{d}^k \boldsymbol{l}_\lambda}{\mathrm{d}\lambda^k}, \frac{\mathrm{d}^k \boldsymbol{s}_\lambda}{\mathrm{d}\lambda^k}, \frac{\mathrm{d}^k \boldsymbol{b}_\lambda}{\mathrm{d}\lambda^k}, k = 1, \dots, n$ exist and are continuous.

**Assumption B.3.** $\delta_k = \Theta(\lambda_t - \lambda_s), k = 1, \dots, n$

Assumption B.1 is required for the Taylor expansion which is regular in high-order numerical methods. Assumption B.2 requires the boundness of the coefficients as well as regularizes the coefficients' smoothness to enable the Taylor expansion for $\boldsymbol{g}_\theta(\boldsymbol{x}_\lambda, \lambda)$, which holds in practice given the smoothness of $\boldsymbol{\epsilon}_\theta(\boldsymbol{x}_\lambda, \lambda)$ and $p_\lambda^\theta(\boldsymbol{x}_\lambda)$. Assumption B.3 makes sure $\delta_k$ and $\lambda_t - \lambda_s$ is of the same order, i.e., there exists some constant $r_k = \mathcal{O}(1)$ so that $\delta_k = r_k(\lambda_t - \lambda_s)$, which is satisfied in regular multistep methods.

### B.1.2 Global

Then we will give the assumptions to bound the global error.

**Assumption B.4.** The noise prediction model $\epsilon_\theta(\boldsymbol{x}, t)$ is Lipschitz w.r.t. to $\boldsymbol{x}$.

**Assumption B.5.** $h = \max_{1 \le i \le M}(\lambda_i - \lambda_{i-1}) = \mathcal{O}(1/M)$.

**Assumption B.6.** The starting values $\hat{\boldsymbol{x}}_i, 1 \le i \le n$ satisfies $\hat{\boldsymbol{x}}_i - \boldsymbol{x}_i = \mathcal{O}(h^{n+1})$.

Assumption B.4 is common in the analysis of ODEs, which assures $\epsilon_\theta(\hat{\boldsymbol{x}}_t, t) - \epsilon_\theta(\boldsymbol{x}_t, t) = \mathcal{O}(\hat{\boldsymbol{x}}_t - \boldsymbol{x}_t)$. Assumption B.5 implies that the step sizes are rather uniform. Assumption B.6 is common in the convergence analysis of multistep methods [5].

## B.2 Order of Accuracy and Convergence

In this section, we prove the local and global order guarantees detailed in Theorem 3.1 and Theorem 3.3.

### B.2.1 Local

*Proof.* (Proof of Theorem 3.1) Denote $h := \lambda_t - \lambda_s$. Subtracting the Taylor-expanded exact solution in Eq. (12) from the local approximation in Eq. (14), we have

$$\hat{\boldsymbol{x}}_t - \boldsymbol{x}_t = -\alpha_t \boldsymbol{A}(\lambda_s, \lambda_t) \sum_{k=1}^{n} \frac{\hat{\boldsymbol{g}}_\theta^{(k)}(\boldsymbol{x}_{\lambda_s}, \lambda_s) - \boldsymbol{g}_\theta^{(k)}(\boldsymbol{x}_{\lambda_s}, \lambda_s)}{k!} \int_{\lambda_s}^{\lambda_t} \boldsymbol{E}_{\lambda_s}(\lambda)(\lambda - \lambda_s)^k \mathrm{d}\lambda + \mathcal{O}(h^{n+2}) \tag{21}$$

First we examine the order of $\boldsymbol{A}(\lambda_s, \lambda_t)$ and $\int_{\lambda_s}^{\lambda_t} \boldsymbol{E}_{\lambda_s}(\lambda)(\lambda - \lambda_s)^k \mathrm{d}\lambda$. Under Assumption B.2, there exists some constant $C_1, C_2$ such that $-\boldsymbol{l}_\lambda < C_1, \boldsymbol{l}_\lambda + \boldsymbol{s}_\lambda < C_2$. So

$$\begin{aligned} \boldsymbol{A}(\lambda_s, \lambda_t) &= e^{-\int_{\lambda_s}^{\lambda_t} \boldsymbol{l}_\tau \mathrm{d}\tau} \\ &< e^{C_1 h} \\ &= \mathcal{O}(1) \end{aligned} \tag{22}$$

$$\begin{aligned} \int_{\lambda_s}^{\lambda_t} \boldsymbol{E}_{\lambda_s}(\lambda)(\lambda - \lambda_s)^k \mathrm{d}\lambda &= \int_{\lambda_s}^{\lambda_t} e^{\int_{\lambda_s}^{\lambda}(\boldsymbol{l}_\tau + \boldsymbol{s}_\tau)\mathrm{d}\tau}(\lambda - \lambda_s)^k \mathrm{d}\lambda \\ &< \int_{\lambda_s}^{\lambda_t} e^{C_2(\lambda - \lambda_s)}(\lambda - \lambda_s)^k \mathrm{d}\lambda \\ &= \mathcal{O}(h^{k+1}) \end{aligned} \tag{23}$$

Next we examine the order of $\frac{\hat{\boldsymbol{g}}_\theta^{(k)}(\boldsymbol{x}_{\lambda_s}, \lambda_s) - \boldsymbol{g}_\theta^{(k)}(\boldsymbol{x}_{\lambda_s}, \lambda_s)}{k!}$. Under Assumption B.1 and Assumption B.2, since $\boldsymbol{g}_\theta$ is elementary function of $\epsilon_\theta$ and $\boldsymbol{l}_\lambda, \boldsymbol{s}_\lambda, \boldsymbol{b}_\lambda$, we know $\boldsymbol{g}_\theta^{(k)}(\boldsymbol{x}_{\lambda_s}, \lambda_s), k = 1, \ldots, n$ exist and are continuous. Adopting the notations in Eq. (13), by Taylor expansion, we have

$$\begin{aligned} \boldsymbol{g}_{i_1} &= \boldsymbol{g}_s + \delta_1 \boldsymbol{g}_s^{(1)} + \delta_1^2 \boldsymbol{g}_s^{(2)} + \cdots + \delta_1^n \boldsymbol{g}_s^{(n)} + \mathcal{O}(\delta_1^{n+1}) \\ \boldsymbol{g}_{i_2} &= \boldsymbol{g}_s + \delta_2 \boldsymbol{g}_s^{(1)} + \delta_2^2 \boldsymbol{g}_s^{(2)} + \cdots + \delta_2^n \boldsymbol{g}_s^{(n)} + \mathcal{O}(\delta_2^{n+1}) \\ &\cdots \\ \boldsymbol{g}_{i_n} &= \boldsymbol{g}_s + \delta_n \boldsymbol{g}_s^{(1)} + \delta_n^2 \boldsymbol{g}_s^{(2)} + \cdots + \delta_n^n \boldsymbol{g}_s^{(n)} + \mathcal{O}(\delta_n^{n+1}) \end{aligned} \tag{24}$$

Comparing it with Eq. (13), we have

$$\begin{pmatrix} \delta_1 & \delta_1^2 & \cdots & \delta_1^n \\ \delta_2 & \delta_2^2 & \cdots & \delta_2^n \\ \vdots & \vdots & \ddots & \vdots \\ \delta_n & \delta_n^2 & \cdots & \delta_n^n \end{pmatrix} \begin{pmatrix} \hat{\boldsymbol{g}}_s^{(1)} - \boldsymbol{g}_s^{(1)} \\ \frac{\hat{\boldsymbol{g}}_s^{(2)} - \boldsymbol{g}_s^{(2)}}{2!} \\ \vdots \\ \frac{\hat{\boldsymbol{g}}_s^{(n)} - \boldsymbol{g}_s^{(n)}}{n!} \end{pmatrix} = \begin{pmatrix} \mathcal{O}(\delta_1^{n+1}) \\ \mathcal{O}(\delta_2^{n+1}) \\ \vdots \\ \mathcal{O}(\delta_n^{n+1}) \end{pmatrix} \tag{25}$$

From Assumption B.3, we know there exists some constants $r_k$ so that $\delta_k = r_k h, k = 1, \ldots, n$. Thus

$$
\begin{pmatrix}
\delta_1 & \delta_1^2 & \cdots & \delta_1^n \\
\delta_2 & \delta_2^2 & \cdots & \delta_2^n \\
\vdots & \vdots & \ddots & \vdots \\
\delta_n & \delta_n^2 & \cdots & \delta_n^n
\end{pmatrix}
=
\begin{pmatrix}
r_1 & r_1^2 & \cdots & r_1^n \\
r_2 & r_2^2 & \cdots & r_2^n \\
\vdots & \vdots & \ddots & \vdots \\
r_n & r_n^2 & \cdots & r_n^n
\end{pmatrix}
\begin{pmatrix}
h & & & \\
& h^2 & & \\
& & \ddots & \\
& & & h^n
\end{pmatrix},
\begin{pmatrix}
\mathcal{O}(\delta_1^{n+1}) \\
\mathcal{O}(\delta_2^{n+1}) \\
\vdots \\
\mathcal{O}(\delta_n^{n+1})
\end{pmatrix}
=
\begin{pmatrix}
\mathcal{O}(h^{n+1}) \\
\mathcal{O}(h^{n+1}) \\
\vdots \\
\mathcal{O}(h^{n+1})
\end{pmatrix}
\tag{26}
$$

And finally, we have

$$
\begin{pmatrix}
\hat{\boldsymbol{g}}_s^{(1)} - \boldsymbol{g}_s^{(1)} \\
\frac{\hat{\boldsymbol{g}}_s^{(2)} - \boldsymbol{g}_s^{(2)}}{2!} \\
\vdots \\
\frac{\hat{\boldsymbol{g}}_s^{(n)} - \boldsymbol{g}_s^{(n)}}{n!}
\end{pmatrix}
=
\begin{pmatrix}
h^{-1} & & & \\
& h^{-2} & & \\
& & \ddots & \\
& & & h^{-n}
\end{pmatrix}
\begin{pmatrix}
r_1 & r_1^2 & \cdots & r_1^n \\
r_2 & r_2^2 & \cdots & r_2^n \\
\vdots & \vdots & \ddots & \vdots \\
r_n & r_n^2 & \cdots & r_n^n
\end{pmatrix}^{-1}
\begin{pmatrix}
\mathcal{O}(h^{n+1}) \\
\mathcal{O}(h^{n+1}) \\
\vdots \\
\mathcal{O}(h^{n+1})
\end{pmatrix}
$$

$$
=
\begin{pmatrix}
\mathcal{O}(h^n) \\
\mathcal{O}(h^{n-1}) \\
\vdots \\
\mathcal{O}(h^1)
\end{pmatrix}
\tag{27}
$$

Substitute Eq. (22), Eq. (23) and Eq. (27) into Eq. (21), we can conclude that $\hat{\boldsymbol{x}}_t - \boldsymbol{x}_t = \mathcal{O}(h^{n+2})$. $\quad\square$

### B.2.2    Global

First, we provide a lemma that gives the local truncation error given inexact previous values when estimating the high-order derivatives.

**Lemma B.7.** *(Local truncation error with inexact previous values) Suppose inexact values $\hat{\boldsymbol{x}}_{\lambda_{i_k}}, k = 1, \ldots, n$ and $\hat{\boldsymbol{x}}_s$ are used in Eq. (13) to estimate the high-order derivatives, then the local truncation error of the local approximation Eq. (14) satisfies*

$$
\Delta_t = \frac{\alpha_t \boldsymbol{A}(\lambda_s, \lambda_t)}{\alpha_s} \Delta_s + \mathcal{O}(h) \left( \mathcal{O}(\Delta_s) + \sum_{k=1}^{n} \mathcal{O}(\Delta_{\lambda_{i_k}}) + \mathcal{O}(h^{n+1}) \right)
\tag{28}
$$

*where $\Delta_\cdot := \hat{\boldsymbol{x}}_\cdot - \boldsymbol{x}_\cdot, h := \lambda_t - \lambda_s$.*

*Proof.* By replacing $\boldsymbol{x}_\cdot$ with $\hat{\boldsymbol{x}}_\cdot$ in Eq. (13) and subtracting Eq. (12) from Eq. (14), the expression for the local truncation error becomes

$$
\Delta_t = \frac{\alpha_t \boldsymbol{A}(\lambda_s, \lambda_t)}{\alpha_s} \Delta_s - \alpha_t \boldsymbol{A}(\lambda_s, \lambda_t) \left( \boldsymbol{g}_\theta(\hat{\boldsymbol{x}}_{\lambda_s}, \lambda_s) - \boldsymbol{g}_\theta(\boldsymbol{x}_{\lambda_s}, \lambda_s) \right) \int_{\lambda_s}^{\lambda_t} \boldsymbol{E}_{\lambda_s}(\lambda) \mathrm{d}\lambda
$$

$$
- \alpha_t \boldsymbol{A}(\lambda_s, \lambda_t) \sum_{k=1}^{n} \frac{\hat{\boldsymbol{g}}_\theta^{(k)}(\boldsymbol{x}_{\lambda_s}, \lambda_s) - \boldsymbol{g}_\theta^{(k)}(\boldsymbol{x}_{\lambda_s}, \lambda_s)}{k!} \int_{\lambda_s}^{\lambda_t} \boldsymbol{E}_{\lambda_s}(\lambda)(\lambda - \lambda_s)^k \mathrm{d}\lambda + \mathcal{O}(h^{n+2})
\tag{29}
$$

And the linear system for solving $\boldsymbol{g}_\theta^{(k)}(\boldsymbol{x}_{\lambda_s}, \lambda_s), k = 1, \ldots, n$ becomes

$$
\begin{pmatrix}
\delta_1 & \delta_1^2 & \cdots & \delta_1^n \\
\delta_2 & \delta_2^2 & \cdots & \delta_2^n \\
\vdots & \vdots & \ddots & \vdots \\
\delta_n & \delta_n^2 & \cdots & \delta_n^n
\end{pmatrix}
\begin{pmatrix}
\hat{\boldsymbol{g}}_s^{(1)} \\
\frac{\hat{\boldsymbol{g}}_s^{(2)}}{2!} \\
\vdots \\
\frac{\hat{\boldsymbol{g}}_s^{(n)}}{n!}
\end{pmatrix}
=
\begin{pmatrix}
\hat{\boldsymbol{g}}_{i_1} - \hat{\boldsymbol{g}}_s \\
\hat{\boldsymbol{g}}_{i_2} - \hat{\boldsymbol{g}}_s \\
\vdots \\
\hat{\boldsymbol{g}}_{i_n} - \hat{\boldsymbol{g}}_s
\end{pmatrix}
\tag{30}
$$

where $\hat{\boldsymbol{g}}_\cdot = \boldsymbol{g}_\theta(\hat{\boldsymbol{x}}_{\lambda_\cdot}, \lambda_\cdot)$. Under Assumption B.4, we know $\hat{\boldsymbol{g}}_\cdot - \boldsymbol{g}_\cdot = \mathcal{O}(\Delta_{\lambda_\cdot})$. Thus, under Assumption B.1, Assumption B.2 and Assumption B.3, similar to the deduction in the last section, we have

$$
\begin{pmatrix}
\hat{\boldsymbol{g}}_s^{(1)} - \boldsymbol{g}_s^{(1)} \\
\frac{\hat{\boldsymbol{g}}_s^{(2)} - \boldsymbol{g}_s^{(2)}}{2!} \\
\vdots \\
\frac{\hat{\boldsymbol{g}}_s^{(n)} - \boldsymbol{g}_s^{(n)}}{n!}
\end{pmatrix}
=
\begin{pmatrix}
h^{-1} & & & \\
& h^{-2} & & \\
& & \ddots & \\
& & & h^{-n}
\end{pmatrix}
\begin{pmatrix}
r_1 & r_1^2 & \cdots & r_1^n \\
r_2 & r_2^2 & \cdots & r_2^n \\
\vdots & \vdots & \ddots & \vdots \\
r_n & r_n^2 & \cdots & r_n^n
\end{pmatrix}^{-1}
\begin{pmatrix}
\mathcal{O}(h^{n+1} + \Delta_s + \Delta_{\lambda_{i_1}}) \\
\mathcal{O}(h^{n+1} + \Delta_s + \Delta_{\lambda_{i_2}}) \\
\vdots \\
\mathcal{O}(h^{n+1} + \Delta_s + \Delta_{\lambda_{i_n}})
\end{pmatrix}
\tag{31}
$$

Besides, under Assumption B.2, the orders of the other coefficients are the same as we obtain in the last section:

$$\boldsymbol{A}(\lambda_s, \lambda_t) = \mathcal{O}(1), \quad \int_{\lambda_s}^{\lambda_t} \boldsymbol{E}_{\lambda_s}(\lambda)(\lambda - \lambda_s)^k \mathrm{d}\lambda = \mathcal{O}(h^{k+1}) \tag{32}$$

Thus

$$
\begin{aligned}
&\sum_{k=1}^{n} \frac{\hat{\boldsymbol{g}}_{\theta}^{(k)}(\boldsymbol{x}_{\lambda_s}, \lambda_s) - \boldsymbol{g}_{\theta}^{(k)}(\boldsymbol{x}_{\lambda_s}, \lambda_s)}{k!} \int_{\lambda_s}^{\lambda_t} \boldsymbol{E}_{\lambda_s}(\lambda)(\lambda - \lambda_s)^k \mathrm{d}\lambda \\
&= \begin{pmatrix} \int_{\lambda_s}^{\lambda_t} \boldsymbol{E}_{\lambda_s}(\lambda)(\lambda - \lambda_s)^1 \mathrm{d}\lambda \\ \int_{\lambda_s}^{\lambda_t} \boldsymbol{E}_{\lambda_s}(\lambda)(\lambda - \lambda_s)^2 \mathrm{d}\lambda \\ \vdots \\ \int_{\lambda_s}^{\lambda_t} \boldsymbol{E}_{\lambda_s}(\lambda)(\lambda - \lambda_s)^n \mathrm{d}\lambda \end{pmatrix}^{\top} \begin{pmatrix} \hat{\boldsymbol{g}}_s^{(1)} - \boldsymbol{g}_s^{(1)} \\ \frac{\hat{\boldsymbol{g}}_s^{(2)} - \boldsymbol{g}_s^{(2)}}{2!} \\ \vdots \\ \frac{\hat{\boldsymbol{g}}_s^{(n)} - \boldsymbol{g}_s^{(n)}}{n!} \end{pmatrix} \\
&= \begin{pmatrix} \mathcal{O}(h^2) \\ \mathcal{O}(h^3) \\ \vdots \\ \mathcal{O}(h^{n+1}) \end{pmatrix}^{\top} \begin{pmatrix} h^{-1} & & & \\ & h^{-2} & & \\ & & \ddots & \\ & & & h^{-n} \end{pmatrix} \begin{pmatrix} r_1 & r_1^2 & \cdots & r_1^n \\ r_2 & r_2^2 & \cdots & r_2^n \\ \vdots & \vdots & \ddots & \vdots \\ r_n & r_n^2 & \cdots & r_n^n \end{pmatrix}^{-1} \begin{pmatrix} \mathcal{O}(h^{n+1} + \Delta_s + \Delta_{\lambda_{i_1}}) \\ \mathcal{O}(h^{n+1} + \Delta_s + \Delta_{\lambda_{i_2}}) \\ \vdots \\ \mathcal{O}(h^{n+1} + \Delta_s + \Delta_{\lambda_{i_n}}) \end{pmatrix} \\
&= \begin{pmatrix} \mathcal{O}(h) \\ \mathcal{O}(h) \\ \vdots \\ \mathcal{O}(h) \end{pmatrix}^{\top} \begin{pmatrix} r_1 & r_1^2 & \cdots & r_1^n \\ r_2 & r_2^2 & \cdots & r_2^n \\ \vdots & \vdots & \ddots & \vdots \\ r_n & r_n^2 & \cdots & r_n^n \end{pmatrix}^{-1} \begin{pmatrix} \mathcal{O}(h^{n+1} + \Delta_s + \Delta_{\lambda_{i_1}}) \\ \mathcal{O}(h^{n+1} + \Delta_s + \Delta_{\lambda_{i_2}}) \\ \vdots \\ \mathcal{O}(h^{n+1} + \Delta_s + \Delta_{\lambda_{i_n}}) \end{pmatrix} \\
&= \sum_{k=1}^{n} \mathcal{O}(h)\mathcal{O}(h^{n+1} + \Delta_s + \Delta_{\lambda_{i_k}})
\end{aligned}
$$

$$\tag{33}$$

Combining Eq. (29), Eq. (32) and Eq. (33), we can obtain the conclusion in Eq. (28). □

Then we prove Theorem 3.3 below.

*Proof.* (Proof of Theorem 3.3)

As we have discussed, the predictor step from $t_{m-1}$ to $t_m$ is a special case of the local approximation Eq. (14) with $(t_{i_n}, \ldots, t_{i_1}, s, t) = (t_{m-n-1}, \ldots, t_{m-2}, t_{m-1}, t_m)$. By Lemma B.7 we have

$$\Delta_m = \frac{\alpha_{t_m} \boldsymbol{A}(\lambda_{t_{m-1}}, \lambda_{t_m})}{\alpha_{t_{m-1}}} \Delta_{m-1} + \mathcal{O}(h) \left( \sum_{k=0}^{n} \mathcal{O}(\Delta_{m-k-1}) + \mathcal{O}(h^{n+1}) \right) \tag{34}$$

It follows that there exists constants $C, C_0$ irrelevant to $h$, so that

$$|\Delta_m| \leq \left( \frac{\alpha_{t_m} \boldsymbol{A}(\lambda_{t_{m-1}}, \lambda_{t_m})}{\alpha_{t_{m-1}}} + Ch \right) |\Delta_{m-1}| + Ch \sum_{k=0}^{n} |\Delta_{m-k-1}| + C_0 h^{n+2} \tag{35}$$

Denote $f_m := \max_{0 \leq i \leq m} |\Delta_i|$, we then have

$$|\Delta_m| \leq \left( \frac{\alpha_{t_m} \boldsymbol{A}(\lambda_{t_{m-1}}, \lambda_{t_m})}{\alpha_{t_{m-1}}} + C_1 h \right) f_{m-1} + C_0 h^{n+2} \tag{36}$$

Since $\frac{\alpha_{t_m} \boldsymbol{A}(\lambda_{t_{m-1}}, \lambda_{t_m})}{\alpha_{t_{m-1}}} \to 1$ when $h \to 0$ and it has bounded first-order derivative due to Assumption B.2, there exists a constant $C_2$, so that for any $C \geq C_2$, $\frac{\alpha_{t_m} \boldsymbol{A}(\lambda_{t_{m-1}}, \lambda_{t_m})}{\alpha_{t_{m-1}}} + Ch > 1$ for sufficiently small $h$. Thus, by taking $C_3 = \max\{C_1, C_2\}$, we have

$$f_m \leq \left( \frac{\alpha_{t_m} \boldsymbol{A}(\lambda_{t_{m-1}}, \lambda_{t_m})}{\alpha_{t_{m-1}}} + C_3 h \right) f_{m-1} + C_0 h^{n+2} \tag{37}$$

Denote $A_{m-1} := \frac{\alpha_{t_m} \boldsymbol{A}(\lambda_{t_{m-1}}, \lambda_{t_m})}{\alpha_{t_{m-1}}} + C_3 h$, by repeating Eq. (37), we have

$$f_M \le \left( \prod_{i=n}^{M-1} A_i \right) f_n + \left( \sum_{i=n+1}^{M} \prod_{j=i}^{M-1} A_j \right) C_0 h^{n+2} \tag{38}$$

By Assumption B.5, $h = \mathcal{O}(1/M)$, so we have

$$
\begin{aligned}
\prod_{i=n}^{M-1} A_i &= \frac{\alpha_{t_M} \boldsymbol{A}(\lambda_{t_n}, \lambda_{t_M})}{\alpha_{t_n}} \prod_{i=n}^{M-1} \left( 1 + \frac{\alpha_{t_{i-1}} C_3 h}{\alpha_{t_i} \boldsymbol{A}(\lambda_{t_{i-1}}, \lambda_{t_i})} \right) \\
&\le \frac{\alpha_{t_M} \boldsymbol{A}(\lambda_{t_n}, \lambda_{t_M})}{\alpha_{t_n}} \prod_{i=n}^{M-1} \left( 1 + \frac{\alpha_{t_{i-1}} C_4}{\alpha_{t_i} \boldsymbol{A}(\lambda_{t_{i-1}}, \lambda_{t_i}) M} \right) \\
&\le \frac{\alpha_{t_M} \boldsymbol{A}(\lambda_{t_n}, \lambda_{t_M})}{\alpha_{t_n}} \left( 1 + \frac{\sigma}{M} \right)^{M-n} \\
&\le C_5 e^{\sigma}
\end{aligned}
\tag{39}
$$

where $\sigma = \max_{n \le i \le M-1} \frac{\alpha_{t_{i-1}} C_4}{\alpha_{t_i} \boldsymbol{A}(\lambda_{t_{i-1}}, \lambda_{t_i})}$. Then denote $\beta := \max_{n+1 \le i \le M} \frac{\alpha_{t_M} \boldsymbol{A}(\lambda_{t_i}, \lambda_{t_M})}{\alpha_{t_i}}$, we have

$$
\begin{aligned}
\sum_{i=n+1}^{M} \prod_{j=i}^{M-1} A_j &\le \sum_{i=n+1}^{M} \frac{\alpha_{t_M} \boldsymbol{A}(\lambda_{t_i}, \lambda_{t_M})}{\alpha_{t_i}} \left( 1 + \frac{\sigma}{M} \right)^{M-i} \\
&\le \beta \sum_{i=0}^{M-n-1} \left( 1 + \frac{\sigma}{M} \right)^i \\
&= \frac{\beta M}{\sigma} \left[ \left( 1 + \frac{\sigma}{M} \right)^{M-n} - 1 \right] \\
&\le C_6 \left( e^{\sigma} - 1 \right) M
\end{aligned}
\tag{40}
$$

Then we substitute Eq. (39) and Eq. (40) into Eq. (38). Note that $M = \mathcal{O}(1/h)$ by Assumption B.5, and $f_n = \mathcal{O}(h^{n+1})$ by Assumption B.6, finally we conclude that $|\Delta_M| \le f_M = \mathcal{O}(h^{n+1})$. $\square$

## B.3  Pseudo-Order Solver

First, we provide a lemma that gives the explicit solution to Eq. (15).

**Lemma B.8.** *The solution to Eq. (15) is*

$$\frac{\hat{\boldsymbol{g}}_s^{(k)}}{k!} = \sum_{p=1}^{k} \frac{\boldsymbol{g}_{i_p} - \boldsymbol{g}_{i_0}}{\prod_{q=0, q \neq p}^{k} (\delta_p - \delta_q)} \tag{41}$$

*Proof.* Denote

$$\boldsymbol{R}_k = \begin{pmatrix} \delta_1 & \delta_1^2 & \cdots & \delta_1^k \\ \delta_2 & \delta_2^2 & \cdots & \delta_2^k \\ \vdots & \vdots & \ddots & \vdots \\ \delta_k & \delta_k^2 & \cdots & \delta_k^k \end{pmatrix} \tag{42}$$

Then the solution to Eq. (15) can be expressed as

$$\frac{\hat{\boldsymbol{g}}_s^{(k)}}{k!} = \sum_{p=1}^{k} (\boldsymbol{R}_k^{-1})_{kp} (\boldsymbol{g}_{i_p} - \boldsymbol{g}_{i_0}) \tag{43}$$

where $(\boldsymbol{R}_k^{-1})_{kp}$ is the element of $\boldsymbol{R}_k^{-1}$ at the $k$-th row and the $p$-th column. From previous studies of the inversion of the Vandermonde matrix [12], we know

$$(\boldsymbol{R}_k^{-1})_{kp} = \frac{1}{\delta_p \prod_{q=1, q \neq p}^{k} (\delta_p - \delta_q)} = \frac{1}{(\delta_p - \delta_0) \prod_{q=1, q \neq p}^{k} (\delta_p - \delta_q)} \tag{44}$$

Substituting Eq. (44) into Eq. (43), we finish the proof. $\square$

Then we prove Theorem 3.4 below:

*Proof.* (Proof of Theorem 3.4) First, we use mathematical induction to prove that

$$D_l^{(k)} = \tilde{D}_l^{(k)} := \sum_{p=1}^{k} \frac{\boldsymbol{g}_{i_{l+p}} - \boldsymbol{g}_{i_l}}{\prod_{q=0,q\neq p}^{k}(\delta_{l+p} - \delta_{l+q})}, \quad 1 \le k \le n, 0 \le l \le n-k \tag{45}$$

For $k = 1$, Eq. (45) holds by the definition of $D_l^{(k)}$. Suppose the equation holds for $k$, we then prove it holds for $k + 1$.

Define the Lagrange polynomial which passes $(\delta_{l+p}, \boldsymbol{g}_{i_{l+p}} - \boldsymbol{g}_{i_l})$ for $0 \le p \le k$:

$$P_l^{(k)}(x) := \sum_{p=1}^{k} \left(\boldsymbol{g}_{i_{l+p}} - \boldsymbol{g}_{i_l}\right) \prod_{q=0,q\neq p}^{k} \frac{x - \delta_{l+q}}{\delta_{l+p} - \delta_{l+q}}, \quad 1 \le k \le n, 0 \le l \le n-k \tag{46}$$

Then $\tilde{D}_l^{(k)} = P_l^{(k)}(x)[x^k]$ is the coefficients before the highest-order term $x^k$ in $P_l^{(k)}(x)$. We then prove that $P_l^{(k)}(x)$ satisfies the following recurrence relation:

$$P_l^{(k)}(x) = \tilde{P}_l^{(k)}(x) := \frac{(x - \delta_l)P_{l+1}^{(k-1)}(x) - (x - \delta_{l+k})P_l^{(k-1)}(x)}{\delta_{l+k} - \delta_l} \tag{47}$$

By definition, $P_{l+1}^{(k-1)}(x)$ is the $(k-1)$-th order polynomial which passes $(\delta_{l+p}, \boldsymbol{g}_{i_{l+p}} - \boldsymbol{g}_{i_l})$ for $1 \le p \le k$, and $P_l^{(k-1)}(x)$ is the $(k-1)$-th order polynomial which passes $(\delta_{l+p}, \boldsymbol{g}_{i_{l+p}} - \boldsymbol{g}_{i_l})$ for $0 \le p \le k-1$.

Thus, for $1 \le p \le k-1$, we have

$$\tilde{P}_l^{(k)}(\delta_{l+p}) = \frac{(\delta_{l+p} - \delta_l)P_{l+1}^{(k-1)}(\delta_{l+p}) - (\delta_{l+p} - \delta_{l+k})P_l^{(k-1)}(\delta_{l+p})}{\delta_{l+k} - \delta_l} = \boldsymbol{g}_{i_{l+p}} - \boldsymbol{g}_{i_l} \tag{48}$$

For $p = 0$, we have

$$\tilde{P}_l^{(k)}(\delta_l) = \frac{(\delta_l - \delta_l)P_{l+1}^{(k-1)}(\delta_l) - (\delta_l - \delta_{l+k})P_l^{(k-1)}(\delta_l)}{\delta_{l+k} - \delta_l} = \boldsymbol{g}_{i_l} - \boldsymbol{g}_{i_l} \tag{49}$$

for $p = k$, we have

$$\tilde{P}_l^{(k)}(\delta_{l+k}) = \frac{(\delta_{l+k} - \delta_l)P_{l+1}^{(k-1)}(\delta_{l+k}) - (\delta_{l+k} - \delta_{l+k})P_l^{(k-1)}(\delta_{l+k})}{\delta_{l+k} - \delta_l} = \boldsymbol{g}_{i_{l+k}} - \boldsymbol{g}_{i_l} \tag{50}$$

Therefore, $\tilde{P}_l^{(k)}(x)$ is the $k$-th order polynomial which passes $k+1$ distince points $(\delta_{l+p}, \boldsymbol{g}_{i_{l+p}} - \boldsymbol{g}_{i_l})$ for $0 \le p \le k$. Due to the uniqueness of the Lagrange polynomial, we can conclude that $P_l^{(k)}(x) = \tilde{P}_l^{(k)}(x)$. By taking the coefficients of the highest-order term, we obtain

$$\tilde{D}_l^{(k)} = \frac{\tilde{D}_{l+1}^{(k-1)} - \tilde{D}_l^{(k-1)}}{\delta_{l+k} - \delta_l} \tag{51}$$

where by the induction hypothesis we have $D_{l+1}^{(k-1)} = \tilde{D}_{l+1}^{(k-1)}, D_l^{(k-1)} = \tilde{D}_l^{(k-1)}$. Comparing Eq. (51) with the recurrence relation of $D_l^{(k)}$ in Eq. (16), it follows that $D_l^{(k)} = \tilde{D}_l^{(k)}$, which completes the mathematical induction.

Finally, by comparing the expression for $\tilde{D}_l^{(k)}$ in Eq. (45) and the expression for $\hat{\boldsymbol{g}}_s^{(k)}$ in Lemma B.8, we can conclude that $\hat{\boldsymbol{g}}_s^{(k)} = k!D_0^{(k)}$. $\qquad\square$

## B.4 Local Unbiasedness

*Proof.* (Proof of Theorem 3.2) Subtracting the local exact solution in Eq. (9) from the $(n + 1)$-th order local approximation in Eq. (14), we have the local truncation error

$$\hat{\boldsymbol{x}}_t - \boldsymbol{x}_t = \alpha_t \boldsymbol{A}(\lambda_s, \lambda_t) \left( \int_{\lambda_s}^{\lambda_t} \boldsymbol{E}_{\lambda_s}(\lambda) \boldsymbol{g}_\theta(\boldsymbol{x}_\lambda, \lambda) \mathrm{d}\lambda - \sum_{k=0}^{n} \hat{\boldsymbol{g}}_\theta^{(k)}(\boldsymbol{x}_{\lambda_s}, \lambda_s) \int_{\lambda_s}^{\lambda_t} \boldsymbol{E}_{\lambda_s}(\lambda) \frac{(\lambda - \lambda_s)^k}{k!} \mathrm{d}\lambda \right)$$

$$= \alpha_t \boldsymbol{A}(\lambda_s, \lambda_t) \int_{\lambda_s}^{\lambda_t} \boldsymbol{E}_{\lambda_s}(\lambda) \left( \boldsymbol{g}_\theta(\boldsymbol{x}_\lambda, \lambda) - \boldsymbol{g}_\theta(\boldsymbol{x}_{\lambda_s}, \lambda_s) \right) \mathrm{d}\lambda$$

$$- \alpha_t \boldsymbol{A}(\lambda_s, \lambda_t) \sum_{k=1}^{n} \hat{\boldsymbol{g}}_\theta^{(k)}(\boldsymbol{x}_{\lambda_s}, \lambda_s) \int_{\lambda_s}^{\lambda_t} \boldsymbol{E}_{\lambda_s}(\lambda) \frac{(\lambda - \lambda_s)^k}{k!} \mathrm{d}\lambda$$

$$= \alpha_t \boldsymbol{A}(\lambda_s, \lambda_t) \int_{\lambda_s}^{\lambda_t} \boldsymbol{E}_{\lambda_s}(\lambda) \left( \boldsymbol{g}_\theta(\boldsymbol{x}_\lambda, \lambda) - \boldsymbol{g}_\theta(\boldsymbol{x}_{\lambda_s}, \lambda_s) \right) \mathrm{d}\lambda$$

$$- \alpha_t \boldsymbol{A}(\lambda_s, \lambda_t) \sum_{k=1}^{n} \left( \sum_{l=1}^{n} (\boldsymbol{R}_n^{-1})_{kl} (\boldsymbol{g}_\theta(\boldsymbol{x}_{\lambda_{i_l}}, \lambda_{i_l}) - \boldsymbol{g}_\theta(\boldsymbol{x}_{\lambda_s}, \lambda_s)) \right) \int_{\lambda_s}^{\lambda_t} \boldsymbol{E}_{\lambda_s}(\lambda) \frac{(\lambda - \lambda_s)^k}{k!} \mathrm{d}\lambda$$

$$(52)$$

where $\boldsymbol{x}_\lambda$ is on the ground-truth ODE trajectory passing $\boldsymbol{x}_{\lambda_s}$, and $(\boldsymbol{R}_n^{-1})_{kl}$ is the element of the inverse matrix $\boldsymbol{R}_n^{-1}$ at the $k$-th row and the $l$-th column, as discussed in the proof of Lemma B.8. By Newton-Leibniz theorem, we have

$$\boldsymbol{g}_\theta(\boldsymbol{x}_\lambda, \lambda) - \boldsymbol{g}_\theta(\boldsymbol{x}_{\lambda_s}, \lambda_s) = \int_{\lambda_s}^{\lambda} \boldsymbol{g}_\theta^{(1)}(\boldsymbol{x}_\tau, \tau) \mathrm{d}\tau \tag{53}$$

Also, since $\boldsymbol{x}_{\lambda_{i_l}}, l = 1, \dots, n$ are on the ground-truth ODE trajectory passing $\boldsymbol{x}_{\lambda_s}$, we have

$$\boldsymbol{g}_\theta(\boldsymbol{x}_{\lambda_{i_l}}, \lambda_{i_l}) - \boldsymbol{g}_\theta(\boldsymbol{x}_{\lambda_s}, \lambda_s) = \int_{\lambda_s}^{\lambda_{i_l}} \boldsymbol{g}_\theta^{(1)}(\boldsymbol{x}_\tau, \tau) \mathrm{d}\tau \tag{54}$$

where

$$\boldsymbol{g}_\theta^{(1)}(\boldsymbol{x}_\tau, \tau) = e^{-\int_{\lambda_s}^{\tau} \boldsymbol{s}_r \mathrm{d}r} \left( \boldsymbol{f}_\theta^{(1)}(\boldsymbol{x}_\tau, \tau) - \boldsymbol{s}_\tau \boldsymbol{f}_\theta(\boldsymbol{x}_\tau, \tau) - \boldsymbol{b}_\tau \right) \tag{55}$$

Note that $\boldsymbol{s}_\lambda, \boldsymbol{l}_\lambda$ are the solution to the least square problem in Eq. (11), which makes sure $\mathbb{E}_{p_\tau^\theta(\boldsymbol{x}_\tau)} \left[ \boldsymbol{f}_\theta^{(1)}(\boldsymbol{x}_\tau, \tau) - \boldsymbol{s}_\tau \boldsymbol{f}_\theta(\boldsymbol{x}_\tau, \tau) - \boldsymbol{b}_\tau \right] = 0$. It follows that $\mathbb{E}_{p_{\lambda_s}^\theta(\boldsymbol{x}_{\lambda_s})} \left[ \boldsymbol{f}_\theta^{(1)}(\boldsymbol{x}_\tau, \tau) - \boldsymbol{s}_\tau \boldsymbol{f}_\theta(\boldsymbol{x}_\tau, \tau) - \boldsymbol{b}_\tau \right] = 0$, since $\boldsymbol{x}_\tau$ is on the ground-truth ODE trajectory passing $\boldsymbol{x}_{\lambda_s}$. Therefore, we have $\mathbb{E}_{p_{\lambda_s}^\theta(\boldsymbol{x}_{\lambda_s})} [\boldsymbol{g}_\theta(\boldsymbol{x}_\lambda, \lambda) - \boldsymbol{g}_\theta(\boldsymbol{x}_{\lambda_s}, \lambda_s)] = 0$ and $\mathbb{E}_{p_{\lambda_s}^\theta(\boldsymbol{x}_{\lambda_s})} \left[ \boldsymbol{g}_\theta(\boldsymbol{x}_{\lambda_{i_l}}, \lambda_{i_l}) - \boldsymbol{g}_\theta(\boldsymbol{x}_{\lambda_s}, \lambda_s) \right] = 0$. Substitute them into Eq. (52), we conclude that $\mathbb{E}_{p_{\lambda_s}^\theta(\boldsymbol{x}_{\lambda_s})} [\hat{\boldsymbol{x}}_t - \boldsymbol{x}_t] = 0$. □

# C  Implementation Details

## C.1  Computing the EMS and Related Integrals in the ODE Formulation

The ODE formulation and local approximation require computing some complex integrals involving $\boldsymbol{l}_\lambda, \boldsymbol{s}_\lambda, \boldsymbol{b}_\lambda$. In this section, we'll give details about how to estimate $\boldsymbol{l}_\lambda^*, \boldsymbol{s}_\lambda^*, \boldsymbol{b}_\lambda^*$ on a few datapoints, and how to use them to compute the integrals efficiently.

### C.1.1  Computing the EMS

First for the computing of $\boldsymbol{l}_\lambda^*$ in Eq. (5), note that

$$\nabla_{\boldsymbol{x}} \boldsymbol{N}_\theta(\boldsymbol{x}_\lambda, \lambda) = \sigma_\lambda \nabla_{\boldsymbol{x}} \boldsymbol{\epsilon}(\boldsymbol{x}_\lambda, \lambda) - \mathrm{diag}(\boldsymbol{l}_\lambda) \tag{56}$$

Since $\mathrm{diag}(\boldsymbol{l}_\lambda)$ is a diagonal matrix, minimizing $\mathbb{E}_{p_\lambda^\theta(\boldsymbol{x}_\lambda)} \left[ \|\nabla_{\boldsymbol{x}} \boldsymbol{N}_\theta(\boldsymbol{x}_\lambda, \lambda)\|_F^2 \right]$ is equivalent to minimizing $\mathbb{E}_{p_\lambda^\theta(\boldsymbol{x}_\lambda)} \left[ \|\mathrm{diag}^{-1}(\nabla_{\boldsymbol{x}} \boldsymbol{N}_\theta(\boldsymbol{x}_\lambda, \lambda))\|_2^2 \right] = \mathbb{E}_{p_\lambda^\theta(\boldsymbol{x}_\lambda)} \left[ \|\mathrm{diag}^{-1}(\sigma_\lambda \nabla_{\boldsymbol{x}} \boldsymbol{\epsilon}(\boldsymbol{x}_\lambda, \lambda)) - \boldsymbol{l}_\lambda\|_2^2 \right]$, where

$\text{diag}^{-1}$ denotes the operator that takes the diagonal of a matrix as a vector. Thus we have $\boldsymbol{l}_\lambda^* = \mathbb{E}_{p_\lambda^\theta(\boldsymbol{x}_\lambda)}\left[\text{diag}^{-1}(\sigma_\lambda \nabla_{\boldsymbol{x}}\boldsymbol{\epsilon}(\boldsymbol{x}_\lambda, \lambda))\right]$.

However, this formula for $\boldsymbol{l}_\lambda^*$ requires computing the diagonal of the full Jacobian of the noise prediction model, which typically has $\mathcal{O}(d^2)$ time complexity for $d$-dimensional data and is unacceptable when $d$ is large. Fortunately, the cost can be reduced to $\mathcal{O}(d)$ by utilizing stochastic diagonal estimators and employing the efficient Jacobian-vector-product operator provided by forward-mode automatic differentiation in deep learning frameworks.

For a $d$-by-$d$ matrix $\boldsymbol{D}$, its diagonal can be unbiasedly estimated by [4]

$$\text{diag}^{-1}(\boldsymbol{D}) = \left[\sum_{k=1}^{s}(\boldsymbol{D}\boldsymbol{v}_k) \odot \boldsymbol{v}_k\right] \oslash \left[\sum_{k=1}^{s} \boldsymbol{v}_k \odot \boldsymbol{v}_k\right] \tag{57}$$

where $\boldsymbol{v}_k \sim p(\boldsymbol{v})$ are $d$-dimensional i.i.d. samples with zero mean, $\odot$ is the element-wise multiplication i.e., Hadamard product, and $\oslash$ is the element-wise division. The stochastic diagonal estimator is analogous to the famous Hutchinson's trace estimator [20]. By taking $p(\boldsymbol{v})$ as the Rademacher distribution, we have $\boldsymbol{v}_k \odot \boldsymbol{v}_k = \boldsymbol{1}$, and the denominator can be omitted. For simplicity, we use regular multiplication and division symbols, assuming they are element-wise between vectors. Then $\boldsymbol{l}_\lambda^*$ can be expressed as:

$$\boldsymbol{l}_\lambda^* = \mathbb{E}_{p_\lambda^\theta(\boldsymbol{x}_\lambda)p(\boldsymbol{v})}\left[(\sigma_\lambda \nabla_{\boldsymbol{x}}\boldsymbol{\epsilon}_\theta(\boldsymbol{x}_\lambda, \lambda)\boldsymbol{v})\boldsymbol{v}\right] \tag{58}$$

which is an unbiased estimation when we replace the expectation with mean on finite samples $\boldsymbol{x}_\lambda \sim p_\lambda^\theta(\boldsymbol{x}_\lambda), \boldsymbol{v} \sim p(\boldsymbol{v})$. The process for estimating $\boldsymbol{l}_\lambda^*$ can easily be paralleled on multiple devices by computing $\sum(\sigma_\lambda \nabla_{\boldsymbol{x}}\boldsymbol{\epsilon}_\theta(\boldsymbol{x}_\lambda, \lambda)\boldsymbol{v})\boldsymbol{v}$ on separate datapoints and gather them in the end.

Next, for the computing of $\boldsymbol{s}_\lambda^*, \boldsymbol{b}_\lambda^*$ in Eq. (11), note that it's a simple least square problem. By taking partial derivatives w.r.t. $\boldsymbol{s}_\lambda, \boldsymbol{b}_\lambda$ and set them to 0, we have

$$\begin{cases} \mathbb{E}_{p_\lambda^\theta(\boldsymbol{x}_\lambda)}\left[\left(\boldsymbol{f}_\theta^{(1)}(\boldsymbol{x}_\lambda, \lambda) - \boldsymbol{s}_\lambda^*\boldsymbol{f}_\theta(\boldsymbol{x}_\lambda, \lambda) - \boldsymbol{b}_\lambda^*\right)\boldsymbol{f}_\theta(\boldsymbol{x}_\lambda, \lambda)\right] = 0 \\ \mathbb{E}_{p_\lambda^\theta(\boldsymbol{x}_\lambda)}\left[\boldsymbol{f}_\theta^{(1)}(\boldsymbol{x}_\lambda, \lambda) - \boldsymbol{s}_\lambda^*\boldsymbol{f}_\theta(\boldsymbol{x}_\lambda, \lambda) - \boldsymbol{b}_\lambda^*\right] = 0 \end{cases} \tag{59}$$

And we obtain the explicit formula for $\boldsymbol{s}_\lambda^*, \boldsymbol{b}_\lambda^*$

$$\boldsymbol{s}_\lambda^* = \frac{\mathbb{E}_{p_\lambda^\theta(\boldsymbol{x}_\lambda)}\left[\boldsymbol{f}_\theta(\boldsymbol{x}_\lambda, \lambda)\boldsymbol{f}_\theta^{(1)}(\boldsymbol{x}_\lambda, \lambda)\right] - \mathbb{E}_{p_\lambda^\theta(\boldsymbol{x}_\lambda)}[\boldsymbol{f}_\theta(\boldsymbol{x}_\lambda, \lambda)]\mathbb{E}_{p_\lambda^\theta(\boldsymbol{x}_\lambda)}\left[\boldsymbol{f}_\theta^{(1)}(\boldsymbol{x}_\lambda, \lambda)\right]}{\mathbb{E}_{p_\lambda^\theta(\boldsymbol{x}_\lambda)}[\boldsymbol{f}_\theta(\boldsymbol{x}_\lambda, \lambda)\boldsymbol{f}_\theta(\boldsymbol{x}_\lambda, \lambda)] - \mathbb{E}_{p_\lambda^\theta(\boldsymbol{x}_\lambda)}[\boldsymbol{f}_\theta(\boldsymbol{x}_\lambda, \lambda)]\mathbb{E}_{p_\lambda^\theta(\boldsymbol{x}_\lambda)}[\boldsymbol{f}_\theta(\boldsymbol{x}_\lambda, \lambda)]} \tag{60}$$

$$\boldsymbol{b}_\lambda^* = \mathbb{E}_{p_\lambda^\theta(\boldsymbol{x}_\lambda)}[\boldsymbol{f}^{(1)}(\boldsymbol{x}_\lambda, \lambda)] - \boldsymbol{s}_\lambda^*\mathbb{E}_{p_\lambda^\theta(\boldsymbol{x}_\lambda)}[\boldsymbol{f}_\theta(\boldsymbol{x}_\lambda, \lambda)] \tag{61}$$

which are unbiased least square estimators when we replace the expectation with mean on finite samples $\boldsymbol{x}_\lambda \sim p_\lambda^\theta(\boldsymbol{x}_\lambda)$. Also, the process for estimating $\boldsymbol{s}_\lambda^*, \boldsymbol{b}_\lambda^*$ can be paralleled on multiple devices by computing $\sum \boldsymbol{f}_\theta, \sum \boldsymbol{f}_\theta^{(1)}, \sum \boldsymbol{f}_\theta\boldsymbol{f}_\theta, \sum \boldsymbol{f}_\theta\boldsymbol{f}_\theta^{(1)}$ on separate datapoints and gather them in the end. Thus, the estimation of $\boldsymbol{s}_\lambda^*, \boldsymbol{b}_\lambda^*$ involving evaluating $\boldsymbol{f}_\theta$ and $\boldsymbol{f}_\theta^{(1)}$ on $\boldsymbol{x}_\lambda$. $\boldsymbol{f}_\theta$ is a direct transformation of $\boldsymbol{\epsilon}_\theta$ and requires a single forward pass. For $\boldsymbol{f}_\theta^{(1)}$, we have

$$\begin{aligned} \boldsymbol{f}_\theta^{(1)}(\boldsymbol{x}_\lambda, \lambda) &= \frac{\partial \boldsymbol{f}_\theta(\boldsymbol{x}_\lambda, \lambda)}{\partial \lambda} + \nabla_{\boldsymbol{x}}\boldsymbol{f}_\theta(\boldsymbol{x}_\lambda, \lambda)\frac{\mathrm{d}\boldsymbol{x}_\lambda}{\mathrm{d}\lambda} \\ &= e^{-\lambda}\left(\boldsymbol{\epsilon}_\theta^{(1)}(\boldsymbol{x}_\lambda, \lambda) - \boldsymbol{\epsilon}_\theta(\boldsymbol{x}_\lambda, \lambda)\right) - \frac{\dot{\boldsymbol{l}}_\lambda\alpha_\lambda - \dot{\alpha}_\lambda\boldsymbol{l}_\lambda}{\alpha_\lambda^2}\boldsymbol{x}_\lambda - \frac{\boldsymbol{l}_\lambda}{\alpha_\lambda}\left(\frac{\dot{\alpha}_\lambda}{\alpha_\lambda}\boldsymbol{x}_\lambda - \sigma_\lambda\boldsymbol{\epsilon}_\theta(\boldsymbol{x}_\lambda, \lambda)\right) \\ &= e^{-\lambda}\left((\boldsymbol{l}_\lambda - 1)\boldsymbol{\epsilon}_\theta(\boldsymbol{x}_\lambda, \lambda) + \boldsymbol{\epsilon}_\theta^{(1)}(\boldsymbol{x}_\lambda, \lambda)\right) - \frac{\dot{\boldsymbol{l}}_\lambda\boldsymbol{x}_\lambda}{\alpha_\lambda} \end{aligned} \tag{62}$$

After we obtain $\boldsymbol{l}_\lambda$, $\dot{\boldsymbol{l}}_\lambda$ can be estimated by finite difference. To compute $\boldsymbol{\epsilon}_\theta^{(1)}(\boldsymbol{x}_\lambda, \lambda)$, we have

$$\begin{aligned} \boldsymbol{\epsilon}_\theta^{(1)}(\boldsymbol{x}_\lambda, \lambda) &= \partial_\lambda\boldsymbol{\epsilon}_\theta(\boldsymbol{x}_\lambda, \lambda) + \nabla_{\boldsymbol{x}}\boldsymbol{\epsilon}_\theta(\boldsymbol{x}_\lambda, \lambda)\frac{\mathrm{d}\boldsymbol{x}_\lambda}{\mathrm{d}\lambda} \\ &= \partial_\lambda\boldsymbol{\epsilon}_\theta(\boldsymbol{x}_\lambda, \lambda) + \nabla_{\boldsymbol{x}}\boldsymbol{\epsilon}_\theta(\boldsymbol{x}_\lambda, \lambda)\left(\frac{\dot{\alpha}_\lambda}{\alpha_\lambda}\boldsymbol{x}_\lambda - \sigma_\lambda\boldsymbol{\epsilon}_\theta(\boldsymbol{x}_\lambda, \lambda)\right) \end{aligned} \tag{63}$$

which can also be computed with the Jacobian-vector-product operator.

In conclusion, for any $\lambda$, $\boldsymbol{l}_\lambda^*, \boldsymbol{s}_\lambda^*, \boldsymbol{b}_\lambda^*$ can be efficiently and unbiasedly estimated by sampling a few datapoints $\boldsymbol{x}_\lambda \sim p_\lambda^\theta(\boldsymbol{x}_\lambda)$ and using the Jacobian-vector-product.

### C.1.2 Integral Precomputing

In the local approximation in Eq. (14), there are three integrals involving the EMS, which are $\boldsymbol{A}(\lambda_s, \lambda_t), \int_{\lambda_s}^{\lambda_t} \boldsymbol{E}_{\lambda_s}(\lambda)\boldsymbol{B}_{\lambda_s}(\lambda)\mathrm{d}\lambda, \int_{\lambda_s}^{\lambda_t} \boldsymbol{E}_{\lambda_s}(\lambda)\frac{(\lambda-\lambda_s)^k}{k!}\mathrm{d}\lambda$. Define the following terms, which are also evaluated at $\lambda_{j_0}, \lambda_{j_1}, \ldots, \lambda_{j_N}$ and can be precomputed in $\mathcal{O}(N)$ time:

$$
\begin{aligned}
\boldsymbol{L}_\lambda &= \int_{\lambda_T}^\lambda \boldsymbol{l}_\tau \mathrm{d}\tau \\
\boldsymbol{S}_\lambda &= \int_{\lambda_T}^\lambda \boldsymbol{s}_\tau \mathrm{d}\tau \\
\boldsymbol{B}_\lambda &= \int_{\lambda_T}^\lambda e^{-\int_{\lambda_T}^r \boldsymbol{s}_\tau \mathrm{d}\tau} \boldsymbol{b}_r \mathrm{d}r = \int_{\lambda_T}^\lambda e^{-\boldsymbol{S}_r} \boldsymbol{b}_r \mathrm{d}r \\
\boldsymbol{C}_\lambda &= \int_{\lambda_T}^\lambda \left( e^{\int_{\lambda_T}^u (\boldsymbol{l}_\tau+\boldsymbol{s}_\tau)\mathrm{d}\tau} \int_{\lambda_T}^u e^{-\int_{\lambda_T}^r \boldsymbol{s}_\tau \mathrm{d}\tau} \boldsymbol{b}_r \mathrm{d}r \right) \mathrm{d}u = \int_{\lambda_T}^\lambda e^{\boldsymbol{L}_u+\boldsymbol{S}_u} \boldsymbol{B}_u \mathrm{d}u \\
\boldsymbol{I}_\lambda &= \int_{\lambda_T}^\lambda e^{\int_{\lambda_T}^r (\boldsymbol{l}_\tau+\boldsymbol{s}_\tau)\mathrm{d}\tau} \mathrm{d}r = \int_{\lambda_T}^\lambda e^{\boldsymbol{L}_r+\boldsymbol{S}_r} \mathrm{d}r
\end{aligned}
\tag{64}
$$

Then for any $\lambda_s, \lambda_t$, we can verify that the first two integrals can be expressed as

$$
\begin{aligned}
\boldsymbol{A}(\lambda_s, \lambda_t) &= e^{\boldsymbol{L}_{\lambda_s}-\boldsymbol{L}_{\lambda_t}} \\
\int_{\lambda_s}^{\lambda_t} \boldsymbol{E}_{\lambda_s}(\lambda)\boldsymbol{B}_{\lambda_s}(\lambda)\mathrm{d}\lambda &= e^{-\boldsymbol{L}_{\lambda_s}}(\boldsymbol{C}_{\lambda_t} - \boldsymbol{C}_{\lambda_s} - \boldsymbol{B}_{\lambda_s}(\boldsymbol{I}_{\lambda_t} - \boldsymbol{I}_{\lambda_s}))
\end{aligned}
\tag{65}
$$

which can be computed in $\mathcal{O}(1)$ time. For the third and last integral, denote it as $\boldsymbol{E}_{\lambda_s,\lambda_t}^{(k)}$, i.e.,

$$
\boldsymbol{E}_{\lambda_s,\lambda_t}^{(k)} = \int_{\lambda_s}^{\lambda_t} \boldsymbol{E}_{\lambda_s}(\lambda)\frac{(\lambda-\lambda_s)^k}{k!}\mathrm{d}\lambda
\tag{66}
$$

We need to compute it for $0 \le k \le n$ and for every local transition time pair $(\lambda_s, \lambda_t)$ in the sampling process. For $k = 0$, we have

$$
\boldsymbol{E}_{\lambda_s,\lambda_t}^{(0)} = e^{-\boldsymbol{L}_{\lambda_s}-\boldsymbol{S}_{\lambda_s}}(\boldsymbol{I}_{\lambda_t} - \boldsymbol{I}_{\lambda_s})
\tag{67}
$$

which can also be computed in $\mathcal{O}(1)$ time. But for $k > 0$, we no longer have such a simplification technique. Still, for any fixed timestep schedule $\{\lambda_i\}_{i=0}^M$ during the sampling process, we can use a lazy precomputing strategy: compute $\boldsymbol{E}_{\lambda_{i-1},\lambda_i}^{(k)}, 1 \le i \le M$ when generating the first sample, store it with a unique key $(k, i)$ and retrieve it in $\mathcal{O}(1)$ in the following sampling process.

### C.2 Algorithm

We provide the pseudocode of the local approximation and global solver in Algorithm 1 and Algorithm 2, which concisely describes how we implement DPM-Solver-v3.

---

**Algorithm 1** $(n+1)$-th order local approximation: $\text{LUpdate}_{n+1}$

---

**Require:** noise schedule $\alpha_t, \sigma_t$, coefficients $\boldsymbol{l}_\lambda, \boldsymbol{s}_\lambda, \boldsymbol{b}_\lambda$
**Input:** transition time pair $(s,t)$, $\boldsymbol{x}_s$, $n$ extra timesteps $\{t_{i_k}\}_{k=1}^n$, $\boldsymbol{g}_\theta$ values $(\boldsymbol{g}_{i_n}, \ldots, \boldsymbol{g}_{i_1}, \boldsymbol{g}_s)$ at $\{(\boldsymbol{x}_{\lambda_{i_k}}, t_{i_k})\}_{k=1}^n$ and $(\boldsymbol{x}_s, s)$
**Input Format:** $\{t_{i_n}, \boldsymbol{g}_{i_n}\}, \ldots, \{t_{i_1}, \boldsymbol{g}_{i_1}\}, \{s, \boldsymbol{x}_s, \boldsymbol{g}_s\}, t$

1: Compute $\boldsymbol{A}(\lambda_s, \lambda_t)$, $\int_{\lambda_s}^{\lambda_t} \boldsymbol{E}_{\lambda_s}(\lambda) \boldsymbol{B}_{\lambda_s}(\lambda) \mathrm{d}\lambda$, $\int_{\lambda_s}^{\lambda_t} \boldsymbol{E}_{\lambda_s}(\lambda) \frac{(\lambda - \lambda_s)^k}{k!} \mathrm{d}\lambda$ (Appendix C.1.2)

2: $\delta_k = \lambda_{i_k} - \lambda_s, \quad k = 1, \ldots, n$

3: $\begin{pmatrix} \hat{\boldsymbol{g}}_s^{(1)} \\ \frac{\hat{\boldsymbol{g}}_s^{(2)}}{2!} \\ \vdots \\ \frac{\hat{\boldsymbol{g}}_s^{(n)}}{n!} \end{pmatrix} \leftarrow \begin{pmatrix} \delta_1 & \delta_1^2 & \cdots & \delta_1^n \\ \delta_2 & \delta_2^2 & \cdots & \delta_2^n \\ \vdots & \vdots & \ddots & \vdots \\ \delta_n & \delta_n^2 & \cdots & \delta_n^n \end{pmatrix}^{-1} \begin{pmatrix} \boldsymbol{g}_{i_1} - \boldsymbol{g}_s \\ \boldsymbol{g}_{i_2} - \boldsymbol{g}_s \\ \vdots \\ \boldsymbol{g}_{i_n} - \boldsymbol{g}_s \end{pmatrix}$ (Eq. (13))

4: $\hat{\boldsymbol{x}}_t \leftarrow \alpha_t \boldsymbol{A}(\lambda_s, \lambda_t) \left( \frac{\boldsymbol{x}_s}{\alpha_s} - \int_{\lambda_s}^{\lambda_t} \boldsymbol{E}_{\lambda_s}(\lambda) \boldsymbol{B}_{\lambda_s}(\lambda) \mathrm{d}\lambda - \sum_{k=0}^n \hat{\boldsymbol{g}}_s^{(k)} \int_{\lambda_s}^{\lambda_t} \boldsymbol{E}_{\lambda_s}(\lambda) \frac{(\lambda - \lambda_s)^k}{k!} \mathrm{d}\lambda \right)$

    (Eq. (14))

**Output:** $\hat{\boldsymbol{x}}_t$

---

---

**Algorithm 2** $(n+1)$-th order multistep predictor-corrector algorithm

---

**Require:** noise prediction model $\boldsymbol{\epsilon}_\theta$, noise schedule $\alpha_t, \sigma_t$, coefficients $\boldsymbol{l}_\lambda, \boldsymbol{s}_\lambda, \boldsymbol{b}_\lambda$, cache $Q_1, Q_2$
**Input:** timesteps $\{t_i\}_{i=0}^M$, initial value $\boldsymbol{x}_0$

1: $Q_1 \overset{cache}{\Leftarrow} \boldsymbol{x}_0$

2: $Q_2 \overset{cache}{\Leftarrow} \boldsymbol{\epsilon}_\theta(\boldsymbol{x}_0, t_0)$

3: **for** $m = 1$ **to** $M$ **do**

4:      $n_m \leftarrow \min\{n+1, m\}$

5:      $\hat{\boldsymbol{x}}_{m-n_m}, \ldots, \hat{\boldsymbol{x}}_{m-1} \overset{fetch}{\Leftarrow} Q_1$

6:      $\hat{\boldsymbol{\epsilon}}_{m-n_m}, \ldots, \hat{\boldsymbol{\epsilon}}_{m-1} \overset{fetch}{\Leftarrow} Q_2$

7:      $\hat{\boldsymbol{g}}_l \leftarrow e^{-\int_{\lambda_{m-1}}^{\lambda_l} \boldsymbol{s}_\tau \mathrm{d}\tau} \frac{\sigma_{\lambda_l} \hat{\boldsymbol{\epsilon}}_l - \boldsymbol{l}_{\lambda_l} \hat{\boldsymbol{x}}_l}{\alpha_{\lambda_l}} - \int_{\lambda_{m-1}}^{\lambda_l} e^{-\int_{\lambda_{m-1}}^{r} \boldsymbol{s}_\tau \mathrm{d}\tau} \boldsymbol{b}_r \mathrm{d}r, \quad l = m - n_m, \ldots, m - 1$

       (Eq. (8))

8:      $\hat{\boldsymbol{x}}_m \leftarrow \text{LUpdate}_{n_m}(\{t_{m-n_m}, \hat{\boldsymbol{g}}_{m-n_m}\}, \ldots, \{t_{m-2}, \hat{\boldsymbol{g}}_{m-2}\}, \{t_{m-1}, \hat{\boldsymbol{x}}_{m-1}, \hat{\boldsymbol{g}}_{m-1}\}, t_m)$

9:      **if** $m \neq M$ **then**

10:        $\hat{\boldsymbol{\epsilon}}_m \leftarrow \boldsymbol{\epsilon}_\theta(\hat{\boldsymbol{x}}_m, t_m)$

11:        $\hat{\boldsymbol{g}}_m \leftarrow e^{-\int_{\lambda_{m-1}}^{\lambda_m} \boldsymbol{s}_\tau \mathrm{d}\tau} \frac{\sigma_{\lambda_m} \hat{\boldsymbol{\epsilon}}_m - \boldsymbol{l}_{\lambda_m} \hat{\boldsymbol{x}}_m}{\alpha_{\lambda_m}} - \int_{\lambda_{m-1}}^{\lambda_m} e^{-\int_{\lambda_{m-1}}^{r} \boldsymbol{s}_\tau \mathrm{d}\tau} \boldsymbol{b}_r \mathrm{d}r$ (Eq. (8))

12:        $\hat{\boldsymbol{x}}_m^c \leftarrow \text{LUpdate}_{n_m}(\{t_{m-n_m+1}, \hat{\boldsymbol{g}}_{m-n_m+1}\}, \ldots, \{t_{m-2}, \hat{\boldsymbol{g}}_{m-2}\}, \{t_m, \hat{\boldsymbol{g}}_m\},$

                          $\{t_{m-1}, \hat{\boldsymbol{x}}_{m-1}, \hat{\boldsymbol{g}}_{m-1}\}, t_m)$

13:        $\hat{\boldsymbol{\epsilon}}_m^c \leftarrow \hat{\boldsymbol{\epsilon}}_m + \boldsymbol{l}_{\lambda_m}(\hat{\boldsymbol{x}}_m^c - \hat{\boldsymbol{x}}_m)/\sigma_{\lambda_m}$ (to ensure $\hat{\boldsymbol{g}}_m^c = \hat{\boldsymbol{g}}_m$)

14:        $Q_1 \overset{cache}{\Leftarrow} \hat{\boldsymbol{x}}_m^c$

15:        $Q_2 \overset{cache}{\Leftarrow} \hat{\boldsymbol{\epsilon}}_m^c$

16:      **end if**

17: **end for**

**Output:** $\hat{\boldsymbol{x}}_M$

---

# D   Experiment Details

In this section, we provide more experiment details for each setting, including the codebases and the configurations for evaluation, EMS computing and sampling. Unless otherwise stated, we utilize the forward-mode automatic differentiation (`torch.autograd.forward_ad`) provided by PyTorch [39] to compute the Jacobian-vector-products (JVPs). Also, as stated in Section 3.4, we draw datapoints

$x_\lambda$ from the marginal distribution $q_\lambda$ defined by the forward diffusion process starting from some data distribution $q_0$, instead of the model distribution $p_\lambda^\theta$.

## D.1 ScoreSDE on CIFAR10

**Codebase and evaluation** For unconditional sampling on CIFAR10 [24], one experiment setting is based on the pretrained pixel-space diffusion model provided by ScoreSDE [51]. We use their official codebase of PyTorch implementation, and their checkpoint `checkpoint_8.pth` under `vp/cifar10_ddpmpp_deep_continuous` config. We adopt their own statistic file and code for computing FID.

**EMS computing** We estimate the EMS at $N = 1200$ uniform timesteps $\lambda_{j_0}, \lambda_{j_1}, \ldots, \lambda_{j_N}$ by drawing $K = 4096$ datapoints $x_{\lambda_0} \sim q_0$, where $q_0$ is the distribution of the training set. We compute two sets of EMS, corresponding to start time $\epsilon = 10^{-3}$ (NFE$\leq$ 10) and $\epsilon = 10^{-4}$ (NFE>10) in the sampling process respectively. The total time for EMS computing is $\sim$7h on 8 GPU cards of NVIDIA A40.

**Sampling** Following previous works [31, 32, 58], we use start time $\epsilon = 10^{-3}$ (NFE$\leq$ 10) and $\epsilon = 10^{-4}$ (NFE>10), end time $T = 1$ and adopt the uniform logSNR timestep schedule. For DPM-Solver-v3, we use the 3rd-order predictor with the 3rd-order corrector by default. In particular, we change to the pseudo 3rd-order predictor at 5 NFE to further boost the performance.

## D.2 EDM on CIFAR10

**Codebase and evaluation** For unconditional sampling on CIFAR10 [24], another experiment setting is based on the pretrained pixel-space diffusion model provided by EDM [21]. We use their official codebase of PyTorch implementation, and their checkpoint `edm-cifar10-32x32-uncond-vp.pkl`. For consistency, we borrow the statistic file and code from ScoreSDE [51] for computing FID.

**EMS computing** Since the pretrained models of EDM are stored within the pickles, we fail to use `torch.autograd.forward_ad` for computing JVPs. Instead, we use `torch.autograd.functional.jvp`, which is much slower since it employs the double backward trick. We estimate two sets of EMS. One corresponds to $N = 1200$ uniform timesteps $\lambda_{j_0}, \lambda_{j_1}, \ldots, \lambda_{j_N}$ and $K = 1024$ datapoints $x_{\lambda_0} \sim q_0$, where $q_0$ is the distribution of the training set. The other corresponds to $N = 120, K = 4096$. They are used when NFE<10 and NFE$\geq$10 respectively. The total time for EMS computing is $\sim$3.5h on 8 GPU cards of NVIDIA A40.

**Sampling** Following EDM, we use start time $t_{\min} = 0.002$ and end time $t_{\max} = 80.0$, but adopt the uniform logSNR timestep schedule which performs better in practice. For DPM-Solver-v3, we use the 3rd-order predictor and additionally employ the 3rd-order corrector when NFE$\leq$ 6. In particular, we change to the pseudo 3rd-order predictor at 5 NFE to further boost the performance.

## D.3 Latent-Diffusion on LSUN-Bedroom

**Codebase and evaluation** The unconditional sampling on LSUN-Bedroom [55] is based on the pretrained latent-space diffusion model provided by Latent-Diffusion [43]. We use their official codebase of PyTorch implementation and their default checkpoint. We borrow the statistic file and code from Guided-Diffusion [10] for computing FID.

**EMS computing** We estimate the EMS at $N = 120$ uniform timesteps $\lambda_{j_0}, \lambda_{j_1}, \ldots, \lambda_{j_N}$ by drawing $K = 1024$ datapoints $x_{\lambda_0} \sim q_0$, where $q_0$ is the distribution of the latents of the training set. The total time for EMS computing is $\sim$12min on 8 GPU cards of NVIDIA A40.

**Sampling** Following previous works [58], we use start time $\epsilon = 10^{-3}$, end time $T = 1$ and adopt the uniform $t$ timestep schedule. For DPM-Solver-v3, we use the 3rd-order predictor with the pseudo 4th-order corrector.

## D.4 Guided-Diffusion on ImageNet-256

**Codebase and evaluation** The conditional sampling on ImageNet-256 [9] is based on the pretrained pixel-space diffusion model provided by Guided-Diffusion [10]. We use their official codebase of PyTorch implementation and their two checkpoints: the conditional diffusion model

`256x256_diffusion.pt` and the classifier `256x256_classifier.pt`. We adopt their own statistic file and code for computing FID.

**EMS computing** We estimate the EMS at $N = 500$ uniform timesteps $\lambda_{j_0}, \lambda_{j_1}, \ldots, \lambda_{j_N}$ by drawing $K = 1024$ datapoints $\boldsymbol{x}_{\lambda_0} \sim q_0$, where $q_0$ is the distribution of the training set. Also, we find that the FID metric on the ImageNet-256 dataset behaves specially, and degenerated $\boldsymbol{l}_\lambda$ ($\boldsymbol{l}_\lambda = \boldsymbol{1}$) performs better. The total time for EMS computing is $\sim$9.5h on 8 GPU cards of NVIDIA A40.

**Sampling** Following previous works [31, 32, 58], we use start time $\epsilon = 10^{-3}$, end time $T = 1$ and adopt the uniform $t$ timestep schedule. For DPM-Solver-v3, we use the 2nd-order predictor with the pseudo 3rd-order corrector.

### D.5 Stable-Diffusion on MS-COCO2014 prompts

**Codebase and evaluation** The text-to-image sampling on MS-COCO2014 [26] prompts is based on the pretrained latent-space diffusion model provided by Stable-Diffusion [43]. We use their official codebase of PyTorch implementation and their checkpoint `sd-v1-4.ckpt`. We compute MSE on randomly selected captions from the MS-COCO2014 validation dataset, as detailed in Section 4.1.

**EMS computing** We estimate the EMS at $N = 250$ uniform timesteps $\lambda_{j_0}, \lambda_{j_1}, \ldots, \lambda_{j_N}$ by drawing $K = 1024$ datapoints $\boldsymbol{x}_{\lambda_0} \sim q_0$. Since Stable-Diffusion is trained on the LAION-5B dataset [46], there is a gap between the images in the MS-COCO2014 validation dataset and the images generated by Stable-Diffusion with certain guidance scale. Thus, we choose $q_0$ to be the distribution of the latents generated by Stable-Diffusion with corresponding guidance scale, using 200-step DPM-Solver++ [32]. We generate these latents with random captions and Gaussian noise different from those we use to compute MSE. The total time for EMS computing is $\sim$11h on 8 GPU cards of NVIDIA A40 for each guidance scale.

**Sampling** Following previous works [32, 58], we use start time $\epsilon = 10^{-3}$, end time $T = 1$ and adopt the uniform $t$ timestep schedule. For DPM-Solver-v3, we use the 2nd-order predictor with the pseudo 3rd-order corrector.

### D.6 License

Table 3: The used datasets, codes and their licenses.

| Name | URL | Citation | License |
|------|-----|----------|---------|
| CIFAR10 | https://www.cs.toronto.edu/~kriz/cifar.html | [24] | \ |
| LSUN-Bedroom | https://www.yf.io/p/lsun | [55] | \ |
| ImageNet-256 | https://www.image-net.org | [9] | \ |
| MS-COCO2014 | https://cocodataset.org | [26] | CC BY 4.0 |
| ScoreSDE | https://github.com/yang-song/score_sde_pytorch | [51] | Apache-2.0 |
| EDM | https://github.com/NVlabs/edm | [21] | CC BY-NC-SA 4.0 |
| Guided-Diffusion | https://github.com/openai/guided-diffusion | [10] | MIT |
| Latent-Diffusion | https://github.com/CompVis/latent-diffusion | [43] | MIT |
| Stable-Diffusion | https://github.com/CompVis/stable-diffusion | [43] | CreativeML Open RAIL-M |
| DPM-Solver++ | https://github.com/LuChengTHU/dpm-solver | [32] | MIT |
| UniPC | https://github.com/wl-zhao/UniPC | [58] | \ |

We list the used datasets, codes and their licenses in Table 3.

## E   Runtime Comparison

As we have mentioned in Section 4, the runtime of DPM-Solver-v3 is almost the same as other solvers (DDIM [48], DPM-Solver [31], DPM-Solver++ [32], UniPC [58], etc.) as long as they use the same NFE. This is because the main computation costs are the serial evaluations of the large neural network $\epsilon_\theta$, and the other coefficients are either analytically computed [48, 31, 32, 58], or precomputed (DPM-Solver-v3), thus having neglectable costs.

Table 4 shows the runtime of DPM-Solver-v3 and some other solvers on a single NVIDIA A40 under different settings. We use `torch.cuda.Event` and `torch.cuda.synchronize` to accurately compute the runtime. We evaluate the runtime on 8 batches (dropping the first batch since it contains

Table 4: Runtime of different methods to generate a single batch (second / batch, ±std) on a single NVIDIA A40, varying the number of function evaluations (NFE). We don't include the runtime of the decoding stage for latent-space DPMs.

| Method | NFE | | | |
|---|---|---|---|---|
| | 5 | 10 | 15 | 20 |
| *CIFAR10 [24], ScoreSDE [51] (batch size = 128)* | | | | |
| DPM-Solver++ [32] | 1.253(±0.0014) | 2.503(±0.0017) | 3.754(±0.0042) | 5.010(±0.0048) |
| UniPC [58] | 1.268(±0.0012) | 2.532(±0.0018) | 3.803(±0.0037) | 5.080(±0.0049) |
| DPM-Solver-v3 | 1.273(±0.0005) | 2.540(±0.0023) | 3.826(±0.0039) | 5.108(±0.0055) |
| *CIFAR10 [24], EDM [21] (batch size = 128)* | | | | |
| DPM-Solver++ [32] | 1.137(±0.0011) | 2.278(±0.0015) | 3.426(±0.0024) | 4.569(±0.0031) |
| UniPC [58] | 1.142(±0.0016) | 2.289(±0.0019) | 3.441(±0.0035) | 4.590(±0.0021) |
| DPM-Solver-v3 | 1.146(±0.0010) | 2.293(±0.0015) | 3.448(±0.0018) | 4.600(±0.0027) |
| *LSUN-Bedroom [55], Latent-Diffusion [43] (batch size = 32)* | | | | |
| DPM-Solver++ [32] | 1.302(±0.0009) | 2.608(±0.0010) | 3.921(±0.0023) | 5.236(±0.0045) |
| UniPC [58] | 1.305(±0.0005) | 2.616(±0.0019) | 3.934(±0.0033) | 5.244(±0.0043) |
| DPM-Solver-v3 | 1.302(±0.0010) | 2.620(±0.0027) | 3.932(±0.0028) | 5.290(±0.0030) |
| *ImageNet256 [9], Guided-Diffusion [10] (batch size = 4)* | | | | |
| DPM-Solver++ [32] | 1.594(±0.0011) | 3.194(±0.0018) | 4.792(±0.0031) | 6.391(±0.0045) |
| UniPC [58] | 1.606(±0.0026) | 3.205(±0.0025) | 4.814(±0.0049) | 6.427(±0.0060) |
| DPM-Solver-v3 | 1.601(±0.0059) | 3.229(±0.0031) | 4.807(±0.0068) | 6.458(±0.0257) |
| *MS-COCO2014 [26], Stable-Diffusion [43] (batch size = 4)* | | | | |
| DPM-Solver++ [32] | 1.732(±0.0012) | 3.464(±0.0020) | 5.229(±0.0027) | 6.974(±0.0013) |
| UniPC [58] | 1.735(±0.0012) | 3.484(±0.0364) | 5.212(±0.0015) | 6.988(±0.0035) |
| DPM-Solver-v3 | 1.731(±0.0008) | 3.471(±0.0011) | 5.211(±0.0030) | 6.945(±0.0022) |

extra initializations) and report the mean and std. We can see that the runtime is proportional to NFE and has a difference of about ±1% for different solvers, which confirms our statement. Therefore, the speedup for the NFE is almost the actual speedup of the runtime.

## F Quantitative Results

Table 5: Quantitative results on LSUN-Bedroom [55]. We report the FID↓ of the methods with different numbers of function evaluations (NFE), evaluated on 50k samples.

| Method | Model | NFE | | | | | | |
|---|---|---|---|---|---|---|---|---|
| | | 5 | 6 | 8 | 10 | 12 | 15 | 20 |
| DPM-Solver++ [32] | | 18.59 | 8.50 | 4.19 | 3.63 | 3.43 | 3.29 | 3.16 |
| UniPC [58] | Latent-Diffusion [43] | 12.24 | 6.19 | 4.00 | 3.56 | 3.34 | 3.18 | 3.07 |
| DPM-Solver-v3 | | **7.54** | **4.79** | **3.53** | **3.16** | **3.06** | **3.05** | **3.05** |

We present the detailed quantitative results of Section 4.1 for different datasets in Table 1, Table 5, Table 6 and Table 7 respectively. They clearly verify that DPM-Solver-v3 achieves consistently better or comparable performance under various settings, especially in 5∼10 NFEs.

## G Ablations

In this section, we conduct some ablations to further evaluate and analyze the effectiveness of DPM-Solver-v3.

Table 6: Quantitative results on ImageNet-256 [9]. We report the FID↓ of the methods with different numbers of function evaluations (NFE), evaluated on 10k samples.

| Method | Model | NFE | | | | | | |
|---|---|---|---|---|---|---|---|---|
| | | 5 | 6 | 8 | 10 | 12 | 15 | 20 |
| DPM-Solver++ [32] | Guided-Diffusion [10] ($s = 2.0$) | 16.87 | 13.09 | 9.95 | 8.72 | 8.13 | 7.73 | 7.48 |
| UniPC [58] | | 15.62 | 11.91 | 9.29 | 8.35 | 7.95 | 7.64 | 7.44 |
| DPM-Solver-v3 | | **15.10** | **11.39** | **8.96** | **8.27** | **7.94** | **7.62** | **7.39** |

Table 7: Quantitative results on MS-COCO2014 [26] prompts. We report the MSE↓ of the methods with different numbers of function evaluations (NFE), evaluated on 10k samples.

| Method | Model | NFE | | | | | | |
|---|---|---|---|---|---|---|---|---|
| | | 5 | 6 | 8 | 10 | 12 | 15 | 20 |
| DPM-Solver++ [32] | Stable-Diffusion [43] ($s = 1.5$) | 0.076 | 0.056 | 0.028 | 0.016 | 0.012 | 0.009 | 0.006 |
| UniPC [58] | | 0.055 | 0.039 | **0.024** | 0.012 | 0.007 | 0.005 | **0.002** |
| DPM-Solver-v3 | | **0.037** | **0.027** | **0.024** | **0.007** | **0.005** | **0.001** | **0.002** |
| DPM-Solver++ [32] | Stable-Diffusion [43] ($s = 7.5$) | 0.60 | 0.65 | 0.50 | 0.46 | **0.42** | 0.38 | 0.30 |
| UniPC [58] | | 0.65 | 0.71 | 0.56 | 0.46 | 0.43 | 0.35 | 0.31 |
| DPM-Solver-v3 | | **0.55** | **0.64** | **0.49** | **0.40** | 0.45 | **0.34** | **0.29** |

### G.1 Varying the Number of Timesteps and Datapoints for the EMS

Table 8: Ablation of the number of timesteps $N$ and datapoints $K$ for the EMS, experimented with ScoreSDE [51] on CIFAR10 [24]. We report the FID↓ with different numbers of function evaluations (NFE), evaluated on 50k samples.

| $N$ | $K$ | NFE | | | | | | |
|---|---|---|---|---|---|---|---|---|
| | | 5 | 6 | 8 | 10 | 12 | 15 | 20 |
| 1200 | 512 | 18.84 | 7.90 | 4.49 | 3.74 | 3.88 | 3.52 | 3.12 |
| 1200 | 1024 | 15.52 | 7.55 | 4.17 | 3.56 | 3.37 | 3.03 | 2.78 |
| 120 | 4096 | 13.67 | 7.60 | 4.09 | 3.49 | 3.24 | **2.90** | **2.70** |
| 250 | 4096 | 13.28 | 7.56 | 4.00 | 3.45 | **3.22** | 2.92 | **2.70** |
| 1200 | 4096 | **12.76** | **7.40** | **3.94** | **3.40** | 3.24 | 2.91 | 2.71 |

First, we'd like to investigate how the number of timesteps $N$ and the number of datapoints $K$ for computing the EMS affects the performance. We conduct experiments with the DPM ScoreSDE [51] on CIFAR10 [24], by decreasing $N$ and $K$ from our default choice $N = 1200, K = 4096$.

We list the FID results using the EMS of different $N$ and $K$ in Table 8. We can observe that the number of datapoints $K$ is crucial to the performance, while the number of timesteps $N$ is less significant and affects mainly the performance in 5∼10 NFEs. When NFE>10, we can decrease $N$ to as little as 50, which gives even better FIDs. Note that the time cost for computing the EMS is proportional to $NK$, so how to choose appropriate $N$ and $K$ for both efficiency and accuracy is worth studying.

### G.2 First-Order Comparison

As stated in Appendix A, the first-order case of DPM-Solver-v3 (*DPM-Solver-v3-1*) is different from DDIM [48], which is the previous best first-order solver for DPMs. Note that DPM-Solver-v3-1 applies no corrector, since any corrector has an order of at least 2.

In Table 9 and Figure 7, we compare DPM-Solver-v3-1 with DDIM both quantitatively and qualitatively, using the DPM ScoreSDE [51] on CIFAR10 [24]. The results verify our statement that DPM-Solver-v3-1 performs better than DDIM.

Table 9: Quantitative comparison of first-order solvers (DPM-Solver-v3-1 and DDIM [48]), experimented with ScoreSDE [51] on CIFAR10 [24]. We report the FID↓ with different numbers of function evaluations (NFE), evaluated on 50k samples.

| Method | NFE | | | | | | |
|---|---|---|---|---|---|---|---|
| | 5 | 6 | 8 | 10 | 12 | 15 | 20 |
| DDIM [48] | 54.56 | 41.92 | 27.51 | 20.11 | 15.64 | 12.05 | 9.00 |
| DPM-Solver-v3-1 | **39.18** | **29.82** | **20.03** | **14.98** | **11.86** | **9.34** | **7.19** |

Figure 7: Random samples by first-order solvers (DPM-Solver-v3-1 and DDIM [48]) of ScoreSDE [51] on CIFAR10 dataset [24], using 5, 10 and 20 NFE.

## G.3 Effects of Pseudo-Order Solver

We now demonstrate the effectiveness of the pseudo-order solver, including the pseudo-order predictor and the pseudo-order corrector.

**Pseudo-order predictor** The pseudo-order predictor is only applied in one case (at 5 NFE on CIFAR10 [24]) to achieve maximum performance improvement. In such cases, without the pseudo-order predictor, the FID results will degenerate from 12.76 to 15.91 for ScoreSDE [51], and from 12.21 to 12.72 for EDM [21]. While they are still better than previous methods, the pseudo-order predictor is proven to further boost the performance at NFEs as small as 5.

**Pseudo-order corrector** We show the comparison between true and pseudo-order corrector in Table 10. We can observe a consistent improvement when switching to the pseudo-order corrector. Thus, it suggests that if we use $n$-th order predictor, we'd better combine it with pseudo $(n + 1)$-th order corrector rather than $(n + 1)$-th order corrector.

## G.4 Effects of Half-Corrector

We demonstrate the effects of half-corrector in Table 11, using the popular Stable-Diffusion model [43]. We can observe that under the relatively large guidance scale of 7.5 which is necessary for producing samples of high quality, the corrector adopted by UniPC [58] has a negative effect on the convergence to the ground-truth samples, making UniPC even worse than DPM-Solver++ [32]. When we employ the half-corrector technique, the problem is partially alleviated. Still, it lags behind our DPM-Solver-v3, since we further incorporate the EMS.

Table 10: Effects of pseudo-order corrector under different settings. We report the FID↓ with different numbers of function evaluations (NFE).

| Method | NFE | | | | | | |
|---|---|---|---|---|---|---|---|
| | 5 | 6 | 8 | 10 | 12 | 15 | 20 |
| LSUN-Bedroom [55], Latent-Diffusion [43] | | | | | | | |
| 4th-order corrector | 8.83 | 5.28 | 3.65 | 3.27 | 3.17 | 3.14 | 3.13 |
| →pseudo (default) | **7.54** | **4.79** | **3.53** | **3.16** | **3.06** | **3.05** | **3.05** |
| ImageNet-256 [9], Guided-Diffusion [10] ($s = 2.0$) | | | | | | | |
| 3rd-order corrector | 15.87 | 11.91 | 9.27 | 8.37 | 7.97 | **7.62** | 7.47 |
| →pseudo (default) | **15.10** | **11.39** | **8.96** | **8.27** | **7.94** | **7.62** | **7.39** |
| MS-COCO2014 [26], Stable-Diffusion [43] ($s = 1.5$) | | | | | | | |
| 3rd-order corrector | **0.037** | 0.028 | 0.028 | 0.014 | 0.0078 | 0.0024 | **0.0011** |
| →pseudo (default) | **0.037** | **0.027** | **0.024** | **0.0065** | **0.0048** | **0.0014** | 0.0022 |

Table 11: Ablation of half-corrector/full-corrector on MS-COCO2014 [26] prompts with Stable-Diffusion model [43] and guidance scale 7.5. We report the MSE↓ of the methods with different numbers of function evaluations (NFE), evaluated on 10k samples.

| Method | Corrector Usage | NFE | | | | | | |
|---|---|---|---|---|---|---|---|---|
| | | 5 | 6 | 8 | 10 | 12 | 15 | 20 |
| DPM-Solver++ [32] | no corrector | 0.60 | 0.65 | 0.50 | 0.46 | 0.42 | 0.38 | 0.30 |
| UniPC [58] | full-corrector | 0.65 | 0.71 | 0.56 | 0.46 | 0.43 | 0.35 | 0.31 |
| | →half-corrector | 0.59 | 0.66 | 0.50 | 0.46 | **0.41** | 0.38 | 0.30 |
| DPM-Solver-v3 | full-corrector | 0.65 | 0.67 | **0.49** | **0.40** | 0.47 | **0.34** | 0.30 |
| | →half-corrector | **0.55** | **0.64** | 0.51 | 0.44 | 0.45 | 0.36 | **0.29** |

## G.5 Singlestep vs. Multistep

Table 12: Quantitative comparison of single-step methods (S) vs. multi-step methods (M), experimented with ScoreSDE [51] on CIFAR10 [24]. We report the FID↓ with different numbers of function evaluations (NFE), evaluated on 50k samples.

| Method | NFE | | | | | | | |
|---|---|---|---|---|---|---|---|---|
| | 5 | 6 | 8 | 10 | 12 | 15 | 20 | 25 |
| DPM-Solver (S) [31] | 290.51 | 23.78 | 23.51 | 4.67 | 4.97 | 3.34 | 2.85 | 2.70 |
| DPM-Solver (M) [32] | 27.40 | 17.85 | 9.04 | 6.41 | 5.31 | 4.10 | 3.30 | 2.98 |
| DPM-Solver++ (S) [32] | 51.80 | 38.54 | 12.13 | 6.52 | 6.36 | 4.56 | 3.52 | 3.09 |
| DPM-Solver++ (M) [32] | 28.53 | 13.48 | 5.34 | 4.01 | 4.04 | 3.32 | 2.90 | 2.76 |
| DPM-Solver-v3 (S) | 21.83 | 16.81 | 7.93 | 5.76 | 5.17 | 3.99 | 3.22 | 2.96 |
| DPM-Solver-v3 (M) | 12.76 | 7.40 | 3.94 | 3.40 | 3.24 | 2.91 | 2.71 | 2.64 |

As we stated in Section 3.2.2, multistep methods perform better than singlestep methods. To study the relationship between parameterization and these solver types, we develop a singlestep version of DPM-Solver-v3 in Algorithm 3. Note that since $(n + 1)$-th order singlestep solver divides each step into $n + 1$ substeps, the total number of timesteps is a multiple of $n + 1$. For flexibility, we follow the adaptive third-order strategy of DPM-Solver [31], which first takes third-order steps and then takes first-order or second-order steps in the end.

**Algorithm 3** $(n+1)$-th order singlestep solver

**Require:** noise prediction model $\boldsymbol{\epsilon}_\theta$, noise schedule $\alpha_t, \sigma_t$, coefficients $\boldsymbol{l}_\lambda, \boldsymbol{s}_\lambda, \boldsymbol{b}_\lambda$, cache $Q_1, Q_2$

**Input:** timesteps $\{t_i\}_{i=0}^{(n+1)M}$, initial value $\boldsymbol{x}_0$

1: $Q_1 \overset{cache}{\leftarrow} \boldsymbol{x}_0$
2: $Q_2 \overset{cache}{\leftarrow} \boldsymbol{\epsilon}_\theta(\boldsymbol{x}_0, t_0)$
3: **for** $m = 1$ **to** $M$ **do**
4:     $i_0 \leftarrow (n+1)(m-1)$
5:     **for** $i = 0$ **to** $n$ **do**
6:         $\hat{\boldsymbol{x}}_{i_0}, \ldots, \hat{\boldsymbol{x}}_{i_0+i} \overset{fetch}{\leftarrow} Q_1$
7:         $\hat{\boldsymbol{\epsilon}}_{i_0}, \ldots, \hat{\boldsymbol{\epsilon}}_{i_0+i} \overset{fetch}{\leftarrow} Q_2$
8:         $\hat{\boldsymbol{g}}_l \leftarrow e^{-\int_{\lambda_{i_0}}^{\lambda_l} \boldsymbol{s}_\tau \mathrm{d}\tau} \dfrac{\sigma_{\lambda_l}\hat{\boldsymbol{\epsilon}}_l - \boldsymbol{l}_{\lambda_l}\hat{\boldsymbol{x}}_l}{\alpha_{\lambda_l}} - \int_{\lambda_{i_0}}^{\lambda_l} e^{-\int_{\lambda_{i_0}}^{r}\boldsymbol{s}_\tau\mathrm{d}\tau} \boldsymbol{b}_r\mathrm{d}r, \quad l = i_0, \ldots, i_0 + i$ (Eq. (8))
9:         $\hat{\boldsymbol{x}}_{i_0+i+1} \leftarrow \text{LUpdate}_{i+1}(\{t_{i_0+1}, \hat{\boldsymbol{g}}_{i_0+1}\}, \ldots, \{t_{i_0+i}, \hat{\boldsymbol{g}}_{i_0+i}\}, \{t_{i_0}, \hat{\boldsymbol{x}}_{i_0}, \hat{\boldsymbol{g}}_{i_0}\}, t_{i_0+i+1})$
10:         **if** $(i+1)m \neq (n+1)M$ **then**
11:             $\hat{\boldsymbol{\epsilon}}_{i_0+i+1} \leftarrow \boldsymbol{\epsilon}_\theta(\hat{\boldsymbol{x}}_{i_0+i+1}, t_{i_0+i+1})$
12:             $Q_1 \overset{cache}{\leftarrow} \hat{\boldsymbol{x}}_{i_0+i+1}$
13:             $Q_2 \overset{cache}{\leftarrow} \hat{\boldsymbol{\epsilon}}_{i_0+i+1}$
14:         **end if**
15:     **end for**
16: **end for**

**Output:** $\hat{\boldsymbol{x}}_{(n+1)M}$

Since UniPC [58] is based on a multistep predictor-corrector framework and has no singlestep version, we only compare with DPM-Solver [31] and DPM-Solver++ [32], which uses noise prediction and data prediction respectively. The results of ScoreSDE model [51] on CIFAR10 [24] are shown in Table 12, which demonstrate that different parameterizations have different relative performance under singlestep and multistep methods.

Specifically, for singlestep methods, DPM-Solver-v3 (S) outperforms DPM-Solver++ (S) across NFEs, but DPM-Solver (S) is even better than DPM-Solver-v3 (S) when NFE≥10 (though when NFE<10, DPM-Solver (S) has worst performance), which suggests that noise prediction is best strategy in such scenarios; for multistep methods, we have DPM-Solver-v3 (M) > DPM-Solver++ (M) > DPM-Solver (M) across NFEs. Moreover, DPM-Solver (S) even outperforms DPM-Solver++ (M) when NFE≥20, and the properties of different parameterizations in singlestep methods are left for future study. Overall, DPM-Solver-v3 (M) achieves the best results among all these methods, and performs the most stably under different NFEs.

# H   FID/CLIP Score on Stable-Diffusion

Table 13: Sample quality and text-image alignment performance on MS-COCO2014 [26] prompts with Stable-Diffusion model [43] and guidance scale 7.5. We report the FID↓ and CLIP score↑ of the methods with different numbers of function evaluations (NFE), evaluated on 10k samples.

| Method | Metric | NFE | | | | | | |
|---|---|---|---|---|---|---|---|---|
| | | 5 | 6 | 8 | 10 | 12 | 15 | 20 |
| DPM-Solver++ [32] | FID | 18.87 | 17.44 | 16.40 | 15.93 | 15.78 | 15.84 | 15.72 |
| | CLIP score | 0.263 | 0.265 | 0.265 | 0.265 | 0.266 | 0.265 | 0.265 |
| UniPC [58] | FID | **18.77** | 17.32 | 16.20 | 16.15 | 16.09 | 16.06 | 15.94 |
| | CLIP score | 0.262 | 0.263 | 0.265 | 0.265 | 0.265 | 0.265 | 0.265 |
| DPM-Solver-v3 | FID | 18.83 | **16.41** | **15.41** | 15.32 | **15.13** | 15.30 | **15.23** |
| | CLIP score | 0.260 | 0.262 | 0.264 | 0.265 | 0.265 | 0.265 | 0.265 |

In text-to-image generation, since the sample quality is affected not only by discretization error of the sampling process, but also by estimation error of neural networks during training, low MSE (faster convergence) does not necessarily imply better sample quality. Therefore, we choose the MSCOCO2014 [26] validation set as the reference, and additionally evaluate DPM-Solver-v3 on Stable-Diffusion model [43] by the standard metrics FID and CLIP score [41] which measure the sample quality and text-image alignment respectively. For DPM-Solver-v3, we use the full-corrector strategy when NFE<10, and no corrector when NFE≥10.

The results in Table 13 show that DPM-Solver-v3 achieves consistently better FID and similar CLIP scores. **Notably, we achieve an FID of 15.4 in 8 NFE, close to the reported FID of Stable-Diffusion v1.4.**

Still, we claim that FID is not a proper metric for evaluating the convergence of latent-space diffusion models. As stated in DPM-Solver++ and Section 4.1, we can see that the FIDs quickly achieve 15.0∼16.0 within 10 steps, even if the latent code does not converge, because of the strong image decoder. Instead, MSE in the latent space is a direct way to measure the convergence. By comparing the MSE, our sampler does converge faster to the ground-truth samples of Stable Diffusion itself.

# I  More Theoretical Analyses

## I.1  Expressive Power of Our Generalized Parameterization

Though the introduced coefficients $l_\lambda, s_\lambda, b_\lambda$ seem limited to guarantee the optimality of the parameterization formulation itself, we claim that the generalized parameterization $g_\theta$ in Eq. (8) can actually cover a wide range of parameterization families in the form of $\psi_\theta(x_\lambda, \lambda) = \alpha(\lambda)\epsilon_\theta(x_\lambda, \lambda) + \beta(\lambda)x_\lambda + \gamma(\lambda)$. Considering the paramerization on $[\lambda_s, \lambda_t]$, by rearranging the terms, Eq. (8) can be written as

$$g_\theta(x_\lambda, \lambda) = e^{-\int_{\lambda_s}^{\lambda} s_\tau \mathrm{d}\tau} \frac{\sigma_\lambda}{\alpha_\lambda}\epsilon_\theta(x_\lambda, \lambda) - e^{-\int_{\lambda_s}^{\lambda} s_\tau \mathrm{d}\tau} \frac{l_\lambda}{\alpha_\lambda}x_\lambda - \int_{\lambda_s}^{\lambda} e^{-\int_{\lambda_s}^{r} s_\tau \mathrm{d}\tau} b_r \mathrm{d}r \qquad (68)$$

We can compare the coefficients before $\epsilon_\theta$ and $x_\lambda$ in $g_\theta$ and $\psi_\theta$ to figure out how $l_\lambda, s_\lambda, b_\lambda$ corresponds to $\alpha(\lambda), \beta(\lambda), \gamma(\lambda)$. In fact, we can not directly let $g_\theta$ equal $\psi_\theta$, since when $\lambda = \lambda_s$, we have $\int_{\lambda_s}^{\lambda}(\cdot) = 0$, and the coefficient before $\epsilon_\theta$ in $g_\theta$ is fixed. Still, $\psi_\theta$ can be equalized to $g_\theta$ by a linear transformation, which only depends on $\lambda_s$ and does not affect our analyses of the discretization error and solver.

Specifically, assuming $\psi_\theta = \omega_{\lambda_s} g_\theta + \xi_{\lambda_s}$, by corresponding the coefficients we have

$$\begin{cases} \alpha(\lambda) = \omega_{\lambda_s} e^{-\int_{\lambda_s}^{\lambda} s_\tau \mathrm{d}\tau} \frac{\sigma_\lambda}{\alpha_\lambda} \\ \beta(\lambda) = -\omega_{\lambda_s} e^{-\int_{\lambda_s}^{\lambda} s_\tau \mathrm{d}\tau} \frac{l_\lambda}{\alpha_\lambda} \\ \gamma(\lambda) = -\omega_{\lambda_s} \int_{\lambda_s}^{\lambda} e^{-\int_{\lambda_s}^{r} s_\tau \mathrm{d}\tau} b_r \mathrm{d}r + \xi_{\lambda_s} \end{cases} \Rightarrow \begin{cases} \omega_{\lambda_s} = e^{\lambda_s}\alpha(\lambda_s) \\ \xi_{\lambda_s} = \gamma(\lambda_s) \\ l_\lambda = -\sigma_\lambda \frac{\beta(\lambda)}{\alpha(\lambda)} \\ s_\lambda = -1 - \frac{\alpha'(\lambda)}{\alpha(\lambda)} \\ b_\lambda = -e^{-\lambda} \frac{\gamma'(\lambda)}{\alpha(\lambda)} \end{cases} \qquad (69)$$

Therefore, as long as $\alpha(\lambda) \neq 0$ and $\alpha(\lambda), \gamma(\lambda)$ have first-order derivatives, our proposed parameterization $g_\theta$ holds the same expressive power as $\psi_\theta$, while at the same time enabling the neat optimality criteria of $l_\lambda, s_\lambda, b_\lambda$ in Eq. (5) and Eq. (11).

## I.2  Justification of Why Minimizing First-order Discretization Error Can Help Higher-order Solver

The EMS $s_\lambda^*, b_\lambda^*$ in Eq. (11) are designed to minimize the first-order discretization error in Eq. (10). However, the high-order solver is actually more frequently adopted in practice (specifically, third-order in unconditional sampling, second-order in conditional sampling), since it incurs lower sampling errors by taking higher-order Taylor expansions to approximate the predictor $g_\theta$.

In the following, we show that the EMS can also help high-order solver. By Eq. (52) in Appendix B.4, the $(n + 1)$-th order local error can be expressed as

$$
\begin{aligned}
\hat{\boldsymbol{x}}_t - \boldsymbol{x}_t = {} & \alpha_t \boldsymbol{A}(\lambda_s, \lambda_t) \int_{\lambda_s}^{\lambda_t} \boldsymbol{E}_{\lambda_s}(\lambda) \left( \boldsymbol{g}_\theta(\boldsymbol{x}_\lambda, \lambda) - \boldsymbol{g}_\theta(\boldsymbol{x}_{\lambda_s}, \lambda_s) \right) \mathrm{d}\lambda \\
& - \alpha_t \boldsymbol{A}(\lambda_s, \lambda_t) \sum_{k=1}^{n} \left( \sum_{l=1}^{n} (\boldsymbol{R}_n^{-1})_{kl} (\boldsymbol{g}_\theta(\boldsymbol{x}_{\lambda_{i_l}}, \lambda_{i_l}) - \boldsymbol{g}_\theta(\boldsymbol{x}_{\lambda_s}, \lambda_s)) \right) \int_{\lambda_s}^{\lambda_t} \boldsymbol{E}_{\lambda_s}(\lambda) \frac{(\lambda - \lambda_s)^k}{k!} \mathrm{d}\lambda
\end{aligned}
\tag{70}
$$

By Newton-Leibniz theorem, it is equivalent to

$$
\begin{aligned}
\hat{\boldsymbol{x}}_t - \boldsymbol{x}_t = {} & \alpha_t \boldsymbol{A}(\lambda_s, \lambda_t) \int_{\lambda_s}^{\lambda_t} \boldsymbol{E}_{\lambda_s}(\lambda) \left( \int_{\lambda_s}^{\lambda} g_\theta^{(1)}(x_\tau, \tau) \mathrm{d}\tau \right) \mathrm{d}\lambda \\
& - \alpha_t \boldsymbol{A}(\lambda_s, \lambda_t) \sum_{k=1}^{n} \left( \sum_{l=1}^{n} (\boldsymbol{R}_n^{-1})_{kl} \int_{\lambda_s}^{\lambda_{i_l}} g_\theta^{(1)}(x_\lambda, \lambda) \mathrm{d}\lambda \right) \int_{\lambda_s}^{\lambda_t} \boldsymbol{E}_{\lambda_s}(\lambda) \frac{(\lambda - \lambda_s)^k}{k!} \mathrm{d}\lambda
\end{aligned}
\tag{71}
$$

We assume that the estimated EMS are bounded (in the order of $\mathcal{O}(1)$, Assumption B.2 in Appendix B.1), which is empirically confirmed as in Section 4.2. By the definition of $\boldsymbol{g}_\theta$ in Eq. (8), we have $\boldsymbol{g}_\theta^{(1)}(\boldsymbol{x}_\tau, \tau) = e^{-\int_{\lambda_s}^{\tau} \boldsymbol{s}_r \mathrm{d}r} \left( \boldsymbol{f}_\theta^{(1)}(\boldsymbol{x}_\tau, \tau) - \boldsymbol{s}_\tau \boldsymbol{f}_\theta(\boldsymbol{x}_\tau, \tau) - \boldsymbol{b}_\tau \right)$. Therefore, Eq. (11) controls $\|\boldsymbol{g}_\theta^{(1)}\|_2$ and further controls $\|\hat{\boldsymbol{x}}_t - \boldsymbol{x}_t\|_2$, since other terms are only dependent on the EMS and are bounded.

### I.3 The Extra Error of EMS Estimation and Integral Estimation

**Analysis of EMS estimation error**   In practice, the EMS in Eq. (5) and Eq. (11) are estimated on finite datapoints by the explicit expressions in Eq. (58) and Eq. (63), which may differ from the true $\boldsymbol{l}_\lambda^*, \boldsymbol{s}_\lambda^*, \boldsymbol{b}_\lambda^*$. Theoretically, on one hand, the order and convergence theorems in Section 3.2 are irrelevant to the EMS estimation error: The ODE solution in Eq. (9) is correct whatever $\boldsymbol{l}_\lambda, \boldsymbol{s}_\lambda, \boldsymbol{b}_\lambda$ are, and we only need the assumption that these coefficients are bounded (Assumption B.2 in Appendix B.1) to prove the local and global order; on the other hand, the first-order discretization error in Eq. (10) is vulnerable to the EMS estimation error, which relates to the performance at few steps. Empirically, to enable fast sampling, we need to ensure the number of datapoints for estimating EMS (see ablations in Table 8), and we find that our method is robust to the EMS estimation error given only 1024 datapoints in most cases (see EMS computing configs in Appendix D).

**Analysis of integral estimation error**   Another source of error is the process of integral estimation ($\int_{\lambda_s}^{\lambda_t} \boldsymbol{E}_{\lambda_s}(\lambda) \boldsymbol{B}_{\lambda_s}(\lambda) \mathrm{d}\lambda$ and $\int_{\lambda_s}^{\lambda_t} \boldsymbol{E}_{\lambda_s}(\lambda) \frac{(\lambda - \lambda_s)^k}{k!} \mathrm{d}\lambda$ in Eq. (14)) by trapezoidal rule. We can analyze the estimation error by the error bound formula of trapezoidal rule: suppose we use uniform discretization on $[a, b]$ with interval $h$ to estimate $\int_a^b f(x) \mathrm{d}x$, then the error $E$ satisfies

$$
|E| \leq \frac{(b-a)h^2}{12} \max |f''(x)|
\tag{72}
$$

Under Assumption B.2, the EMS and their first-order derivative are bounded. Denote $\boldsymbol{f}_1(\lambda) = \boldsymbol{E}_{\lambda_s}(\lambda)\boldsymbol{B}_{\lambda_s}(\lambda)$, $\boldsymbol{f}_2(\lambda) = \boldsymbol{E}_{\lambda_s}(\lambda)\frac{(\lambda-\lambda_s)^k}{k!}$, then

$$\boldsymbol{f}_1(\lambda) = e^{\int_{\lambda_s}^{\lambda}(\boldsymbol{l}_\tau+\boldsymbol{s}_\tau)\mathrm{d}\tau}\int_{\lambda_s}^{\lambda}e^{-\int_{\lambda_s}^{r}\boldsymbol{s}_\tau\mathrm{d}\tau}\boldsymbol{b}_r\mathrm{d}r$$

$$\boldsymbol{f}_1'(\lambda) = (\boldsymbol{l}_\lambda+\boldsymbol{s}_\lambda)e^{\int_{\lambda_s}^{\lambda}(\boldsymbol{l}_\tau+\boldsymbol{s}_\tau)\mathrm{d}\tau}\int_{\lambda_s}^{\lambda}e^{-\int_{\lambda_s}^{r}\boldsymbol{s}_\tau\mathrm{d}\tau}\boldsymbol{b}_r\mathrm{d}r + \boldsymbol{b}_\lambda e^{\int_{\lambda_s}^{\lambda}\boldsymbol{l}_\tau\mathrm{d}\tau}\int_{\lambda_s}^{\lambda}e^{-\int_{\lambda_s}^{r}\boldsymbol{s}_\tau\mathrm{d}\tau}\boldsymbol{b}_r\mathrm{d}r$$

$$\boldsymbol{f}_1''(\lambda) = (\boldsymbol{l}_\lambda'+\boldsymbol{s}_\lambda')e^{\int_{\lambda_s}^{\lambda}(\boldsymbol{l}_\tau+\boldsymbol{s}_\tau)\mathrm{d}\tau}\int_{\lambda_s}^{\lambda}e^{-\int_{\lambda_s}^{r}\boldsymbol{s}_\tau\mathrm{d}\tau}\boldsymbol{b}_r\mathrm{d}r + (\boldsymbol{l}_\lambda+\boldsymbol{s}_\lambda)^2 e^{\int_{\lambda_s}^{\lambda}(\boldsymbol{l}_\tau+\boldsymbol{s}_\tau)\mathrm{d}\tau}\int_{\lambda_s}^{\lambda}e^{-\int_{\lambda_s}^{r}\boldsymbol{s}_\tau\mathrm{d}\tau}\boldsymbol{b}_r\mathrm{d}r$$

$$+ (\boldsymbol{l}_\lambda+\boldsymbol{s}_\lambda)\boldsymbol{b}_\lambda e^{\int_{\lambda_s}^{\lambda}\boldsymbol{l}_\tau\mathrm{d}\tau}\int_{\lambda_s}^{\lambda}e^{-\int_{\lambda_s}^{r}\boldsymbol{s}_\tau\mathrm{d}\tau}\boldsymbol{b}_r\mathrm{d}r + \boldsymbol{b}_\lambda'e^{\int_{\lambda_s}^{\lambda}\boldsymbol{l}_\tau\mathrm{d}\tau}\int_{\lambda_s}^{\lambda}e^{-\int_{\lambda_s}^{r}\boldsymbol{s}_\tau\mathrm{d}\tau}\boldsymbol{b}_r\mathrm{d}r$$

$$+ \boldsymbol{b}_\lambda\boldsymbol{l}_\lambda e^{\int_{\lambda_s}^{\lambda}\boldsymbol{l}_\tau\mathrm{d}\tau}\int_{\lambda_s}^{\lambda}e^{-\int_{\lambda_s}^{r}\boldsymbol{s}_\tau\mathrm{d}\tau}\boldsymbol{b}_r\mathrm{d}r + \boldsymbol{b}_\lambda^2 e^{\int_{\lambda_s}^{\lambda}(\boldsymbol{l}_\tau-\boldsymbol{s}_\tau)\mathrm{d}\tau}\int_{\lambda_s}^{\lambda}e^{-\int_{\lambda_s}^{r}\boldsymbol{s}_\tau\mathrm{d}\tau}\boldsymbol{b}_r\mathrm{d}r$$

$$(73)$$

$$\boldsymbol{f}_2(\lambda) = e^{\int_{\lambda_s}^{\lambda}(\boldsymbol{l}_\tau+\boldsymbol{s}_\tau)\mathrm{d}\tau}\frac{(\lambda-\lambda_s)^k}{k!}$$

$$\boldsymbol{f}_2'(\lambda) = (\boldsymbol{l}_\lambda+\boldsymbol{s}_\lambda)e^{\int_{\lambda_s}^{\lambda}(\boldsymbol{l}_\tau+\boldsymbol{s}_\tau)\mathrm{d}\tau}\frac{(\lambda-\lambda_s)^k}{k!} + e^{\int_{\lambda_s}^{\lambda}(\boldsymbol{l}_\tau+\boldsymbol{s}_\tau)\mathrm{d}\tau}\frac{(\lambda-\lambda_s)^{k-1}}{(k-1)!}$$

$$\boldsymbol{f}_2''(\lambda) = (\boldsymbol{l}_\lambda'+\boldsymbol{s}_\lambda')e^{\int_{\lambda_s}^{\lambda}(\boldsymbol{l}_\tau+\boldsymbol{s}_\tau)\mathrm{d}\tau}\frac{(\lambda-\lambda_s)^k}{k!} + (\boldsymbol{l}_\lambda+\boldsymbol{s}_\lambda)^2 e^{\int_{\lambda_s}^{\lambda}(\boldsymbol{l}_\tau+\boldsymbol{s}_\tau)\mathrm{d}\tau}\frac{(\lambda-\lambda_s)^k}{k!} \qquad (74)$$

$$+ (\boldsymbol{l}_\lambda+\boldsymbol{s}_\lambda)e^{\int_{\lambda_s}^{\lambda}(\boldsymbol{l}_\tau+\boldsymbol{s}_\tau)\mathrm{d}\tau}\frac{(\lambda-\lambda_s)^{k-1}}{k-1!} + (\boldsymbol{l}_\lambda+\boldsymbol{s}_\lambda)e^{\int_{\lambda_s}^{\lambda}(\boldsymbol{l}_\tau+\boldsymbol{s}_\tau)\mathrm{d}\tau}\frac{(\lambda-\lambda_s)^{k-1}}{(k-1)!}$$

$$+ e^{\int_{\lambda_s}^{\lambda}(\boldsymbol{l}_\tau+\boldsymbol{s}_\tau)\mathrm{d}\tau}\frac{(\lambda-\lambda_s)^{k-2}}{(k-2)!}$$

Since $\boldsymbol{l}_\lambda, \boldsymbol{s}_\lambda, \boldsymbol{b}_\lambda, \boldsymbol{l}_\lambda', \boldsymbol{s}_\lambda', \boldsymbol{b}_\lambda'$ are all $\mathcal{O}(1)$, we can conclude that $\boldsymbol{f}_1''(\lambda) = \mathcal{O}(h)$, $\boldsymbol{f}_2''(\lambda) = \mathcal{O}(h^{k-2})$, and the errors of $\int_{\lambda_s}^{\lambda_t}\boldsymbol{E}_{\lambda_s}(\lambda)\boldsymbol{B}_{\lambda_s}(\lambda)\mathrm{d}\lambda$, $\int_{\lambda_s}^{\lambda_t}\boldsymbol{E}_{\lambda_s}(\lambda)\frac{(\lambda-\lambda_s)^k}{k!}\mathrm{d}\lambda$ are $O(h_0^2 h^2), O(h_0^2 h^{k-1})$ respectively, where $h_0$ is the stepsize of EMS discretization ($h_0 = \frac{\lambda_t-\lambda_s}{n}$, $n$ corresponds to 120∼1200 timesteps for our EMS computing), and $h = \lambda_t - \lambda_s$. Therefore, the extra error of integral estimation is under high order and ignorable.

## J   More Discussions

### J.1   Extra Computational and Memory Costs

The extra memory cost of DPM-Solver-v3 is rather small. The extra coefficients $\boldsymbol{l}_\lambda, \boldsymbol{s}_\lambda, \boldsymbol{b}_\lambda$ are discretized and computed at $N$ timesteps, each with a dimension $D$ same as the diffused data. The extra memory cost is $\mathcal{O}(ND)$, including the precomputed terms in Appendix C.1.2, and is rather small compared to the pretrained model (e.g. only ∼125M in total on Stable-Diffusion, compared to ∼4G of the model itself).

The pre-computation time for estimating EMS is rather short. The EMS introduced by our method can be effectively estimated on around 1k datapoints within hours (Appendix D), which is rather short compared to the long training/distillation time of other methods. Moreover, the integrals of these extra coefficients are just some vector constants that can be pre-computed within seconds, as shown in Appendix C.1.2. The precomputing is done only once before sampling.

The extra computational overhead of DPM-Solver-v3 during sampling is negligible. Once we obtain the estimated EMS and their integrals at discrete timesteps, they can be regarded as constants. Thus, during the subsequent sampling process, the computational overhead is the same as previous training-free methods (such as DPM-Solver++) with negligible differences (Appendix E).

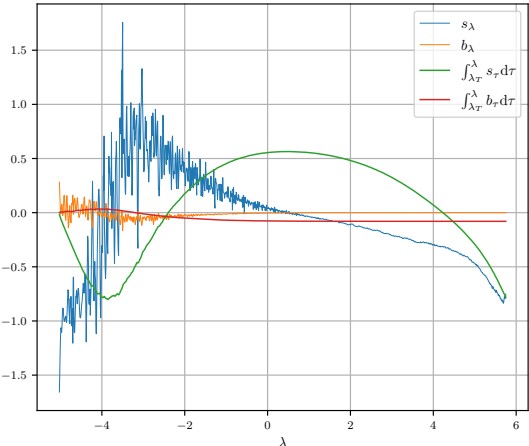

Figure 8: Visualization of the EMS $s_\lambda, b_\lambda$ and their integrals w.r.t. $\lambda$, estimated on ScoreSDE [51] on CIFAR10 [24]. $s_\lambda, b_\lambda$ are rather fluctuating, but their integrals are smooth enough to ensure the stability of DPM-Solver-v3.

## J.2 Flexibility

The pre-computed EMS can be applied for any time schedule during sampling without re-computing EMS. Besides, we compute EMS on unconditional models and it can be used for a wide range of guidance scales (such as cfg=7.5 in Stable Diffusion). In short, EMS is flexible and easy to adopt in downstream applications.

- Time schedule: The choice of $N, K$ in EMS is disentangled with the timestep scheduler in sampling. Once we have estimated the EMS at $N$ (e.g., 1200) timesteps, they can be flexibly adapted to any schedule (uniform $\lambda$/uniform $t$...) during sampling, by corresponding the actual timesteps during sampling to the $N$ bins. For different time schedule, we only need to re-precompute $E_{\lambda_s,\lambda_t}^{(k)}$ in Appendix C.1.2, and the time cost is within seconds.

- Guided sampling: We compute the EMS on the unconditional model for all guided cases. Empirically, the EMS computed on the model without guidance (unconditional part) performs more stably than those computed on the model with guidance, and can accelerate the sampling procedure in a wide range of guidance scales (including the common guidance scales used in pretrained models). We think that the unconditional model contains some common information (image priors) for all the conditions, such as color, sketch, and other image patterns. Extracting them helps correct some common biases such as shallow color, low saturation level and lack of details. In contrast, the conditional model is dependent on the condition and has a large variance.

*Remark* J.1. EMS computed on models without guidance cannot work for extremely large guidance scales (e.g., cfg scale 15 for Stable Diffusion), since in this case, the condition has a large impact on the denoising process. Note that at these extremely large scales, the sampling quality is very low (compared to cfg scale 7.5) and they are rarely used in practice. Therefore, our proposed EMS without guidance is suitable enough for the common applications with the common guidance.

## J.3 Stability

As shown in Section 4.2, the estimated EMS $s_\lambda, b_\lambda$ appear much fluctuating, especially for ScoreSDE on CIFAR10. We would like to clarify that the unstable $s_\lambda, b_\lambda$ is not an issue, and our sampler is stable:

- The fluctuation of $s_\lambda, b_\lambda$ on ScoreSDE is intrinsic and not due to the estimation error. As we increase the number of samples to decrease the estimation error, the fluctuation is not reduced. We attribute it to the periodicity of trigonometric functions in the positional timestep embedding as stated in Section 4.2.

- Moreover, we only need to consider the integrals of $s_\lambda, b_\lambda$ in the ODE solution Eq. (9). As shown in Figure 8, the integrals of $s_\lambda, b_\lambda$ are rather smooth, which ensures the stability of our method. Therefore, there is no need for extra smoothing.

## J.4 Practical Value

When NFE is around 20, our improvement of sample quality is small because all different fast samplers based on diffusion ODEs almost converge. Therefore, what matters is that our method has a faster convergence speed to good sample quality. As we observed, the less diverse the domain is, the more evidently the speed-up takes effect. For example, on LSUN-Bedroom, our method can achieve up to 40% faster convergence.

To sum up, the practical value of DPM-Solver-v3 embodies the following aspects:

1. 15∼30% speed-up can save lots of costs for online text-to-image applications. Our speed-ups are applicable to large text-to-image diffusion models, which are an important part of today's AIGC community. As the recent models become larger, a single NFE requires more computational resources, and 15∼30% speed-up can save a lot of the companies' expenses for commercial usage.

2. Improvement in 5∼10 NFEs benefits image previewing. Since the samples are controlled by the random seed, coarse samples with 5∼10 NFEs can be used to *preview* thousands of samples with low costs and give guidance on choosing the random seed, which can be then used to generate fine samples with best-quality sampling strategies. This is especially useful for text-to-image generation. Since our method achieves better quality and converges faster to the ground-truth sample, it can provide better guidance when used for preview.

## K  Additional Samples

We provide more visual samples in Figure 9, Figure 10, Figure 11 and Table 14 to demonstrate the qualitative effectiveness of DPM-Solver-v3. It can be seen that the visual quality of DPM-Solver-v3 outperforms previous state-of-the-art solvers. Our method can generate images that have reduced bias (less "shallow"), higher saturation level and more visual details, as mentioned in Section 4.3.

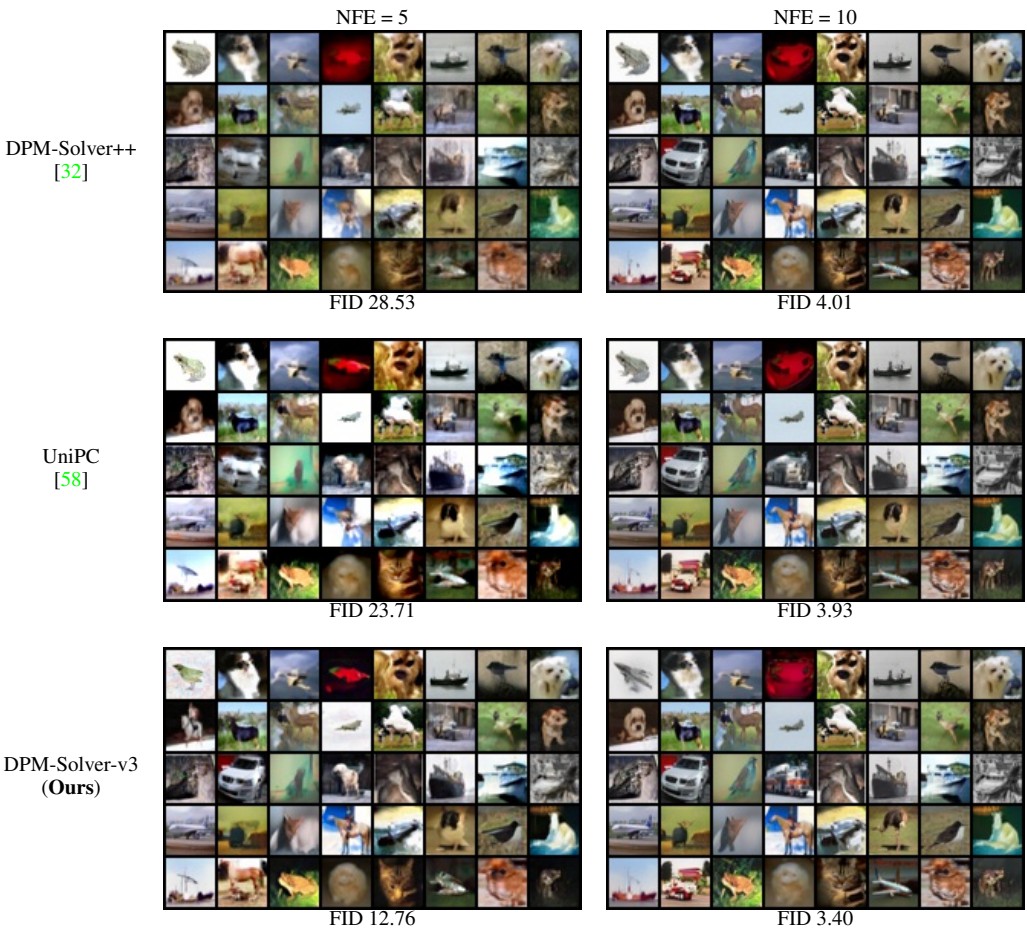

Figure 9: Random samples of ScoreSDE [51] on CIFAR10 dataset [24] with only 5 and 10 NFE.

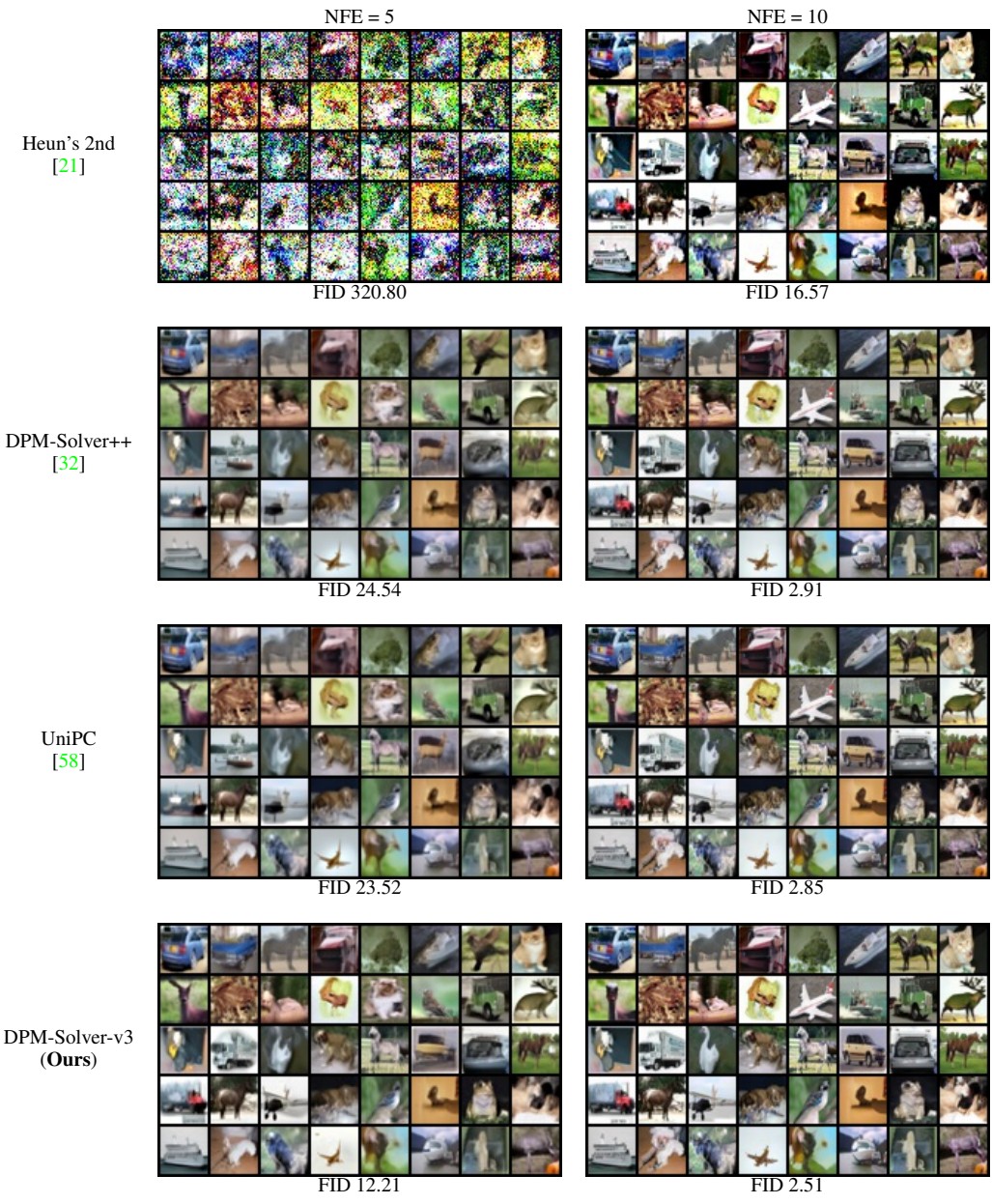

Figure 10: Random samples of EDM [21] on CIFAR10 dataset [24] with only 5 and 10 NFE.

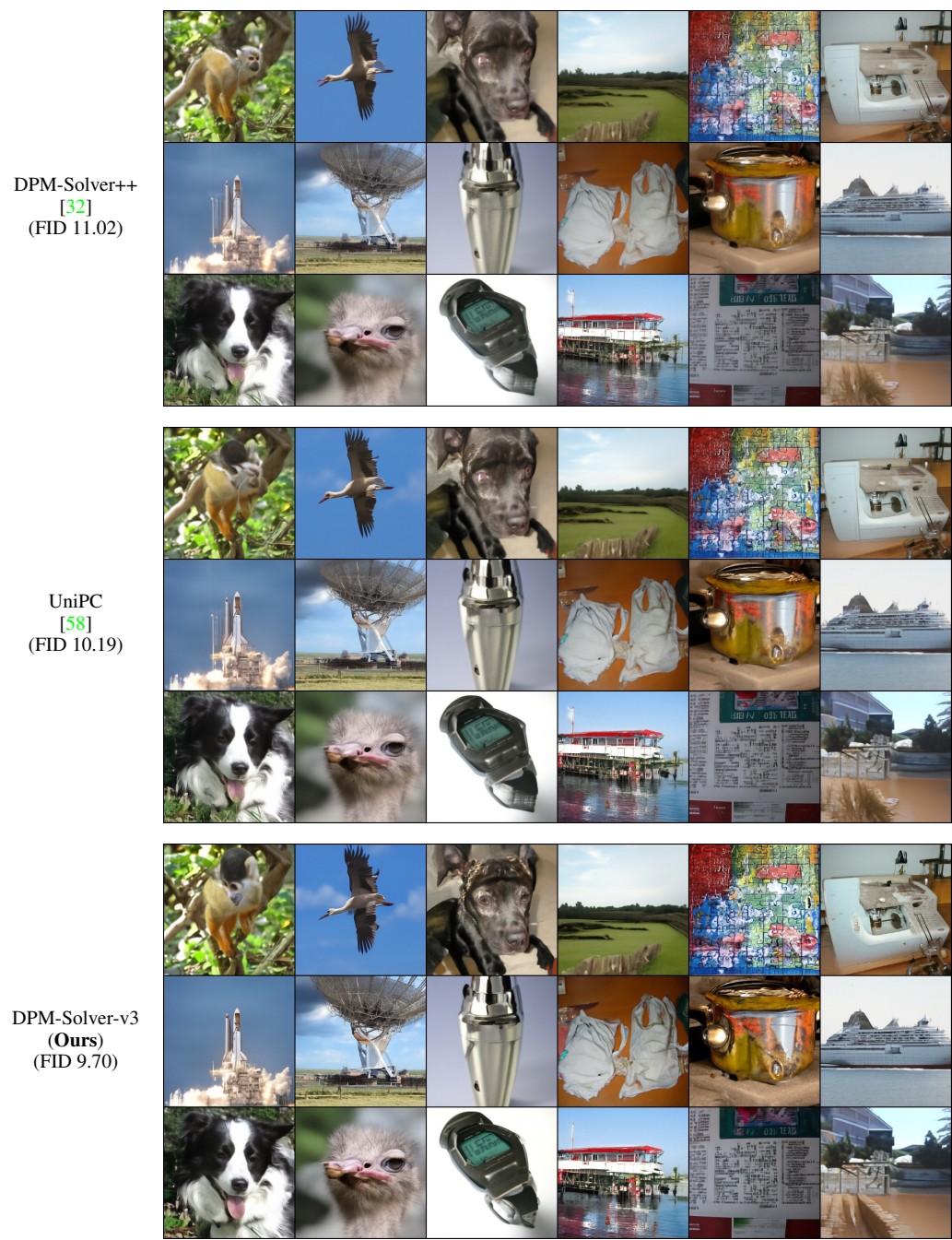

DPM-Solver++
[32]
(FID 11.02)

UniPC
[58]
(FID 10.19)

DPM-Solver-v3
(**Ours**)
(FID 9.70)

Figure 11: Random samples of Guided-Diffusion [10] on ImageNet-256 dataset [9] with a classifier guidance scale 2.0, using only 7 NFE. We manually remove the potentially disturbing images such as those containing snakes or insects.

Table 14: Additional samples of Stable-Diffusion [43] with a classifier-free guidance scale 7.5, using only 5 NFE and selected text prompts. Some displayed prompts are truncated.

| Text Prompts | DPM-Solver++ [32] (MSE 0.60) | UniPC [58] (MSE 0.65) | DPM-Solver-v3 (**Ours**) (MSE 0.55) |
|---|---|---|---|
| *"pixar movie still portrait photo of madison beer, jessica alba, woman, as hero catgirl cyborg woman by pixar, by greg rutkowski, wlop, rossdraws, artgerm, weta, marvel, rave girl, leeloo, unreal engine, glossy skin, pearlescent, wet, bright morning, anime, sci-fi, maxim magazine cover"* |  |  |  |
| *"oil painting with heavy impasto of a pirate ship and its captain, cosmic horror painting, elegant intricate artstation concept art by craig mullins detailed"* |  |  |  |
| *"environment living room interior, mid century modern, indoor garden with fountain, retro, m vintage, designer furniture made of wood and plastic, concrete table, wood walls, indoor potted tree, large window, outdoor forest landscape, beautiful sunset, cinematic, concept art, sunstainable architecture, octane render, utopia, ethereal, cinematic light"* |  |  |  |
| *"the living room of a cozy wooden house with a fireplace, at night, interior design, concept art, wallpaper, warm, digital art. art by james gurney and larry elmore."* |  |  |  |
| *"Full page concept design how to craft life Poison, intricate details, infographic of alchemical, diagram of how to make potions, captions, directions, ingredients, drawing, magic, wuxia"* |  |  |  |
| *"Fantasy art, octane render, 16k, 8k, cinema 4d, back-lit, caustics, clean environment, Wood pavilion architecture, warm led lighting, dusk, Landscape, snow, arctic, with aqua water, silver Guggenheim museum spire, with rays of sunshine, white fabric landscape, tall building, zaha hadid and Santiago calatrava, smooth landscape, cracked ice, igloo, warm lighting, aurora borialis, 3d cgi, high definition, natural lighting, realistic, hyper realism"* |  |  |  |
| *"tree house in the forest, atmospheric, hyper realistic, epic composition, cinematic, landscape vista photography by Carr Clifton & Galen Rowell, 16K resolution, Landscape veduta photo by Dustin Lefevre & tdraw, detailed landscape painting by Ivan Shishkin, DeviantArt, Flickr, rendered in Enscape, Miyazaki, Nausicaa Ghibli, Breath of The Wild, 4k detailed post processing, artstation, unreal engine"* |  |  |  |
| *"A trail through the unknown, atmospheric, hyper realistic, 8k, epic composition, cinematic, octane render, artstation landscape vista photography by Carr Clifton & Galen Rowell, 16K resolution, Landscape veduta photo by Dustin Lefevre & tdraw, 8k resolution, detailed landscape painting by Ivan Shishkin, DeviantArt, Flickr, rendered in Enscape, Miyazaki, Nausicaa Ghibli, Breath of The Wild, 4k detailed post processing, artstation, rendering by octane, unreal engine"* |  |  |  |
| *"postapocalyptic city turned to fractal glass, ctane render, 8 k, exploration, cinematic, trending on artstation, by beeple, realistic, 3 5 mm camera, unreal engine, hyper detailed, photo–realistic maximum detai, volumetric light, moody cinematic epic concept art, realistic matte painting, hyper photorealistic, concept art, cinematic epic, octane render, 8k, corona render, movie concept art, octane render, 8 k, corona render, trending on artstation, cinematic composition, ultra–detailed, hyper–realistic, volumetric lighting"* |  |  |  |
| *""WORLDS": zoological fantasy ecosystem infographics, magazine layout with typography, annotations, in the style of Elena Masci, Studio Ghibli, Caspar David Friedrich, Daniel Merriam, Doug Chiang, Ivan Aivazovsky, Herbert Bauer, Edward Tufte, David McCandless"* |  |  |  |

