# OpenReview forum: "DPM-Solver-v3: Improved Diffusion ODE Solver with Empirical Model Statistics"
_NeurIPS.cc/2023/Conference — NeurIPS 2023 poster_

### Official Review · Reviewer_wxE7 · 2023-07-04

**Soundness:** 3 good
**Presentation:** 2 fair
**Contribution:** 2 fair
**Rating:** 5
**Confidence:** 4

**Summary:**

The paper proposes a new sampling algorithm, DPM-Solver-v3, to solve the diffusion ODE. Applying the Rosenbrock-type exponential integrators, DPM-Solver-v3 pre-computes several coefficients (empirical model statistics) to minimize the norm of the gradient of non-linear term, which leads to reduced discretization error. Empirical results of DPM-Solver-v3 demonstrate its effectiveness in unconditional/conditional sampling and classifier-free guidance on Stable Diffusion.

**Strengths:**

- By introducing empirical model statistics arising from Rosenbrock-type exponential integrators, DPM-Solver-v3 has better model parametrization than data/noise prediction ones, which leads to lower discretization error.

- Experiments show that DPM-Solver-v3 outperforms DPM-Solver++ and UniPC for both conditional and unconditional sampling. DPM-Solver-v3 also incurs a smaller MSE on Stable Diffusion, across NFEs.

**Weaknesses:**

- The sampling algorithm is based on the pre-computed empirical model statistics $l_{\lambda}, s_{\lambda}, b_{\lambda}$ at each time step $\lambda$, and their corresponding estimated integrals. This seems to lead to extra memory cost and makes the model not able to be flexibly adapted to new use cases (e.g., a different time schedule, guided sampling, etc.)

- When developing the error order and convergence of the algorithm in section 3.2, the estimation error of EMS is not taken into account. However, the unstable $s_{\lambda}$ w.r.t $\lambda$ in Figure 5 (especially for the one estimated on ScoreSDE) indicates that the EMS estimation error is not ignorable. Is it possible to provide such analysis or guarantees given the error of EMS?

- In the low FID region (which is closer to practical usage), the benefit of DPM-Solver-v3 seems not very significant compared with UniPC in Figure 3 and Figure 4

- Applying Rosenbrock-type exponential integrators to diffusion ODE is a valid contribution, but other components (e.g. higher-order derivative estimation) of the method seem to be quite relevant to those proposed by existing works like the RK45 Butcher tableau or [1,2]. The real improvement of the Rosenbrock-type technique is unclear when compounded with many other components. Could the authors maybe ablate on its individual effect?

- I think it would be very helpful to compare the FID score / CLIP score on Stable-Diffusion since low MSE does not necessarily imply better sample quality (the quality is affected not only by discretization error but also by estimation error of neural networks).

- Could the authors also compares DPM-Solver-V3 with some typical stochastic samplers, like Alg 2 in EDM, Gonna Go Fast [3], or more recent works on stochastic samplers? Is it possible to integrate the solver in these stochastic samplers, as these samplers seem to provide significantly better sample quality compared to their deterministic counterparts on complex datasets?


[1] Fast Sampling of Diffusion Models with Exponential Integrator, Zhang et al., ICML 2023

[2] DPM-Solver: A Fast ODE Solver for Diffusion Probabilistic Model Sampling in Around 10 Steps, Lu et al., NeurIPS 2022

[3] Gotta go fast when generating data with score-based models.  A. Jolicoeur-Martineau, K. Li, R. Piché-Taillefer, T. Kachman, and I. Mitliagkas. CoRR, abs/2105.14080, 2021.

**Questions:**

- The author mentioned that EMS computed on the model without guidance can work for guided sampling cases within a common range of guidance scales. I'm wondering how large is the performance gap between algorithms using EMS computed on the model with guidance and the model without guidance. How is this affected by different magnitudes of guidance scales?

- Theoretically and empirically, is the model robust to estimation error of EMS?

- How is DPM-Solver-v3's performance affected by single-step versus multistep sampling, in comparison with the baselines?

- The estimated $s_{\lambda}$ seems to be unstable w.r.t $\lambda$ in some cases. Would using moving average or other smoothing methods help with the sampling algorithm?

- I think it would be very helpful to compare the FID score / CLIP score on Stable-Diffusion since low MSE does not necessarily imply better sample quality (the quality is affected not only by discretization error but also by estimation error of neural networks).

- How well is DPM-Solver-v3 compatible with current SDE samplers?

**Limitations:**

yes

---

> ### Author Rebuttal · Authors · 2023-08-09
>
> Thank you for the interest and acknowledgment of our theoretical and empirical contributions. Below are our explanations to the questions, which we hope may clarify some misunderstandings. We kindly request that you consider raising the score accordingly if you are satisfied.
>
> *W1: The method needs extra memory cost and makes the model not able to be flexibly adapted to new use cases (e.g., a different time schedule, guided sampling, etc.)*
>
> **A**: **The extra memory cost is rather small**. Please refer to *common response, Q1*; And **the EMS are flexible for your mentioned cases**. Please refer to *common response, Q2*.
>
> *W2: In the low FID region, the benefit of DPM-Solver-v3 seems not very significant compared with UniPC in Figure 3 and Figure 4.*
>
> **A**: When NFE is around 20, our improvement of sample quality is small because all different fast samplers almost converge. However, what matters is that our method has a **faster convergence speed** to good sample quality. A notable example is that, on LSUN-Bedroom, we reach the FID of 3.06 with 12 NFE, while the previous best method requires 20 NFE, which means our computation cost is approximately 60%.
>
> *W3: Could the authors ablate on the individual effect of Rosenbrock-type exponential integrators?*
>
> **A**: Other components (e.g. higher-order derivative estimation) are just what we formulate to adapt high-order solvers to our new parameterization. Therefore, the EMS estimation itself is what we are different from existing works. If we set $l_\lambda,s_\lambda,b_\lambda$ to special values, our solvers degenerate to previous ones such as DPM-Solver and DPM-Solver++ (Appendix A.2). When we compare our method to DPM-Solver++/UniPC, we are already ablating on the EMS' effect.
>
> *Q1: The performance gap between EMS computed with guidance and without guidance? How is this affected by different magnitudes of guidance scales?*
>
> **A**: Empirically, the EMS computed on the model without guidance (unconditional part) performs more stably than those computed on the model with guidance. See *common response, Q2* for details.
>
> *Q2: Theoretically and empirically, is the model robust to estimation error of EMS?*
>
> **A**: Theoretically, **the order and convergence theorems are irrelevant to the EMS estimation error**. The ODE solution Eq. (9) is correct whatever $l_\lambda,s_\lambda,b_\lambda$ are, and we only need the assumption that these coefficients are bounded (Assumption B.2 in Appendix B.1.1) to prove the local and global order; Empirically, we need to ensure the amount of datapoints for estimating EMS (see examples in Appendix G.1), and we find that our method is robust to the estimation error of EMS **given only 1024 datapoints**.
>
> *Q3: Single-step vs multistep sampling?*
>
> **A**: We implement the singlestep DPM-Solver-v3 and compare it with the singlestep version of DPM-Solver++, since UniPC only has multistep version. We report the FID on CIFAR10 (ScoreSDE). See *response pdf, Table 1* for details.
>
> The results show that DPMv3 (singlestep) is better than DPM++ (singlestep), but multistep version is better than singlestep and can achieve the best results.
>
> *Q4: The estimated $s_\lambda$ seems to be unstable. Would moving average help with the sampling algorithm?*
>
> **A**: We want to clarify that the unstable $s_\lambda$ is not an issue, and **our sampler is stable**:
> - The instability of $s_\lambda$ on ScoreSDE is intrinsic and not due to the estimation error. As we increase the number of samples to decrease the estimation error, the fluctuation is not reduced. We attribute it to the periodicity of trigonometric functions as stated in Sec.4.2.
> - Moreover, as we only consider the integrals of $s_\lambda$ and the sign of $s_\lambda$ does not change much, the integrals of $s_\lambda$ is rather smooth, which ensures the stability of our method.
>
> Therefore, it is no need for smoothing since the involved form is not $s_\lambda$ but the integral of $s_\lambda$, and it's actually already smooth.
>
> *Q5: Compare the FID score / CLIP score on Stable-Diffusion since low MSE does not necessarily imply better sample quality.*
>
> **A**: We choose the MSCOCO2014 validation set as reference, and compute the FID/CLIP score with 10k samples of cfg scale 7.5. See detailed results in *response pdf, Table 2*.
>
> The results show that our methods achieve consistently better FID and similar CLIP scores. **Notably, we achieve an FID of 15.4 in 8 NFE, close to the reported FID of Stable-Diffusion v1.4**.
>
> Still, we claim that FID is not a proper metric for evaluating the convergence of latent-space diffusion models. As stated in DPM-Solver++, The FIDs quickly achieves 15.0~16.0 within 10 steps, even if the latent code does not converge, because of the strong image decoder. Instead, MSE in the latent space is a direct way to measure the convergence. By comparing the MSE, our sampler does converge faster to the ground-truth samples of Stable Diffusion itself.
>
> *Q6: Compare DPM-Solver-V3 with stochastic samplers? Is it possible to integrate the solver in these stochastic samplers?*
>
> **A**:
> - **Stochastic samplers have bad performance for NFE <= 20**. SDE solvers are good at best-quality sampling, not fast sampling. In the papers of SDE solvers, the reported NFE is usually greater than 100, and the sample quality is quite low at around 20 NFE, even using the recent advanced SDE solver (Appendix E.7 in [1]).
> - Yes, our EMS can also be used in SDE solvers. Note that previous SDE solvers also limit their parameterization to noise/data prediction, and we can find better parameterization by the proposed EMS. Therefore, It is promising to improve SDE solvers with our EMS for both fast and best-quality sampling.
>
> [1] Gonzalez, Martin, et al. "SEEDS: Exponential SDE Solvers for Fast High-Quality Sampling from Diffusion Models."
>
> Once again, thank you for your constructive feedback and for considering our paper for acceptance. We'll revise our paper according to your suggestions.

---

> > ### Author Response · Authors · 2023-08-17
> > **Looking forward to further feedback**
> >
> > Dear Reviewer wxE7,
> >
> > We thank you very much again for the great efforts on reviewing our manuscript and providing the valuable comments to further improve. We hope you may find our response satisfactory and increase the rating accordingly.
> >
> > If you have any further feedback, we would be very happy to reply.
> >
> > Best,
> > Authors

---

### Official Review · Reviewer_oHv2 · 2023-07-07

**Soundness:** 3 good
**Presentation:** 3 good
**Contribution:** 2 fair
**Rating:** 4
**Confidence:** 3

**Summary:**

This work proposes a method to speed up the DPM solver by using empirical model statistics (pre-trained model) as well as employing multistep methods with a predictor-corrector framework for improving sample quality. Promising results were reported.

**Strengths:**

- This work addresses an important problem of fast sampling in diffusion models without retraining.
- The proposed parametric model is simple and seems neat with optimality criteria.

**Weaknesses:**

- The improvement of the proposed method over prior works such as DPM-solver++ seems incremental. It is unclear if the proposed empirical model statistics is good enough. While this work discusses an optimality under this parametric model, it is unclear if it is truly optimal in terms of modeling itself. This work may also discuss the prior work such as [H Zheng et al., Fast Sampling of Diffusion Models via Operator Learning, ICML 2023] and its arXiv version, which was able to achieve incredible results with only a single model evaluation.
- The experiment results seem to need more improvements and should ensure fairness. For example, Figures 3, 4 indicate that the proposed method is faster than other methods in early iterations, but in order to ensure reasonable FID, one may need a certain number of NFE and in those cases, the differences of FIDs look negligible. Moreover, the comparisons in Fig 6 seems unfair - will other methods achieve similar image quality and FID scores with 15-30% more evaluations? Incorporating the factor of computation should be incorporated for fair comparisons.

**Questions:**

Please address the comments in the Weaknesses section.

- Will the speed-up of 15-30% have a practical value? Please discuss.

**Limitations:**

Yes

---

> ### Author Rebuttal · Authors · 2023-08-09
>
> Thank you for recognizing our work's importance and theoretical neatness. However, we feel that there are some fundamental misunderstandings. We will address your questions and concerns below. We kindly request that you consider raising the score accordingly if you are satisfied.
>
> *W1: The improvement seems incremental. Discuss with [H Zheng et al. 2023].*
>
> **A**: First of all, **we respectfully disagree with the comments on incremental results**. We appreciate you for informing us of the nice work of [H Zheng et al. 2023] and will add proper citations in related works, but we have to point out that [H Zheng et al. 2023] is a distillation-based method, different from what our work focuses on. We would like to make the following clarifications:
> 1. We achieve the **most SOTA sample quality among all the training-free sampling methods** on a fair benchmark, under various settings, against the most recent and strong baselines.
> 2. **Our method is training-free samplers, while [H Zheng et al. 2023] needs heavy training of an additional generative model.** In general, although distillation-based methods can achieve good performance even in one step, it has obvious flaws (as discussed in Appendix A.1) such as onerous extra training and lose of information. Moreover, **their application scope is usually limited to unconditional cases, and it is much costly to distill a text-to-image model**. Since our training-free approach **can be easily applied to large text-to-image models** to boost the sampling with negligible extra cost, while the distillation-based methods such as [H Zheng et al. 2023] needs much more costs to do so, we argue that our work has high practical value.
>
> *W2: It is unclear if the proposed empirical model statistics is good enough. It is unclear if the parameterization is truly optimal in terms of modeling itself.*
>
> **A**: The proposed empirical model statistics can greatly speed up the convergence and improve the sample quality, especially within 5~10 NFEs. **It has already achieves the SOTA performance among all training-free fast samplers**.
>
> Moreover, we did not claim the optimality of the modeling parameterization. We **first** systematically study the parameterization of exponential integrators for diffusion models and **first** explain the superiority of previous data-pred over noise-pred (Appendix A.2). Also, this parameterization is elegantly designed by Rosenbrock-type methods and elucidating the design space of previous parameterizations. We understand that a better form of parameterization may be derived, but our introduced parameterization is already novel, insightful and efficient.
>
> *W3: Experiment results does not have much improvement and unfair.*
>
> **A**: **We respectfully disagree with the comments on unfairness and poor results**. We would like to make the following clarifications:
>
> 1. **The extra computation is negligible**. Please refer to *common response, Q1*.
> 2. **Our evaluation is fair**. We feel that the expression "early iterations" implies a critical misunderstanding: when users generate images with a pretrained diffusion model, they predetermine a certain NFE. Therefore, comparing FID at the same NFE, or comparing NFE for the same FID is completely fair, and **are adopted by all the previous fast diffusion sampling methods**.
> 3. **Our method converges much faster**. Our method converges 15-30% faster than previous samplers, thus saving 15-30% costs. **On a fair benchmark, under various settings, we achieve the best convergence speed among all the training-free samplers (Appendix F/H)**. A notable example is that, on LSUN-Bedroom, we reach the FID of 3.06 with 12 NFE, while the previous best method requires 20 NFE, which means our computation cost is approximately 60%.
>
> *Q1: Will the speed-up of 15-30% have a practical value?*
>
> **A**: Yes, the practical value embodies the following aspects:
>
> 1. **15~30% speed-up can save lots of costs for online text-to-image applications**. As we have clarified, our speed-ups are applicable to large text-to-image diffusion models, which are an important part of today's AIGC community. As the recent models become larger, a single NFE requires more computational resources, and 15-30% speed-up can save a lot of the companies' expenses for commercial usage.
> 2. **Improvement in 5~10 NFEs benefits image previewing**. Since the samples are controlled by the random seed, coarse samples with 5~10 NFEs can be used to ***preview*** thousands of samples with low costs and give guidance on choosing the random seed, which can be then used to generate fine samples with best-quality sampling strategies. This is especially useful for text-to-image generation. Since our method achieves better quality and converges better to the ground-truth sample, it can provide better guidance when used for preview.
>
>
> Finally, **we respectfully disagree with the comments on limited contribution**. We would like to emphasize our main contributions as follows:
>
> - Theoretically, we are the **first** to systematically study the parameterization problem of exponential integrators for diffusion sampling, the **first** to explain the superiority of previous parameterizations, and the **first** to introduce Rosenbrock-type methods into diffusion sampling. We also propose the pseudo-order solver and the half-corrector technique, which are **novel, insightful and efficient**. These ideas may inspire future works for training-free sampling of diffusion models, such as SDE-based samplers.
> - Empirically, we achieve the **most SOTA sample quality among all the training-free sampling methods** on a fair benchmark, under various settings, against the most recent and strong baselines. Our methods have **high practical value** in the AIGC community regarding the performance at small NFEs and the applicability to the text-to-image Stable Diffusion, for the usage of **generating previews** and high-quality samples of text-to-image diffusion models with low costs.

---

> > ### Author Response · Authors · 2023-08-17
> > **Looking forward to further feedback**
> >
> > Dear Reviewer oHv2,
> >
> > We thank you very much again for the great efforts on reviewing our manuscript and providing the valuable comments to further improve. We hope you may find our response (as well as other reviews) satisfactory and increase the rating accordingly.
> >
> > If you have any further feedback, we would be very happy to reply.
> >
> > Best,
> > Authors

---

> > ### Comment · Reviewer_oHv2 · 2023-08-20
> >
> > I would like to thank the authors for detailed responses. I increased my scores to +1 (in contribution / in overall score).

---

> > > ### Author Response · Authors · 2023-08-20
> > > **Thank you**
> > >
> > > We appreciate it very much that you acknowledge our contribution and increase the score accordingly. We wonder if our response has addressed your concerns, and we are happy to give further replies.
> > >
> > > Best, Authors

---

### Official Review · Reviewer_yHtS · 2023-07-07

**Soundness:** 3 good
**Presentation:** 3 good
**Contribution:** 3 good
**Rating:** 6
**Confidence:** 2

**Summary:**

This work proposes DPM-Solver-v3, an ODE solver for diffusion model inference. The method builds upon existing formulations involving exponential integrators, where the linear term of the ODE is canceled and only the noise predictor needs to be approximated. A new model parameterization is derived which involves an optimal set of coefficients, empirical model statistics (EMS). The authors describe methods for computing the EMS in practice, as well as a pseudo high-order method in the few NFE regime and a half-corrector technique when the guidance scale in guided sampling is large.

**Strengths:**

- Although the method builds upon existing methods involving exponential integrators in the diffusion literature, the model parameterization involving EMS is novel.
- Experiments and ablations are thorough and clearly done. Results are compelling and the proposed method consistently outperforms existing ODE solvers across datasets/models, especially in few NFE regime.
- Theoretical results for local/global error are clearly presented and assumptions seem reasonable.

**Weaknesses:**

- In the experiments, the authors mention that "we tune the strategies of whether to use pseudo-order predictor/corrector at each NFE on CIFAR10" (lines 254-255). This seems to be an important caveat and should be reiterated in the appendix where the detailed empirical results are presented (Table 4).
- Similarly, for Stable Diffusion, s=7.5, the authors mention that the best results among no/half/full corrector are reported. This should be reiterated in the appendix in Table 7.
- In general, I wonder if the presentation of the paper would be more compelling if some of the quantitative results could be moved to the main body from the appendix.

**Questions:**

- FID degenerating from 12.76 to 15.91 for ScoreSDE on CIFAR10 (line 822, G.3) in whether or not to use pseudo-order predictor/corrector seems quite substantial. In which cases is the pseudo-order predictor necessary? It might be compelling to produce a table similar to Table 10 (pseudo-order corrector experiments) examining the influence of the pseudo-order predictor.
- How does choice of N, K for the EMS change as the timestep scheduler changes?

**Limitations:**

The authors adequately address the limitations of the work.

---

> ### Author Rebuttal · Authors · 2023-08-09
>
>
> Thank you for your detailed review, especially for reading the proofs and ablations in the Appendix. Below we provide detailed responses for the questions and clarify misunderstandings. We kindly request that you consider raising the score accordingly if you are satisfied.
>
> *W1 & W2: Some tuning strategies should be reiterated in the Appendix (Table 4/Table 7).*
>
> **A**: Thank you for carefully reading the Appendix! We fully agree that although they have been mentioned in the main text, they should be reiterated where the quantitative table results are presented. We will revise relative paragraphs in the final version.
>
> *W3: Some of the quantitative results could be moved to the main body from the appendix.*
>
> **A**: Thank you for the valuable suggestion. Due to the page limitation of 9, we will move some of them to the main body in the final version.
>
> *Q1: In which cases is the pseudo-order predictor necessary? It might be compelling to produce a table similar to Table 10 (pseudo-order corrector experiments) examining the influence of the pseudo-order predictor.*
>
> **A**: The pseudo-order predictor only provided improvement in our mentioned case (5 NFE on CIFAR10). It can be seen as a specialized trick at 5 NFE on CIFAR10.
>
> *Q2: How does choice of N, K for the EMS change as the timestep scheduler changes?*
>
> **A**: The choice of $N,K$ is **disentangled** with the timestep scheduler in sampling: once we have estimated the EMS at $N$ (e.g., 1200) timesteps, they can be **flexibly adapted** to any schedule (uniform logSNR/uniform t...) in sampling, by corresponding the actual timesteps during sampling to the $N$ bins. The choice of $N,K$ and timestep scheduler are dependent on the pretrained model. See more discussions in *common response, Q2*.
>
> Once again, thank you for your constructive feedback and for considering our paper for acceptance. We'll revise our paper according to your suggestions.

---

> > ### Comment · Reviewer_yHtS · 2023-08-19
> >
> > Thanks to the authors for the detailed responses. I will retain my score of weak accept.

---

> > > ### Author Response · Authors · 2023-08-19
> > > **Thank you**
> > >
> > > We are happy to hear that you find our response satisfactory and are positive on the rating! We appreciate it that you are one of the very few reviewers who have read our Appendix when writing the reviews. We are more than willing to discuss if you have further questions.
> > >
> > > Best, Authors

---

### Official Review · Reviewer_dqUr · 2023-07-12

**Soundness:** 2 fair
**Presentation:** 3 good
**Contribution:** 3 good
**Rating:** 6
**Confidence:** 5

**Summary:**

This paper proposed an improved/generalized version of DPM-solver. It include three more coefficients based on the semi-linear ODE solution proposed by DPM-solver, accounting for minimizing the 'linearity' for the nonlinear component of the ODE and minimizing the first-order discretization error respectively. Such optimal coefficients can be approximately calculated by Monte Carlo samples from the learned model. Together with multistep formulation, predictor-corrector framework by UniPC, and some practical techniques, the paper empirically demonstrated the proposed sample can outperform existing ones especially when NFE is very small.

**Strengths:**

- The paper is well-written and easy to follow.
- The paper built on DPM-solve formulation and generalized it with additional coefficients that are optimized based on trained model. The added coefficients are intuitive. Theoretical analysis has been carried out to ensure local/global convergence.
- Two practical techniques has been proposed and seems to be useful in general.
- Extensive experiments have been performed to justify the effectiveness.

**Weaknesses:**

- I'm not convinced by optimizing $s_\tau$ with Eq. (11):
  - In equation 10, the first coefficient inside the big integration should be $e^{\int_{\lambda_s}^\lambda (l_\tau + s_\tau) d\tau}$ instead of  $e^{\int_{\lambda_s}^\lambda l_\tau d\tau}$. Therefore, the first-order discretization error not only depends on $f_\theta^{(1)} - s_\lambda f_\theta - b_\lambda$. There could be a chance that Eq. (11) is minimized but this term $e^{\int_{\lambda_s}^\lambda (l_\tau + s_\tau) d\tau}$ is amplified.
  - Main results shown in this paper are not really by first-order solver but by higher-order solvers. A justification of why minimizing first-order discretization error can help higher-order solvers is needed.

- A downside of the introduced coefficients is that the integrals in Eq. (14) become intractable, which has to be approximated by trapezoidal rule. It could introduce extra approximation error and add computational cost. I'd like to see a detailed analysis on the extra error and computational cost.

-  It is understandable and acceptable that this work is heavily built on DPM-solver, but certain sentences in this paper is duplicated from DPM solver. E.g., line 94 our first key insight is..., line 105 our second key insight is ..., line 135-136: the proposed solvers and analysis ....The paper should at least rephrase these sentences.

- The paper can do a better job on clarifying which is the unique contribution of this paper while which part has been proposed by previous works. For example, most techniques and theoretical conclusions in section 3.2 have already been developed in DPM-solver / DPM-solver++ / UniPC, which should be further admitted and clarified.

**Questions:**

- Why EMS without guidance can be directly applied to model with guidance? Does it still hold if the guidance weight is really high?
- Why for conditional setting, the best setting is 2nd-solver instead of higher-order counterparts?

**Limitations:**

I'd expect to see more discussion on limitations due to the intractable integrals.

---

> ### Author Rebuttal · Authors · 2023-08-09
>
> Thank you for your positive comments. Below we provide detailed responses for the questions and clarify misunderstandings. We kindly request that you consider raising the score accordingly if you are satisfied.
>
> *W1: Equation 10 is wrong, the first-order discretization error may be amplified.*
>
> **A**: **We respectfully disagree. This is a misunderstanding**. Specifically, by the definition of $g_\theta$, we have
>
> $$
> g_\theta^{(1)}(x_\lambda,\lambda) = e^{-\int_{\lambda_s}^{\lambda}s_\tau d\tau}\left(
>         f_\theta^{(1)}(x_\lambda,\lambda)-s_\lambda f_\theta(x_\lambda,\lambda)
>         - b_\lambda
>     \right)
> $$
>
> By using the Taylor expansion $g_\theta(x_{\lambda_s},\lambda_s) = g_\theta(x_{\lambda},\lambda ) + (\lambda_s-\lambda)g_\theta^{(1)}(x_{\lambda},\lambda) + O((\lambda-\lambda_s)^2)$ stated in the paper (**expand it at each $\lambda$**), we can see that the term $e^{\int_{\lambda_s}^{\lambda}s_\tau d\tau}$ in Eq. (9) is cancelled by the term $e^{-\int_{\lambda_s}^{\lambda}s_\tau d\tau}$ in $g_\theta^{(1)}(x_\lambda,\lambda)$, and we obtain exactly Eq. (10).
>
> *W2: A justification of why minimizing first-order discretization error can help higher-order solvers is needed*.
>
> **A**: Sure. Below we provide the corresponding theoretical justification about how minimizing first-order discretization error helps high-order cases.
>
> We assume that the EMS are bounded (Assumption B.2 in Appendix B.1.1), which is empirically confirmed as in Section 4.2. Here we take the 2nd-order case as an example. By Eq. (52) in Appendix B.4, the 2nd-order local error is
>
> $$
> \hat x_t-x_t=\alpha_tA(\lambda_s,\lambda_t)\int_{\lambda_s}^{\lambda_t}E_{\lambda_s}(\lambda)(g_\theta(x_\lambda,\lambda)-g_\theta(x_{\lambda_s},\lambda_s))d\lambda-\frac{\alpha_tA(\lambda_s,\lambda_t)}{\lambda_{i_1}-\lambda_s}(g_\theta(x_{\lambda_{i_1}},\lambda_{i_1})-g_\theta(x_{\lambda_s},\lambda_s))\int_{\lambda_s}^{\lambda_t}E_{\lambda_s}(\lambda)(\lambda-\lambda_s)d\lambda
> $$
>
> which is equivalent to
>
> $$
> \alpha_tA(\lambda_s,\lambda_t)\int_{\lambda_s}^{\lambda_t}e^{\int_{\lambda_s}^{\lambda}(l_{\tau}+s_{\tau})d\tau}\left(\int_{\lambda_s}^{\lambda}g^{(1)}\_\theta(x_{\tau},\tau)d\tau\right)d\lambda-\frac{\alpha_tA(\lambda_s,\lambda_t)}{\lambda_{i_1}-\lambda_s}\int_{\lambda_s}^{\lambda_t}E_{\lambda_s}(\lambda)(\lambda-\lambda_s)d\lambda\int_{\lambda_s}^{\lambda_{i_1}}g^{(1)}\_\theta(x_{\lambda},\lambda)d\lambda
> $$
>
> Which is controlled by $\|g_\theta^{(1)}\|$. Other terms are only dependent on the EMS and are bounded. For higher orders, the derivation is similar, since they all involve the form $g_{\theta,t_2}-g_{\theta,t_1}$, which is the integral of $g_\theta^{(1)}$ . We will put the detailed analysis in the revised paper.
>
> *W3: A detailed analysis on the extra error and computational cost of the integrals w.r.t. the introduced coefficients.*
>
> **A**: Below we provide a detailed analysis correspondingly.
> - **The computational cost is ignorable**: See *Common response, Q1*.
> - **The extra error of trapezoidal rule is under higher order and ignorable**: We assume that the EMS and their derivatives are bounded, then by the error bound formula of trapezoidal rule,
> $$
> |E|\leq \frac{(b-a)^3}{12n^2}\max |f''(x)|
> $$
> We can easily conclude that the errors of $\int_{\lambda_s}^{\lambda_t}E_{\lambda_s}(\lambda)B_{\lambda_s}(\lambda)d\lambda$ and $\int_{\lambda_s}^{\lambda_t}E_{\lambda_s}(\lambda)\frac{(\lambda-\lambda_s)^k}{k!}d\lambda$ in Eq. (14) are $O(h_0^2h)$ and $O(h_0^2h^{k-1})$ respectively, where $h_0$ is the stepsize of EMS discretization, and $h=\lambda_t-\lambda_s$. We'll put detailed analysis in the revised paper.
>
> *W4: Certain sentences is duplicated from DPM solver.*
>
> **A**: Thank you for pointing it out. We take these expressions since they concisely emphasize our motivation, novelty and how we get inspiration from previous works, and did not notice the duplicated expressions. We will rephrase them in the revised paper.
>
> *W5: Clarifying the unique contribution in section 3.2.*
>
> **A**: Thank you for the valuable suggestion. We coarsely admit that by saying *"...highly motivated by ... in the field of diffusion models"* at the beginning of Section 3.2, and the specific technical details are different. We will further clarify detailedly at each part of Section 3.2 to seperate the unique contributions of ours from the previous works in the revised version.
>
> *Q1: Why EMS without guidance can be directly applied to model with guidance? Does it still hold if the guidance weight is really high?*
>
> **A**: It is because the unconditional model contains some common information, please refer to *common response, Q2* for details.
>
> However, it no longer holds for extremely large guidance scale (e.g., cfg scale 15 for Stable Diffusion), since in this case the condition has a large impact on the denoising process. Note that at these extreme large scales, the sampling quality is very low (compared to cfg scale 7.5) and they are rarely used in practice. Therefore, our proposed EMS without guidance is suitable enough for the common applications with the common guidance.
>
> *Q2: Why for conditional setting, the best setting is 2nd-solver instead of higher-order counterparts?*
>
> **A**: The high-order solvers are more unstable compared to low-order, especially for large guidance scales[1]. However, a slightly large guidance scales are often preferred in practice to improve the condition-sample alignment, as shown in Imagen, Stable Diffusion. Therefore, we and previous works (e.g., DPM-Solver++, UniPC) all find that 3rd-solver will produce artifacts in conditional setting, and 2nd-solver is better.
>
> [1] Wizadwongsa, Suttisak, et al. "Diffusion Sampling with Momentum for Mitigating Divergence Artifacts.".
>
> Once again, thank you for your constructive feedback and for considering our paper for acceptance. We'll revise our paper according to your suggestions.

---

> > ### Comment · Reviewer_dqUr · 2023-08-16
> >
> > Thank the authors for the detailed rebuttal. My questions were well address as long as the paper will be further revised as the authors promised in the rebuttal. I tend to retain my original score of leaning towards acceptance.

---

> > > ### Author Response · Authors · 2023-08-17
> > > **Thank you**
> > >
> > > We are happy to hear that you find our response satisfactory and are positive on the rating! We will definitely further revise in the final version as promised.
> > >
> > > Best,
> > > Authors

---

### Author Rebuttal · Authors · 2023-08-10

We sincerely thank all reviewers' effort for the detailed and insightful suggestions. First we'd like to present all the paper revisions.

### Summary of paper revisions

**Theories**

- Theoretical justification of why minimizing first-order discretization error can help higher-order solvers (Reviewer dqUr, W2)
- Theoretical analysis on the extra error of the trapezoidal rule (Reviewer dqUr, W3)

**Experiments**

- DPM-Solver-v3’s performance affected by single-step versus multistep sampling (Reviewer wxE7, Q3)
- FID score / CLIP score on Stable-Diffusion (Reviewer wxE7, Q5)

**Writing**

- Rephrase the duplicate expressions from DPM-Solver (Reviewer dqUr, W4)
- Clarify which is the unique contribution in Section 3.2 (Reviewer dqUr, W5)
- Reiterate the tuning strategies in Appendix Table 4/Table 7 (Reviewer yHtS, W1/W2)
- Moving the quantitative results to the main body from the appendix (Reviewer yHtS, W3)
- Clarify the fluctuation of $s_\lambda$ is not due to estimation error, and there is no need for moving average or other smoothing methods (Reviewer wxE7, W2/Q4)
- Discuss why the EMS of unconditional model can be applied to guided sampling (Reviewer dqUr/wxE7, Q1)
- Discuss why 2nd-order is better for conditional setting (Reviewer dqUr, Q2)
- Discuss the practical value of 15\~30% speed-up and improvement in 5\~10 NFE (Reviewer oHv2, Q1)

We find there are common misunderstandings to our paper, and we'd like to clarify here again.

### Response for common questions

*Q1: Extra computation/memory costs of the proposed method? (from dqUr,oHv2,wxE7,wxE7)*

**A**: **The extra memory cost is rather small**. The extra coefficients are discretized and computed at $N$ timesteps, each with a dimension $D$ same as the diffused data. The extra memory cost is $O(ND)$, including the precomputed terms in Appendix C.1.2, and is rather small compared to the pretrained model (e.g. only ~125M in total on Stable-Diffusion, compared to ~4G of the model itself).

**The pre-computation time for estimating EMS is rather short**. The EMS introduced by our method can be effectively estimated on around 1k datapoints within hours, which is rather short comparing to the long training / distillation time of other methods. Moreover, the integrals of these extra coefficients are just some vector constants which can be pre-computed **within seconds**, as shown in Appendix C.1.2. The precomputing is done only once before sampling.

**The extra computational overhead during sampling is negligible**. Once we obtain the estimated EMS and their integrals at discrete timesteps, they can be regarded as constants. Thus, during the subsequent sampling process, **the computational overhead is the same as previous training-free methods** (such as DPM-Solver++) with negligible difference (Appendix E).

*Q2: How does EMS apply for different time schedules and guided sampling? (from dqUr,yHtS,wxE7)*

**A**: The pre-computed EMS can be applied for any time schedules during sampling without re-computing EMS. Besides, we compute EMS on unconditional models and it can be used for a wide range of guidance scales (such as cfg=7.5 in Stable Diffusion). In short, EMS is flexible and easy to adopt in downstream applications.

- **Time schedule**: The choice of $N,K$ in EMS is **disentangled** with the timestep scheduler in sampling. Once we have estimated the EMS at $N$ (e.g., 1200) timesteps, they can be flexibly adapted to any schedule (uniform logSNR/uniform t...) during sampling, by corresponding the actual timesteps during sampling to the $N$ bins. For different time schedule, we only need to re-precompute $E_{\lambda_s,\lambda_t}^{(k)}$ in Appendix C.1.2, and the time cost is within seconds.
- **Guided sampling**. We compute the EMS on unconditional model for all guided cases. Empirically, the EMS computed on the model without guidance (unconditional part) performs more stably than those computed on the model with guidance, and can accelerate the sampling procedure in a wide range of guidance scales (including the common guidance scales used in pretrained models). We think that the unconditional model contains some common information (image priors) for all the conditions, such as color, sketch, and other image patterns. Extracting them helps correct some common bias such as shallow color, low saturation level and lacking of details. In contrast, the conditional model is dependent on the condition and has large variance.
    - In addition, EMS computed on models without guidance cannot work for extremely large guidance scale (e.g., cfg scale 15 for Stable Diffusion), since in this case the condition has a large impact on the denoising process. Note that at these extreme large scales, the sampling quality is very low (compared to cfg scale 7.5) and they are rarely used in practice. Therefore, our proposed EMS without guidance is suitable enough for the common applications with the common guidance.

We think the the quality of our work has been improved a lot. We are welcome for further questions.

---

### Decision · Program_Chairs · 2023-09-21

**Decision:**

Accept (poster)

**Comment:**

This paper proposes an improved/generalized version of DPM-solver with additional coefficients that are optimized based on trained model. Theoretical analysis has been carried out to ensure local/global convergence, and extensive experiments have been performed to justify the effectiveness. Reviewers are overall positive on the soundness, presentation, and contribution of this work.